# Extratropical circulation associated with Mediterranean droughts during the Last Millennium in CMIP5 simulations

Woon Mi Kim[1,2,3], Santos J. González-Rojí[1,2], and Christoph C. Raible[1,2]

[1]Climate and Environmental Physics, University of Bern, Bern, Switzerland
[2]Oeschger Centre for Climate Change Research, University of Bern, Bern, Switzerland
[3]Now at: National Center for Atmospheric Research, Boulder Colorado, United States

**Correspondence:** Woon Mi Kim (wmikim@ucar.edu)

**Abstract.**

The Mediterranean region is expected to experience significant changes in hydroclimate, reflected in increases in the duration and severity of soil moisture droughts. While numerous studies have explored Mediterranean droughts in coupled climate models under present and future scenarios, understanding droughts in past climate simulations remains relatively underexplored. Such simulations can offer insights into long-term drought variability that observational records cannot capture. Therefore, our study investigates circulation patterns in the Euro-Atlantic domain associated with multi-year soil moisture droughts over the Mediterranean region during the last millennium (850–2005 CE) in climate simulations. For this, we use the fifth phase of the Climate Model Intercomparison Project - Paleoclimate Model Intercomparison Project Phase 3 (CMIP5-PMIP3) and the CESM Last Millennium Ensemble Project. Primarily, we examine the differences among the models in representing drought variability and related circulation patterns. For the analysis, we exclude the anthropogenic trends from 1850–2005 CE, and to detect the circulation patterns, we perform k-means clustering combined with linear correlation analyses.

The findings confirm that Mediterranean drought occurrence during the last millennium is associated with internal variability of the climate system. Drought variability, the associated circulation patterns, and the frequencies of these patterns vary across the models. Some climate models exhibit a multi-decadal anti-phase occurrence of some drought periods between the western and eastern Mediterranean regions, although the exact periods of coherence differ among the models. This anti-phase co-variability, which agrees with some proxy records, can be explained by the dominant circulation patterns in each region detected by the models: western Mediterranean droughts are dominated by a high-pressure system over central Europe and a North Atlantic Oscillation (NAO)-like pattern, while eastern Mediterranean droughts are linked to positive pressure anomalies in the southern and eastern Mediterranean, negative NAO, Eastern Atlantic, and Eastern Atlantic-Western Russia-like patterns. The frequencies of these modes of climate variability are strongly model-dependent, i.e., some patterns occur more frequently or only in some models, suggesting that the main drivers of droughts differ among the models. Although it is complicated to evaluate the representation of droughts and associated circulation among the models, in general, the models with lower horizontal and vertical spatial land resolutions exhibit drought variability and patterns that distinctly differ from other models. These model differences and preferences toward some circulation patterns can be a source of uncertainties in the model-proxy comparison of Mediterranean droughts and potentially influence future climate projections.

# 1 Introduction

Droughts are recurrent climate events in the Mediterranean region where precipitation shows a high spatial and interannual variability (Lionello et al., 2006). The complex spatial and temporal characteristics of Mediterranean precipitation and hydroclimate are due to its geographical location, extending latitudinally from the semi-arid subtropics to temperate mid-latitudes. The region is influenced by seasonal variations of the subtropical high in the Atlantic Ocean and other teleconnection patterns (Rodwell and Hoskins, 1996; Krichak and Alpert, 2005; Lionello et al., 2006). While considerable spatial and temporal variability exists across the Mediterranean region, the climate is characterized by a wet cold season and a dry hot season (Xoplaki et al., 2004; Lionello et al., 2006). The wet cold season, typically extending from fall to spring, is a crucial period for moisture supply, where a large proportion of precipitation is provided by mid-latitude circulations and westerlies. Therefore, this season dictates the intensity and impacts of droughts on the region.

Although droughts are natural fluctuations of the region with highly variable seasonal hydroclimate conditions, future climate projections indicate the Mediterranean regions as a climate hotspot with strong changes in the hydroclimate conditions (Giorgi, 2006; Cook et al., 2020). The region is expected to experience robust increases in drought frequencies and intensities even under stricter mitigation scenarios (Lehner et al., 2017; Cook et al., 2020; Seneviratne et al., 2021). This change is principally attributed to the intensification of the global hydrological cycle caused by anthropogenic influences in the atmosphere (Douville et al., 2021). In the region, precipitation is expected to decrease due to an expanding Hadley Cell (Previdi and Liepert, 2007) and a poleward shift of the mid-latitude storm tracks (Yin, 2005; Wu et al., 2011). The most significant change will occur in the winter precipitation and its associated circulation, partially contributed by a decrease in the regional land-sea temperature gradient (Tuel and Eltahir, 2020). In addition to that, the future increase in temperature will amplify land-atmosphere feedback and vapor pressure deficit, which will potentially enhance the intensity of droughts (Zhou et al., 2019).

Up until now, circulations associated with future potential changes in hydroclimate, including persistent droughts, have been extensively studied in the region (e.g., Dubrovskỳ et al., 2014; Spinoni et al., 2020; Tramblay et al., 2020; Cos et al., 2022). However, how drought-associated circulations looked before the anthropogenic greenhouse gas (GHG) era remains comparably underexplored. Long past climate data enable us to examine persistent droughts, which are the conditions that can be comparable to the future drying over the region, and their long-term variability that cannot be captured in the limited-length observational-based records. Among the past periods, the last millennium is particularly interesting as it is a relatively close period to the present day. The period is characterized by typical natural forcing of the pre-industrial conditions, with minimal changes, such as those in orbital parameters. In addition, the last millennium also presents abundant high-quality proxy-based reconstructions (PAGES Hydro2k Consortium et al., 2017).

Several proxy-based reconstructions of past hydroclimate have contributed to a better understanding of drought variability and their associated climate modes during the last millennium (PAGES Hydro2k Consortium et al., 2017). One of the widely used proxy-based datasets is a tree ring-based reconstruction of summer dryness and wetness in Europe, known as the Old World Drought Atlas (OWDA; Cook et al., 2015). Using OWDA, Cook et al. (2016) identified a multidecadal in-phase drought occurrence across the Mediterranean basin and an anti-phase occurrence within the eastern Mediterranean. This result con-

tradicts the finding by Roberts et al. (2012) based on the lake sediment records, which has shown an anti-phase hydroclimate variability between the western and eastern Mediterranean from the Medieval Climate Anomaly period. Different proxy-based studies have indicated that the Mediterranean hydroclimate's variability is influenced by several modes of climate variability and atmospheric circulation, such as Eastern Atlantic Pattern (EA; Cook et al., 2015), and North Atlantic Oscillation (NAO; Cook et al., 2015; Baek et al., 2017; Markonis et al., 2018). In a long-term analysis for the entire last millennium, Markonis et al. (2018) found that over southern Europe, including the Mediterranean, drier conditions have become dominant in the 20th century, exceeding the millennial hydroclimate boundaries. Regarding natural external forcing, volcanic eruptions that are the natural radiative forcing with the strongest impact on the global climate during the last millennium, have caused wetter conditions that last up to a few years after the first emissions (Gao and Gao, 2017; Rao et al., 2017).

Besides proxy-based reconstructions, fully coupled climate models have been useful tools to support findings from proxy-based studies. Since climate simulations can cover the temporal resolutions and time periods that go beyond the time scales of proxy records, they can allow us a broader exploration of the variability and mechanisms of past hydroclimate and droughts. For instance, the impact of volcanic eruptions on hydroclimate-related variables and long-lasting droughts has been assessed on a global scale (Stevenson et al., 2017; Roldán-Gómez et al., 2020) and the Mediterranean region (Kim and Raible, 2021) using the climate simulations from the Community Earth System Model (CESM; Lehner et al., 2015; Otto-Bliesner et al., 2016) and the fifth phase of the Climate Model Intercomparison Project (CMIP5; Taylor et al., 2012) - Paleoclimate Model Intercomparison Project Phase 3 (PMIP3; Schmidt et al., 2012). In the latter study, Kim and Raible (2021) found that persistent Mediterranean droughts in the last millennium are associated with modes of internal climate variability, with a strong influence of positive NAO and higher than normal regional temperatures in a CESM1 simulation. Other studies have drawn a similar conclusion. On a global scale, Roldán-Gómez et al. (2020) indicated that there is not a clear relationship between soil moisture and external forcing that can be detected during the last millennium in the CMIP5-PMIP3 simulations. Using simulations from the CCSM4 and MPI-ESM climate models, Xoplaki et al. (2018) showed that multidecadal variations in the eastern Mediterranean hydroclimate are explained by internal climate dynamics. By comparing three historical periods with large hydroclimate events over the region, they found notable differences in the climate patterns during the same periods between the two models. The observed discrepancies in climate patterns and timing of hydroclimate events between the models and between the models and the proxy records indicate that exact temporal and spatial agreement of events between climate models and proxy records cannot be expected due to the model-dependent internal climate dynamics.

Despite these past studies on the Mediterranean hydroclimate, temporal variability and circulation patterns associated with multi-year persistent droughts during the last millennium are not fully explored in currently available climate simulations. Therefore, this study aims to examine how different coupled climate models depict the temporal variability of droughts and associated circulation patterns during the last millennium. Additionally, we evaluate the differences among the models in representing drought-related circulations. The focus is on the mid-latitude circulation patterns that have more impacts on the Mediterranean hydroclimate (Xoplaki et al., 2012; Kim and Raible, 2021). We examine circulation patterns in the Euro-Atlantic domain associated with the western (12°W–19°E, 32°–43°N) and eastern (19°–37°E, 32°–43°N) Mediterranean droughts

during 850–2005 CE using the model simulations from CMIP5-PMIP3 (Taylor et al., 2012; Schmidt et al., 2012) and the
CESM Last Millennium Ensemble Project (CESM-LME; Otto-Bliesner et al., 2016).

All the details of the datasets are provided in Section 2. Soil moisture anomalies are employed as a drought metric in this
study, and commonly used statistical techniques in climate sciences are applied to detect circulation patterns. The information
about these methods is presented in Section 3. Drought periods and main circulation patterns are identified in the climate model
simulations and compared to each other in Section 4. This study ends with a discussion of the results in Section 5, followed by
the concluding remarks in Section 6.

## 2 Data

### 2.1 CMIP5-PMIP3 model simulations

To study drought variability and the associated extratropical circulation patterns during 850–2005 CE, we use several forced
transient simulations of Last Millennium (LM; 850–1849 CE) and historical period (Hist; 1850–2005 CE) from the CMIP5-
PMIP3 (Schmidt et al., 2012; Taylor et al., 2012) and CESM-LME projects (Otto-Bliesner et al., 2016). Our focus is on droughts
that affect deep soil moisture conditions, known as soil moisture droughts (Dai, 2011). Therefore, we consider only simulations
that provide the vertical water content of each soil layer (variable names *mrlsl* in PMIP3-CMIP5, and *SOILLIQ* and *SOILICE*
in CESM-LME). With this criterion, only four CMIP5-PMIP3 climate models are used for the analysis: GISS-E2-R, CCSM4,
bcc-csm1-1, and MIROC-ESM. For CESM-LME, we use 12 ensemble members of CESM1 from members 2 to 13 available
in https://www.earthsystemgrid.org. The vertical water content of soil layers up to 70 cm depth is employed to quantify soil
moisture droughts. Surface temperature and geopotential height at 500 hPa are retrieved to describe the mean temperature and
circulation conditions during the Mediterranean drought years. All variables have a monthly temporal resolution but different
horizontal and vertical land resolutions, as shown in Table 1. Additional specifications of each model are summarized there.
All simulations are transient simulations and were run with the volcanic, solar, and GHG forcings of 850–2005 CE agreed
upon by the PMIP3-CMIP5 protocol (Schmidt et al., 2012; Taylor et al., 2012). For GISS-E2-R in the LM experiment, three
realizations (*i1r1p122*, *i1r1p125*, and *i1r1p128*) that were run with the same volcanic forcing are considered in the analysis.

**Table 1.** Details of the CMIP5-PMIP3 (Schmidt et al., 2012; Taylor et al., 2012) and CESM-LME (Otto-Bliesner et al., 2016) climate models and simulations used for the analysis. LM and Hist indicate the last millennium and historical experiments, respectively.

| Model | Number of Ensemble Members | | Land model | Horizontal resolution of the land model | Vertical soil layers up to 70 cm | Number of grid points | | Reference |
| --- | --- | --- | --- | --- | --- | --- | --- | --- |
| | LM | Hist | | | | West | East | |
| CESM1 | 12 | 12 | CLM4 | $1.875° \times 2.5°$ | 7 | 85 | 46 | Otto-Bliesner et al. (2016) |
| GISS-E2-R | 3 | 3 | ModelE Land surface model | $2° \times 2.5°$ | 3 | 42 | 24 | Schmidt et al. (2014) |
| CCSM4 | 1 | 3 | CLM4 | $1° \times 1.25°$ | 7 | 173 | 99 | Gent et al. (2011) |
| bcc-csm1-1 | 1 | 3 | BCC-AVIM1.0 | $2.8° \times 2.8°$ | 7 | 38 | 25 | Wu et al. (2014) |
| MIROC-ESM | 1 | 3 | MATSIRO + SEIB-DGVM | $2.8° \times 2.8°$ | 3 | 22 | 12 | Watanabe et al. (2011) |

## 2.2 Observation-based data

Since observations of subsurface soil moisture content are spatially scarce, we use soil moisture content from the Noah land surface model (NOAH-LSM), which is a part of the Global Land Data Assimilation System 2.0 (GLDAS2.0; Rodell et al.,
2004). NOAH-LSM is a physical land model that solves and quantifies the transfer of heat and moisture at the surface to the subsurface levels and interactions between the soil, atmosphere, and vegetation. The model is forced by atmospheric conditions from satellite- and ground-based observational data products. The soil moisture variable from NOAH-LSM has four layers that extend to a depth of two meters, with monthly temporal and $1°$ spatial resolutions extending from 1948 to the present.

To characterize the atmospheric circulation, we use the monthly mean geopotential height at 500 hPa from the ERA5 reanal-
125 ysis (Hersbach et al., 2018, 2020). ERA5 is the latest reanalysis product of the European Centre for Medium-Range Weather Forecasts (ECMWF), and the products are generated with the 2016 version of the ECMWF numerical weather prediction model and the integrated forecasting system Cy41r2 data assimilation. The spatial resolution of ERA5 is $0.25°$, and the temporal extent is from 1950 to the present. For the analysis, ERA5 data is horizontally interpolated to $1°$ spatial resolution to match the resolution of NOAH-LSM. We use the NOAH-LSM land variable instead of the one from ERA5, as NOAH-LSM
is forced with the biases-corrected observation-based datasets. Therefore, NOAH-LSM is a better choice that reflects more realistic present-day soil moisture variability.

## 3 Methods

### 3.1 Calculation of anomalies

The following variables from the NOAH-LSM, ERA5, and climate simulations are used in the analysis: the surface tempera-
135 ture, geopotential height at 500 hPa, and soil moisture content of vertical soil layers. The soil moisture content of soil layers

from NOAH-LSM and the climate models are vertically integrated to 70 cm. If the exact 70 cm level is unavailable, a linear interpolation is applied to estimate the vertically integrated soil moisture content at that level.

The variables are all transformed into annual mean anomalies. Wet seasons are critical for supplying moisture to the region and are associated with strong circulation patterns (Xoplaki et al., 2004; Lionello et al., 2006). Therefore, we use annual mean anomalies to capture the mean variability of hydroclimate throughout the entire year instead of focusing on some particular seasons, i.e., summer growing seasons. This also includes the influences of wet seasons in the analysis. The annual mean anomalies are calculated by subtracting the multi-year means from the annually averaged time series. Different reference periods are used to calculate the multi-year means depending on the data set and the experiment. For the observation-based data, NOAH-LSM and ERA5, the reference period is 1950–1979 CE. For the LM and Hist experiments in the climate models, the reference period is 850–1849 CE.

In the next step, we remove the strong unprecedented trends in the Hist simulations (1850–2005 CE). This means the effects of increased GHG on droughts are not included in the analysis. To achieve this, we calculate the ensemble means of the annual anomalies for each climate model during the period 1850–2005 CE. Then, for each model, we subtract their corresponding ensemble mean from each of the ensemble member anomalies. The approach follows a similar method to Maher et al. (2018), which aims to exclude a trend caused by increasing anthropogenic GHG concentration from a time series. Note that the method is not applied to the LM period. Therefore, the anomalies during LM still contain forced signals, such as those from volcanic eruptions. The number of ensemble members can influence the final output of Hist. The potential implications of this method are discussed in Section 5.

## 3.2 Region of study

Similar to Cook et al. (2016), we separate the Mediterranean into two subregions: the western Mediterranean encompassing 13°W–19°E, 32°–43°N and the eastern Mediterranean occupying 19°–37°E, 32°–43°N. The separation is motivated by the suggested influences of the circulation patterns over the Mediterranean region (Dünkeloh and Jacobeit, 2003; Lionello et al., 2006), such as the NAO, East Atlantic (EA), and Eastern Atlantic-West Russian (EA-WR). The western Mediterranean is more intensely influenced by NAO than the eastern region, and the eastern region is not only affected by NAO but also strongly linked with East Atlantic-type patterns.

## 3.3 Comparison between present-day observation-based data and model simulations

Pearson correlation coefficients (PCC) are calculated between the annual soil moisture anomalies (SOIL) of the western and eastern Mediterranean and the geopotential height anomalies (Z500) in the north Atlantic and European domain (70°W–70°E, 21°–85°N). SOIL is from NOAH-LSM and Z500 from ERA5, both variables calculated following Section 3.1. The PCC allows us to measure the link between soil moisture variability and circulation in the present day. The period for the correlation is 56 years from 1950 to 2005 CE. Assuming that NOAH-LSM realistically represents soil moisture variability, the correlation patterns of other climate models are compared against the pattern of NOAH-LSM.

In addition, to quantify the spatial similarity of the correlation fields obtained after the PCC between the observation-based data (NOAH-LSM–ERA5) and the climate models, the pattern correlations are calculated between the two datasets for the western and eastern Mediterranean regions. For this calculation, all data are interpolated to match the resolutions of the coarser climate models, which are bcc-csm1-1 and MIROC-ESM (Table 1). The pattern correlations assess the overall resemblance between the correlation field of NOAH-LSM–ERA5 and the climate models for the present day.

## 3.4 Drought definition

The annual anomalies of vertically integrated soil moisture at 70 cm (SOIL) in LM and Hist are used to quantify annual droughts. The 70 cm level is a deep soil level that can reflect the impacts of soil moisture change on vegetation and ecosystems, hence, better representing soil moisture droughts, including their persistent characteristics (Dirmeyer, 2011; Ghannam et al., 2016; Esit et al., 2021).

Two temporal and spatial criteria are applied to detect droughts over each of the study regions:

i) We combine two definitions used by Kim and Raible (2021) and Coats et al. (2013): A drought commences after two consecutive years of negative SOIL and continues until two consecutive years of positive anomalies occur (Coats et al., 2013). These two wet years are excluded from the drought period, ensuring that droughts consist only of negative SOIL (Kim and Raible, 2021). This definition guarantees the intensity of droughts by considering only negative SOIL without interruption by a particularly wet year in between. It also assures a minimum drought duration of two years and at least two wet years of separation between drought events. Using this definition, droughts are detected at each horizontal grid point in the study regions.

ii) A spatial restriction is applied to the detected drought years: at least 60% of all horizontal grid points (which is approximately from 56% to 61% of the areal extent) within the region (the western or eastern Mediterranean) need to be under negative SOIL conditions during all consecutive drought years. This criterion ensures that a substantial portion of the region experiences drought conditions and that droughts are regional-scale events and not local.

At this step, we do not apply any horizontal interpolation in SOIL. Thus, regional coverage (geographical extension and number of grid cells) differs slightly between the models (as shown in Table 1). The reason is that the hydroclimate variables associated with precipitation can be sensitive to the horizontal grid resolution (Champion et al., 2011; Kopparla et al., 2013; Haren et al., 2015). Also, in this way, the effects of the number of land grid cells on drought estimation and drought-associated circulation can be identified.

Finally, a drought is considered regional (only west- or east-occurring) when the temporal overlap between the two regions is less than 50% during one drought event. When the temporal overlap is more than 50%, the event is considered a pan-Mediterranean drought. The analysis focuses on only west and east-occurring droughts, and the reason is that pan-Mediterranean droughts are rather rare events to occur.

When these temporal and spatial criteria are fulfilled, droughts are identified separately for the west and east Mediterranean. The frequencies of droughts, in terms of the number of drought years, are estimated during the entire 850–2005 CE by combin-

ing the SOIL of LM and Hist experiments for each model and region. The number of drought years is counted with a moving window of a century, and a percentage of total drought years (years with droughts divided by the total model years) and a mean duration of droughts are calculated for each of the models and regions.

We perform wavelet coherence analysis (Grinsted et al., 2004) on the time series of the number of droughts and the time series of soil moisture anomalies (SOIL) between the west and east Mediterranean. For the latter, we generated the spatially weighted time series of SOIL for each of the regions. The wavelet coherence analysis aims to assess the phase relationship, dominant frequencies, and temporal variations of correlations in soil moisture anomalies and drought occurrences between these two regions.

## 3.5   Extratropical circulation pattern detection: principal component, k-mean clustering and Pearson correlation analyses

Drought-associated circulation patterns are detected in the simulations using the geopotential height anomalies at 500 hPa (Z500) over 70°W–70°E, 21°–85°N. For this, only Z500 during drought years are considered. All Z500 fields are interpolated to a consistent horizontal resolution matching that of CESM1 (Table 1).

The method for pattern detection consists of a combination of several standard statistical methods in climate sciences: a principal component (PCA), k-means clustering (KCA), and Pearson correlation analyses (PC-KCA-PCC method). The flow chart in Fig. 1 illustrates all the steps from the definition of drought (Step 1, Section 3.4) to the detection of associated circulation patterns.

A detailed explanation of the method is as follows: after obtaining all drought years in Step 1, in Step 2, PCA is applied to the Z500 anomalies during drought periods. PCA is employed to increase the performance of KCA in the next step, which is the primary tool for detecting circulation patterns in this study. Given a spatiotemporal field $U(t,l)$ where $l$ is the spatial dimensions ($latitude \times longitude$), and $t$ is the time steps in years ($t$ is the total drought years), the PCA decomposes the field into $M$ number of modes or principal components (PC) according to the following equation (Hannachi et al., 2007):

$$U = \sum_{i=1}^{M} \lambda_i \cdot a_i x_i^T \tag{1}$$

where $a_i$ is the i'th standardized PC of the U data, the $x_i$ is the i'th empirical orthogonal function of the original data (also the eigenvector), and $\lambda_i^2$ is the corresponding eigenvalue that represents the explained variance of the i'th PC ($a_i$). The resulting EOFs are orthogonal, and the PCs are uncorrelated.

By truncating the number of PCs according to a certain threshold of explained variance, PCA reduces the dimension of a multi-dimensional data set (Hannachi et al., 2007). We consider a threshold of 70% of the total explained variance for the truncation of PCs. The method substantially reduces the number of retained PCs from the total number $M$ to those until the truncation $N$, thereby enhancing the performance of KCA in the next step. The resulting new PC field of Z500 has a spatio-temporal dimension of $N \times t$, with $t$ being the total droughts.

In Step 3, KCA is applied to the $N \times t$ PC field to detect similar patterns among $t$ drought years and group them. KCA is an unsupervised classification technique that aims to cluster variables based on their similarities, minimizing the geometrical

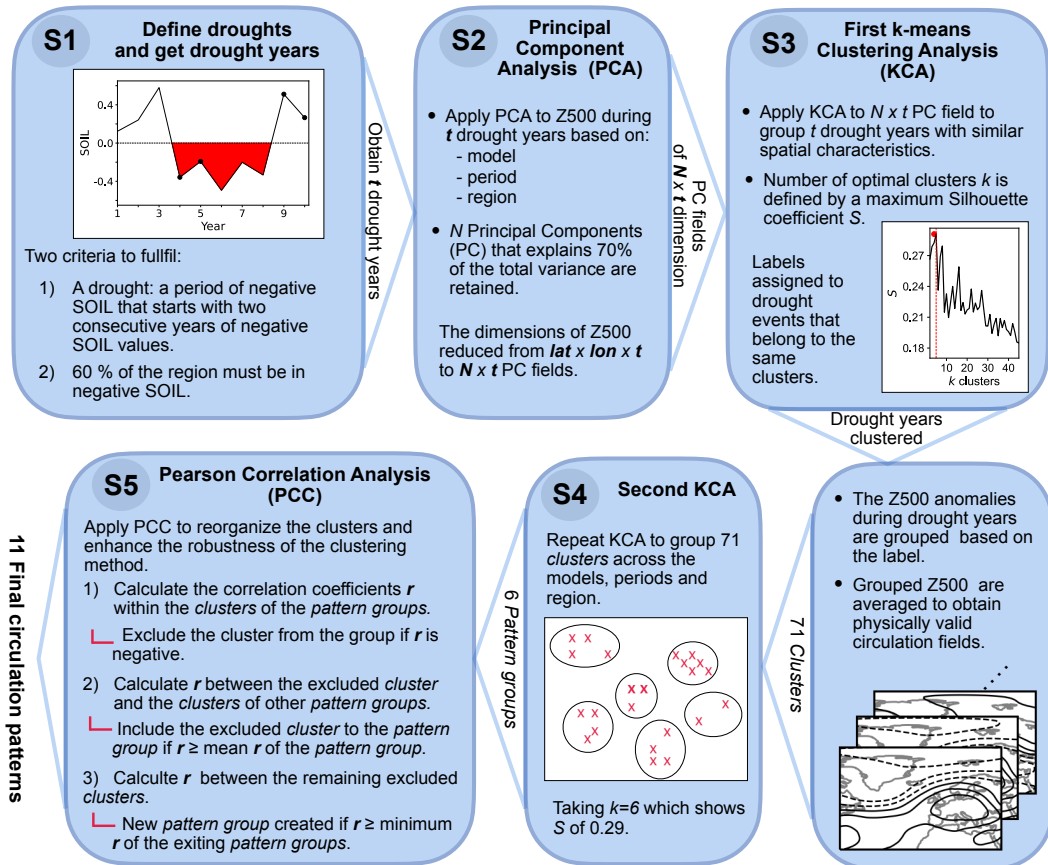

**Figure 1.** Flow chart summarizing the procedure for detecting the extratropical circulation patterns associated with the Mediterranean droughts. The diagram includes the following steps: S1) define droughts and get the drought years; S2) apply the principal component analysis to Z500; S3 and S4) apply the k-means clustering analysis to the PC fields and the temporarily clustered circulation patterns (*clusters*); S5) perform the Pearson correlation analysis to obtain the final circulation patterns. Note that the figures in S1, S3, and S4 are included for illustrative purposes. A detailed description of each step can be found in Sections 3.4 and 3.5.

distance between the data points $a_i$ (in our case, the PC fields) and between the potential clusters $c_k$ (Wilks, 2011; Zscheischler

et al., 2012). Hence, it attempts to minimize the clustering objective function $Q$ by

$$Q(c_1, ..., c_k) = \frac{1}{t} \sum_{i=1}^{t} min_{k=1,...,k} \|a_i - c_k\|^2 \tag{2}$$

where $k$ is the number of clusters, and $t$ is the total data points, which are the drought years in this study. KCA has the advantage of repeating the estimation in eq. 2 until the best set of $k$ number of clusters is found.

The numbers of PC $N$ and cluster $k$ are not initially defined. We calculate the mean Silhouette coefficients $S$ (Shahapure

and Nicholas, 2020) for a range of $N$ and $k$ to find the optimal $N$ and $k$ (Step 3 in Fig. 1). A mean Silhouette coefficient $S$ is a metric that measures the quality of clustering by considering the similarity within the same cluster and the dissimilarity between the clusters. To get the mean Silhouette coefficient $S$, a Silhouette coefficient $s$ for each data point is first calculated by

$$s = \frac{d_m - d_c}{max(d_c, d_m)} \tag{3}$$

where $d_c$ is the mean distance within the same cluster, and $d_m$ is the mean distance to the nearest neighbouring cluster. All the $s$ for each point are averaged to obtain $S$. $S$ ranges between -1 and 1, 1 indicating an optimum cluster with a high similarity among its elements and a high distance from the neighbouring clusters. $S$ are calculated across a range of $N$ (with at least 70% of explained variance) and $k$. Then, the $N$ and $k$ yielding the highest $S$ are selected. Once an optimal k is chosen, the PC-KCA method is not sensitive to changes in N. After this selection procedure, PC-KCA is applied to the $N \times t$ PC field

with a parameter $k$ for the number of clusters.

Through this PC-KCA process, drought years with similar spatial characteristics are identified and grouped, with labels assigned to the events that belong to the same cluster. Subsequently, the Z500 anomalies during droughts are grouped based on the label assigned by the PC-KCA process to retrieve circulation fields of physical significance. Then, the averages of these grouped Z500 anomalies are computed for each group to obtain clusters representing drought circulation patterns.

The PC-KCA procedure is conducted individually for each model — CESM1, GISS-E2-R, CCSM4, bcc-csm1-1, and MIROC-ESM —, period — LM or Hist —, and region — western or eastern Mediterranean. After Step 3, 3 to 7 clusters are obtained for each model, period, and region (Table 2), totaling 71 clusters. The $N$ number of PCs and their explained variance, the $k$ number of clusters, and the mean Silhouette coefficients $S$ obtained during the PC-KCA can be found in Table 2. The supplement to this paper includes the full range of Silhouette coefficients $S$ obtained for each model and period (LM and

Hist simulations in Fig. S1 and Fig. S2, respectively).

In Step 4, KCA is applied again to these 71 clusters (from now on, referred to as *cluster*) to group similar clusters across all models, periods, and regions. Initially, $k = 6$ is used since $S$ decreases abruptly after this $k$ (Step 4 in Fig. 1). These six clusters obtained by grouping cluster are referred to as the *pattern groups*.

In the final Step 5, Pearson correlation analysis (PCC) is performed on the clusters within the same and across different

pattern groups to create new pattern groups based on the Pearson correlation coefficients $r$. This step is to ensure the similarity between the clusters within the same pattern group. Initially, PCC is performed on the clusters within the six pattern groups

**Table 2.** Percentages of variance explained by $N$ number of principal components (PC), $k$ number of clusters, and mean Silhouette coefficients $S$ obtained by the PC-KCA steps of the pattern detection according to the model, Mediterranean region and period considered.

| | | Last Millennium (LM) | | | Historical (Hist) | | |
|---|---|---|---|---|---|---|---|
| **Model** | **Region** | **% Variance ($N$ PC)** | **$k$ Clusters** | **$S$ Coefficient** | **% Variance ($N$ PC)** | **$k$ Clusters** | **$S$ Coefficient** |
| CESM1 | West | 77.91 (5) | 3 | 0.17 | 79.82 (5) | 3 | 0.23 |
| | East | 78.60 (5) | 3 | 0.18 | 79.82 (5) | 3 | 0.20 |
| GISS-E2-R | West | 75.11 (5) | 3 | 0.19 | 75.78 (5) | 7 | 0.22 |
| | East | 73.63 (6) | 3 | 0.17 | 77.60 (5) | 3 | 0.22 |
| CCSM4 | West | 77.03 (5) | 4 | 0.18 | 80.20 (5) | 3 | 0.18 |
| | East | 74.43 (5) | 3 | 0.19 | 80.56 (5) | 3 | 0.24 |
| bcc-csm1-1 | West | 77.01 (5) | 3 | 0.19 | 83.93 (5) | 3 | 0.26 |
| | East | 75.34 (5) | 6 | 0.19 | 80.95 (5) | 3 | 0.23 |
| MIROC-ESM | West | 72.90 (5) | 5 | 0.18 | 76.05 (5) | 3 | 0.24 |
| | East | 75.13 (6) | 3 | 0.17 | 79.85 (5) | 4 | 0.19 |
| Total 71 clusters | | | | | | | |

obtained in the previous step. $r$ are calculated between all possible pairs of the clusters within a pattern group. Then, any cluster negatively correlated with one or more clusters of the same pattern group is excluded from the group. Next, $r$ is calculated between each excluded cluster and the clusters of the other five pattern groups. This is to determine whether the excluded cluster from one pattern group can join any other groups. The criterion for joining the other pattern group requires the excluded cluster to exhibit positive $r$ with all clusters of the target pattern group, and the $r$ must be more than the mean $r$ of the target group. This process is repeated for all excluded clusters from the six pattern groups.

If this reorganization of clusters is not successful for some excluded clusters, $r$ is calculated between those clusters that have not been assigned to any of the six pattern groups. If some remaining clusters are positively correlated with each other and have a mean $r$ higher than the minimum mean $r$ among the six pattern groups, these clusters are gathered to form a new pattern group. This re-assignment is repeated until no further grouping is possible.

Finally, this thorough process results in 11 definitive pattern groups that characterize the mean circulation patterns associated with drought conditions in the Mediterranean region during the last millennium and historical periods (850–2005 CE). The sequence of the PC-KCA-PCC technique involves repeating the clustering procedure three times. This procedure serves to enhance the robustness of the clustering method. The approach also allows us to trace clustered patterns throughout the entire procedure, exhibiting metrics ($S$ and $r$) in each step that measure how well the patterns are grouped.

# 4 Results

## 4.1 Observation-model comparison (1950–2005 CE)

The annual soil moisture anomalies (SOIL) from NOAH-LSM and the forced PMIP3-CESM1 simulations during 1950–2005 CE and 850–2005 CE over the Mediterranean region are presented in Fig. 2. SOIL during 850–2005 CE are standardized with respect to 850–1849 CE. During 1950–2005 CE (Fig. 2a), the variability of SOIL from NOAH-LSM (the range of maximum and minimum over 1950–2005 CE is 18.29 mm month$^{-1}$) is within the range of variability of the PMIP3-CESM1 model SOIL values (the range of maximum and minimum across the models of 22.75 mm month$^{-1}$). This observation indicates that the overall magnitudes of variability between the observation-based data and the models are comparable. However, the standard deviation ($\sigma$) of SOIL across the four models over the entire time (3.06 mm month$^{-1}$) is lesser than the $\sigma$ of SOIL of NOAH-LSM (4.37 mm month$^{-1}$). This distinct $\sigma$ indicates that there is some degree of discrepancies in SOIL variability among the models. GISS-E2-R, CESM1, and CCSM4 show $\sigma$ of SOIL of 7.73, 5.44, and 4.43 mm month$^{-1}$, respectively, which are higher than that of NOAH-LSM, while bcc-csm1-1 and MIROC-ESM presents $\sigma$ of 2.03, and 2.46 mm month$^{-1}$, respectively.

The ensemble means of standardized SOIL during 850–1849 CE (Fig. 2b) shows no apparent monotonic trend during LM. In general, the ensemble means of each model exhibits decreases in SOIL since 1850 CE. The decreases are noticeable even considering the ensemble spreads of each model, except for MIROC-ESM. In MIROC-ESM, a decline of SOIL is observed at the beginning of Hist, but then SOIL increases around 1950 CE. This trend of SOIL in MIROC-ESM is remarkably different from other models and may emphasize model-dependent response to forcing drivers, i.e., GHG forcing.

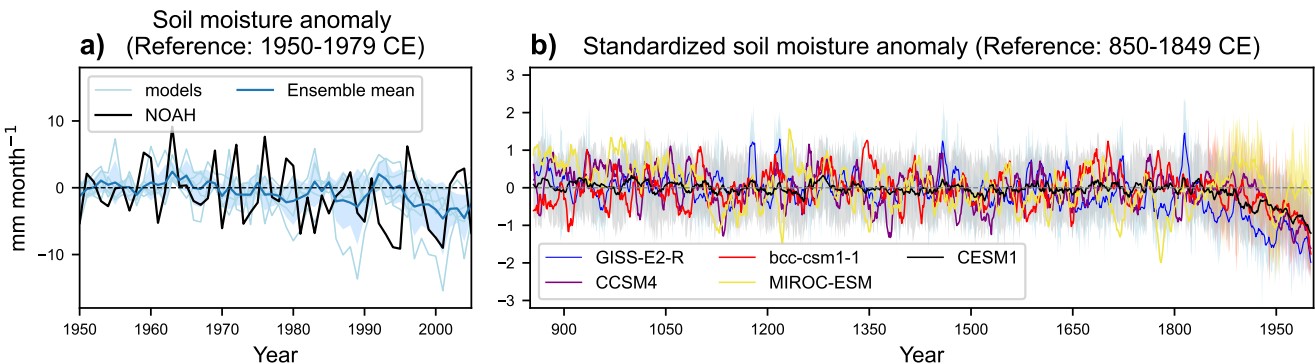

**Figure 2.** (a) The annual mean time series of SOIL over the Mediterranean region (13°W–37°E, 32°–43°N) for the period 1950–2005 CE with respect to 1950–1979 CE. SOIL from NOAH-LSM is in black and light blue for PMIP3-CESM1. Color shading indicates the ensemble spread denoted as one standard deviation ($\sigma$) across the five climate models in Table 1. (b) 10-year running means (thick lines) and ensemble spreads (color-shaded) of standardized SOIL during 850–2005 CE with respect to 850–1849 CE, from the PMIP3-CESM1 transient LM and Hist simulations. Note that the post-1850 trends in Hist are not detrended in these time series.

The spatial correlations between SOIL and Z500 of NOAH-LSM–ERA5 and climate simulations are presented in Fig. 3. In addition, the pattern correlations obtained between these spatial correlation fields of NOAH-LSM–ERA5 and each of the climate models are shown on the top of each panel. In all simulations, SOIL is negatively correlated with Z500 in the western (Fig. 3a) or eastern Mediterranean (Fig. 3b). This indicates that a decrease in SOIL is associated with a high Z500, a typical atmospheric circulation during dry conditions. A high-pressure system and a ridge over a region are associated with stable atmospheric conditions with a clear sky, low precipitation, and high temperature. The latter can also induce an initial increase in evapotranspiration that accelerates soil moisture loss. Additionally, the anticyclonic circulation hampers incoming moisture fluxes from the Atlantic. However, the spatial extent of these negative correlations centered on the focus regions varies across the models, and the signals of correlations outside of the regions differ noticeably between some models (e.g., CESM1 and bcc-csm1-1).

In the western region (Fig. 3a), NOAH-LSM–ERA5 is characterized by positive correlations at high latitudes over 60°N and negative correlations at the mid-latitudes, a pattern that resembles the NAO. CESM1 and CCSM4 show, to some extent, the most similar correlation patterns to NOAH-LSM. GISS-E2-R and MIROC-ESM exhibit similar positive correlations over the high latitudes, but the spatial extent and location differ from those in NOAH-LSM–ERA5. The bcc-csm1-1 does not present statistically significant correlations over 50°N.

In the eastern region (Fig. 3b), NOAH-LSM–ERA5 shows negative correlations over southern Europe, and positive correlations are located at high latitudes over the Scandinavian Peninsula. This pattern is similar to that for the western Mediterranean but with lower values and a slightly different spatial extent. CESM1 exhibits the most similar pattern to the NOAH-LSM–ERA5. The rest of the models also show negative correlations over southern Europe. For GISS-E2-R and MIROC-ESM, the correlations are mostly significant in the European domain but not outside the continent.

The numbers on the top of the panels indicate the pattern correlations between the correlation patterns of NOAH-LSM–ERA5 and each climate model presented in Fig. 3. The pattern correlation analysis is performed on the entire regions in Fig. 3, which means that these values measure the overall closeness of the entire correlation fields of the models to that of NOAH-LSM–ERA5 without considering the statistical significance of individual grid locations. We use this quantity to compare the model's representation of SOIL-related Z500 patterns in the western and eastern Mediterranean regions. Overall, the correlation coefficients are higher in the western region than in the eastern Mediterranean. In the western region, the maximum coefficient is 0.82, shown by bcc-csm1-1, while the maximum in the eastern region is 0.62, also by the same model. The minimum value in the western region is 0.42 by GISS-E2-R, and in the eastern region, it is 0.14 by CCSM4. The overall comparison of pattern correlation coefficients implies that the variability of Z500 associated with SOIL in the climate models is closer to that from NOAH-LSM–ERA5 over the western region than over the eastern region. A potential implication of the difference between the models and the regions is discussed again in the coming sections.

Nevertheless, all models present similarities to the NOAH-LSM–ERA5, fed with the present observational data, exhibiting a center of negative correlations in southern Europe with larger values over the focus regions. In general, CESM1 resembles the NOAH-LSM–ERA5 in both target regions better than the other models, although some difference exists in the spatial extent of significant correlations. The difference in correlation patterns between the models can be due to the model-dependent internal

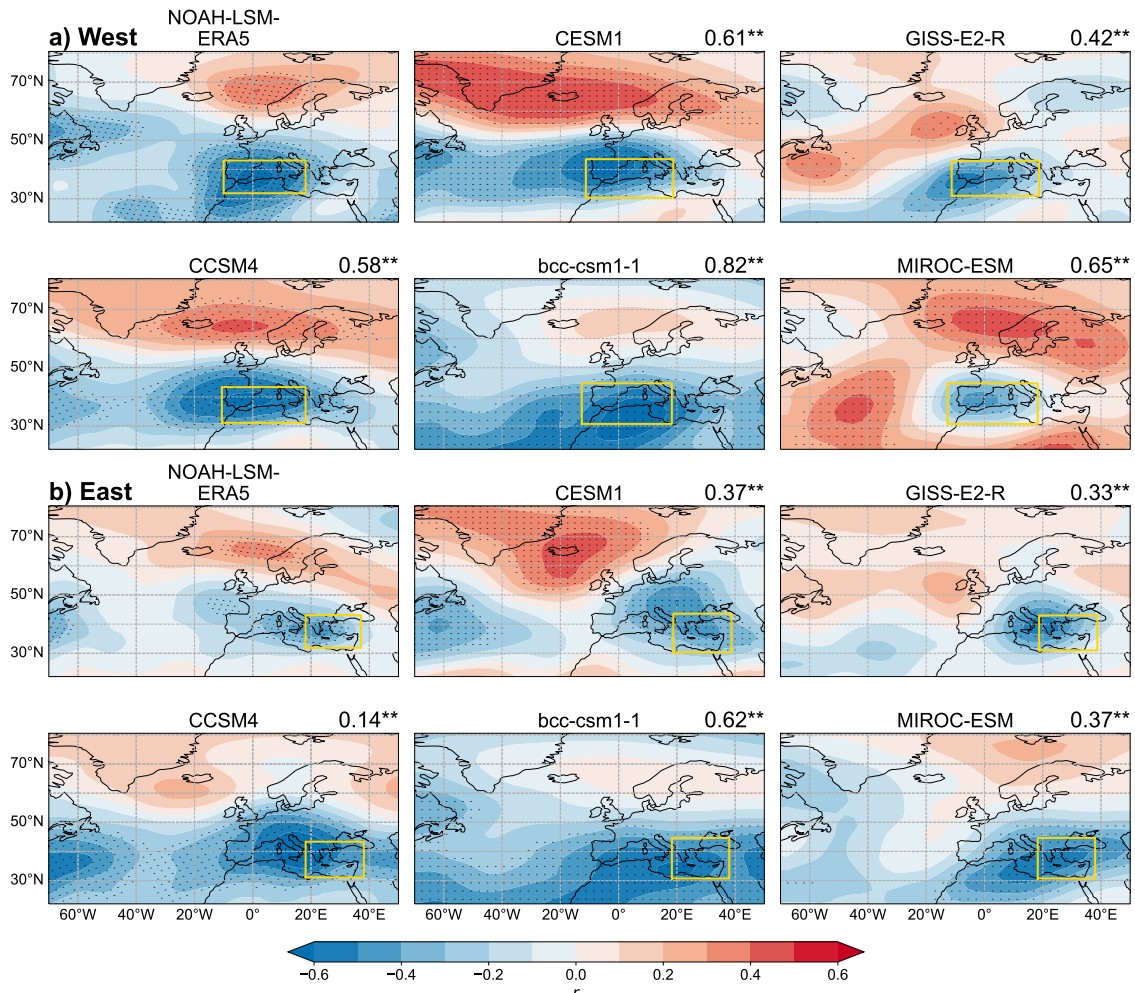

**Figure 3.** Pearson correlation coefficients ($r$) between the time series of SOIL for (a) the western or (b) eastern Mediterranean and Z500, during the period 1950–2005 CE. For the climate models — CESM1, GISS-E2-R, CCSM4, bcc-csm1-1, and MIROC-ESM — the first ensemble member is used for the calculation. The dotted regions highlight where the correlations are statistically significant at the 5% level. Yellow squares indicate (a) the western and (b) eastern Mediterranean regions where droughts are estimated, and the spatially-weighted soil anomalies are obtained. Note that the yellow squares delimiting each region differ slightly between the models since the horizontal resolutions are not interpolated to the same grid. The numbers on the top left of the panels indicate the pattern correlations between the correlation fields of NOAH-LSM–ERA5 and the corresponding climate model. The values are denoted with ** when they are statistically significant at 95% confidence level.

climate dynamics and a relatively shorter period (56 years) considered for the correlation analysis. A time series of 56 years may not include all possible variability of SOIL and Z500, including low-frequency variability on multi-decadal or longer

time scales present in the model simulations (Fig. 2b). This unaccounted factor could also influence the comparison between NOAH-LSM–ERA5 and climate models, hence, the significance level of the statistical tests.

## 4.2 Mediterranean drought characteristics during the last millennium (850–2005 CE)

Before examining droughts for the entire last millennium in the climate simulations, it is necessary to remark that the drought definition is sensitive to the reference period. For instance, when 850–1849 CE (LM) is considered as the reference period to calculate the soil moisture anomalies, the Mediterranean region is under a constant long-term dry condition during the entire historical period (figure not shown). This means that the mean climate was wetter in LM than in the following decades, a change that is mostly attributed to anthropogenic effects on global and regional climate (Douville et al., 2021; Seneviratne

et al., 2021). These continuous negative SOIL conditions are not apparent when the recent period (1950–1979) is considered for the anomaly calculation (Fig. 2a).

Therefore, to analyze the entire last millennium continuously without the influence of the recent anthropogenic forcing, each model's ensemble mean of SOIL is extracted from the corresponding model's ensemble members of Hist (Fig. 2c) as explained in Section 3.1. The time series of the number of drought years (per century) for 850–2005 CE are presented in Fig. 4. The

350 percentage of total drought years and the mean duration of droughts are included in the figure. In addition, Tables A2 and A3 show the values of mean percentage and duration of drought years for each period (LM and Hist) over western and eastern Mediterranean together with the standard deviations across the ensemble members of each model.

The number of total drought years and the mean duration vary across the models. Based on the ensemble means, MIROC-ESM shows a reduced percentage of total drought years in both regions (7.02% for the western region and 7.63% for the

355 eastern region) compared to the other models. The percentages of droughts in the other models range from 9.26% (GISS-E2-R) to 11.24% (CCSM4) for the west and from 8.77% (GISS-E2-R) to 11.44% (bcc-csm1-1) for the east. The mean duration varies from 4.07 years (CCSM4) to 4.62 years (bcc-csm1-1) for the west and 3.95 years (CESM1) to 4.71 years (MIROC-ESM) for the east. For the duration, unlike for the total drought years, MIROC-ESM shows comparable duration to other models and higher values than others in the east, indicating that in MIROC-ESM, there are fewer droughts but with longer duration.

The percentage of drought years and the mean duration of droughts in Fig. 4 for each climate model and period, including their respective standard deviations, are presented in the appendix Tables A2 and A3. The tables show that the percentage of drought years and the mean duration of droughts vary across the ensemble members. For the percentage of drought years, particularly CESM1, CCSM4, and MIROC-ESM exhibit larger standard deviations during Hist. In the case of the mean duration, bcc-csm1-1 and MIROC-ESM show larger standard deviations during Hist over the eastern Mediterranean region.

The time series of drought years (Fig. 4) show that no simultaneous period of increasing or decreasing drought events is observed across the models. For instance, the apparent decreases in drought events in 1600–1650 CE in GISS-E2-R in the west do not appear in any of the other models. Increased drought events in the west during mid-1500 CE are also only shown in CCSM4. The same is also observed across the ensemble members of the same model. The ensemble members of CESM1 and GISS-E2-R do not exhibit unanimous periods of low or high drought occurrence (figure not shown), which aligns with the

difference in the drought years and the duration of droughts across the ensemble members as presented in Tables A2 and A3.

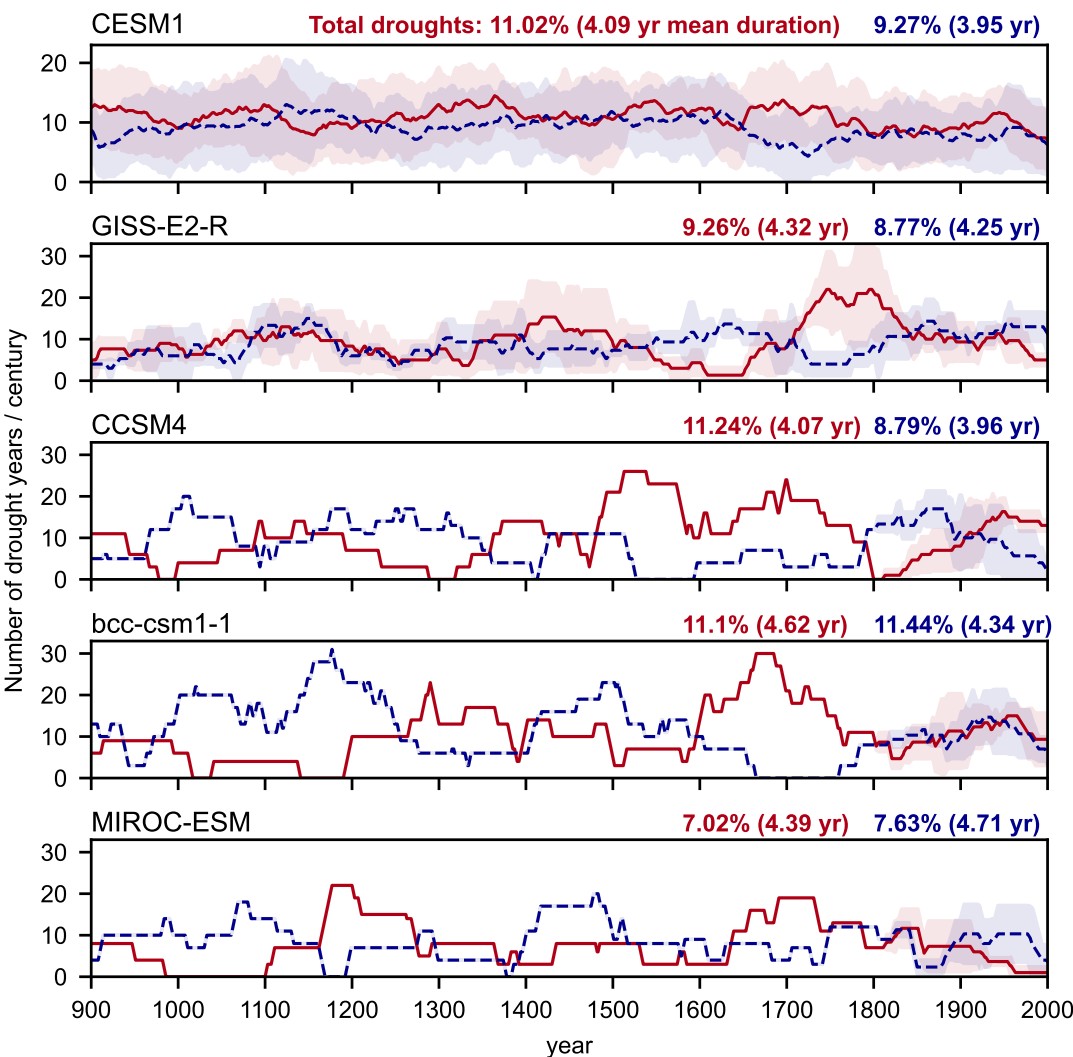

**Figure 4.** Number of drought years in a moving window of a century in the western (red) and eastern Mediterranean (blue). Thick lines for those models and periods that have more than one ensemble member correspond to the ensemble mean, and color-shading indicates the ensemble spread represented by a unit standard deviation. The mean percentages of total drought years and the mean duration of droughts, calculated from the ensemble means, are shown in the top-right part of each panel. The values for each model and period (LM and Hist), including the standard deviations, can be found in Tables A2 and A3. Note that the y-axis in CESM1 (first panel) is different from others.

The large variability in drought occurrence across the models and the ensemble members, depicted by the ensemble spreads in Figs. 4 and 2b, implies that internal climate dynamics is the primary driver of droughts in the region during LM and Hist with the anthropogenic forcing removed. In these time series, external or volcanic forcing signals are not visible in the variability of droughts. More precisely, although external forcing can affect drought occurrence, their influences are masked

by internal climate dynamics. This observation is in line with previous climate model-based studies (Xoplaki et al., 2018; Kim and Raible, 2021) and proxy-based studies (Cook et al., 2016; Rao et al., 2017). Volcanic eruptions that strongly affect the global hydroclimate on a multi-year time scale are known to be related to wetter conditions over the Mediterranean (Iles and Hegerl, 2014; Gao and Gao, 2017; Kim and Raible, 2021). Although other external forcings could be associated with a much longer fluctuation of dryness and wetness over the region (e.g., millennial to orbital scales; Stockhecke et al., 2016), these time scales are beyond the scope of this study.

In the number of drought years in Fig. 4, an anti-phase of drought occurrence with multi-decadal time scales is observed between the western and the eastern regions, more clearly in those models and periods with one ensemble member. This anti-phase occurrence seems to be in line with Roberts et al. (2012). At the same time, for some periods, simultaneous drought occurrence in both the western and eastern regions is detected, for instance, 1100-1150 CE in CCSM4 and 1400-1450 CE in bcc-csm1-1. This result seems to agree with Cook et al. (2016) that have shown the simultaneous occurrence of hydroclimate variability between the western and eastern Mediterranean on a multidecadal time scale.

To evaluate closely the dominant frequency of the association between the two regions and their temporal co-variability, the wavelet coherence analysis is performed on the time series of SOIL (Fig. 2b) and presented in Fig. 5a. At first glance, the wavelet coherence analysis suggests that timing and frequencies of co-variability are not the same across the models for both the soil moisture anomalies and the number of droughts. Also, the association is not uniform across the time-frequency space. The analysis performed on SOIL between the western and eastern regions (Fig. 5a) indicates co-variability that ranges from interannual to multi-decadal time scales, depending on the model. CCSM4 shows co-variability of higher frequencies (less than a 32-year period.). In general, the association between the two regions is in-phase. The co-variability between the western and eastern regions from all models is less pronounced and less significant in all time-frequency bands compared to the result presented by Cook et al. (2016) based on OWDA, which has shown significant in-phase co-variability of SOIL between east and west in diverse timescales. The analysis was also repeated for the summer (JJA) SOIL (figure not shown). The summer SOIL shows the same result as the annual variability, indicating no apparent uniform phase co-variability in the climate models.

The time series of SOIL does not necessarily indicate that droughts or dry periods are in phase. To compare the two time series, specifically during drought periods, the wavelet coherence analysis is performed on the number of drought years (Fig. 4), which is presented in Fig. 5b. The co-variability between the two regions is significant in some time scales, for instance, on a multi-decadal time scale (around 32 years and higher) with an anti-phase relationship in CESM1, bcc-csm1-1, and MIROC-ESM. For GISS-E2-R, CCSM4, and MIROC-ESM, the anti-phase association is also significant on a high-frequency band (of less than 32 years). This result seems to agree with the observation from Fig. 4. It is observed that the anti-phase co-variability also depends on the time period, for instance, CCSM4 during 1400–1500 CE, the co-variability is in-phase. Again, the overall result points out the associations that are not uniform across time and vary across the models. The occurrence of droughts and dry periods can be associated with dominant drought-driving circulation patterns of each region. More details on this are provided in the next sections.

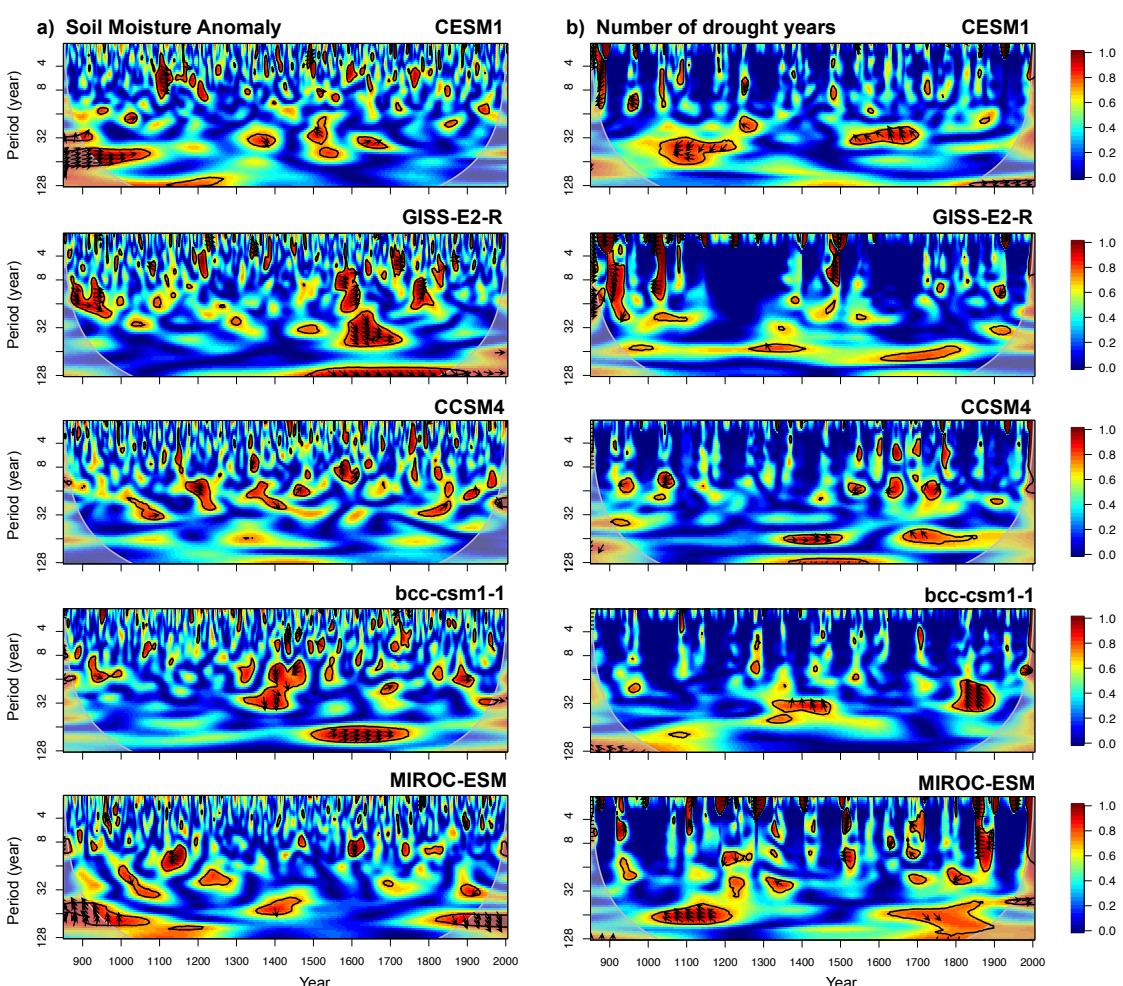

**Figure 5.** Wavelet coherence analysis between the western and eastern Mediterranean for the time series of (a) SOIL and (b) 100-year moving number of drought years, depicted in the time-frequency space. For those models and periods with more than one ensemble member, the analysis using only the first member is shown. For (a), SOIL in the west is obtained over a smaller area domain $10°$W–$0°$, $32°$–$42°$N, mostly encompassing the Iberian Peninsula and northern Morocco, to make it consistent with Cook et al. (2016). The red-shaded regions indicate where the co-variability between the western and eastern regions are statistically significant at 95% confidence level, estimated from Monte Carlo resampling of the time series. The arrows indicate the phase relationship: to the right (left) when the two time series are in-phase (anti-phase), and to the up (down) when the first (second) time series is the leading one.

## 4.3 Circulation patterns associated with droughts

The PC-KCA-PCC method (Section 3.5) is used to detect drought-related circulation patterns in Z500 of the climate sim-
410 ulations. In total, 11 drought-associated circulation patterns are detected. These patterns are presented in Fig. 6, with their frequencies (in the number of occurrences per century) and the mean composites of soil moisture anomalies corresponding

to each circulation pattern during 850–2005 CE. Each pattern group is a mean of a certain number of clusters. The first five patterns contain 83% of the entire drought years (Fig. 6). The models and periods that compose each pattern group are listed in Table A1.

Some pattern groups resemble the well-known modes of climate variability: P1 is the combination of a high-pressure system over Europe and a positive NAO pattern. This circulation pattern is similar to those observed in the correlation composite during the present day in the western Mediterranean (Fig. 3a), and it is commonly associated with Mediterranean droughts (Xoplaki et al., 2018; Kim and Raible, 2021). P2 is a negative NAO pattern with a center of negative anomalies in the mid-latitudes extending from the Atlantic Ocean to Europe. P3 is the opposite phase of P2, which also resembles a positive NAO. These three patterns enclose 62% of the total occurrence, highlighting the importance of the NAO for Mediterranean droughts. Positive NAO is known to be the dominant climate mode that drives a drier condition over the Mediterranean region (Lionello et al., 2006; Kim and Raible, 2021). This explains a large percentage of occurrence of P1 and P3 (42%) during droughts. Although it seems contradictory that P2 depicting a negative NAO condition occupies a significant percentage of the occurrence (20%), the occurrence of P2 reflects the fluctuation of NAO patterns throughout multi-year drought periods. Additionally, during P2, negative soil moisture anomalies associated with droughts are located predominantly in the southern Mediterranean region, indicating a higher occurrence of drought conditions in the south compared to the northern Mediterranean region. In contrast, central Europe experiences wetter conditions with negative Z500 anomalies.

P4 shows a similar high-pressure system over central and southern Europe but with negative Z500 anomalies in the Atlantic Ocean and high latitudes. P8 seems to be the opposite phase of P4. It is a wave-train pattern extending over the northern Atlantic and Europe, resembling an EA-WR pattern. P5 is similar to the Eastern Atlantic pattern.

Besides the patterns that are comparable to the well-known modes of variability in the mid-latitudes, some patterns exhibit more unique characteristics and are derived from one model only. The spatial structure of P6 with positive Z500 in the mid-latitudes and negative Z500 in the high latitudes is similar to a positive NAO in P3 but with a distinctly different Z500 anomaly over land. This pattern only appears in GISS-E2-R. P10, characterized by positive anomalies in the Atlantic and negative anomalies in central and northern Europe, and P9 and P11, with a high-pressure system centered over the eastern Mediterranean, are all derived from MIROC-ESM.

The frequencies of occurrences of the patterns (panels below each map in Fig. 6) indicate that droughts are associated with different circulation patterns. However, some patterns occur more frequently than others and are apparent only in some periods. The western and eastern Mediterranean do not always share the same patterns. P1 (high-pressure system and positive NAO-like) and P2 (negative NAO-like) appear in both regions with a similar frequency over time. P3 (positive NAO-like) and P4 (high-pressure system in central and southern Europe) occur more frequently in the western region, and P6 from GISS-E2-R is only associated with droughts over the western Mediterranean. P5 (EA-like) and P8 (low-pressure system in Europe) are more apparent in the east. P7 (EA-WR-like), P9 and P11 (high-pressure system in the east), and P8 are only associated with droughts over the eastern Mediterranean. Note that the last three pattern groups are from MIROC-ESM.

The frequencies of occurrences of the patterns presented in the lower panels of Fig. 6 indicate that the occurrence is not uniform over time, with certain patterns appearing only during specific periods. Nevertheless, the dominant frequency of each

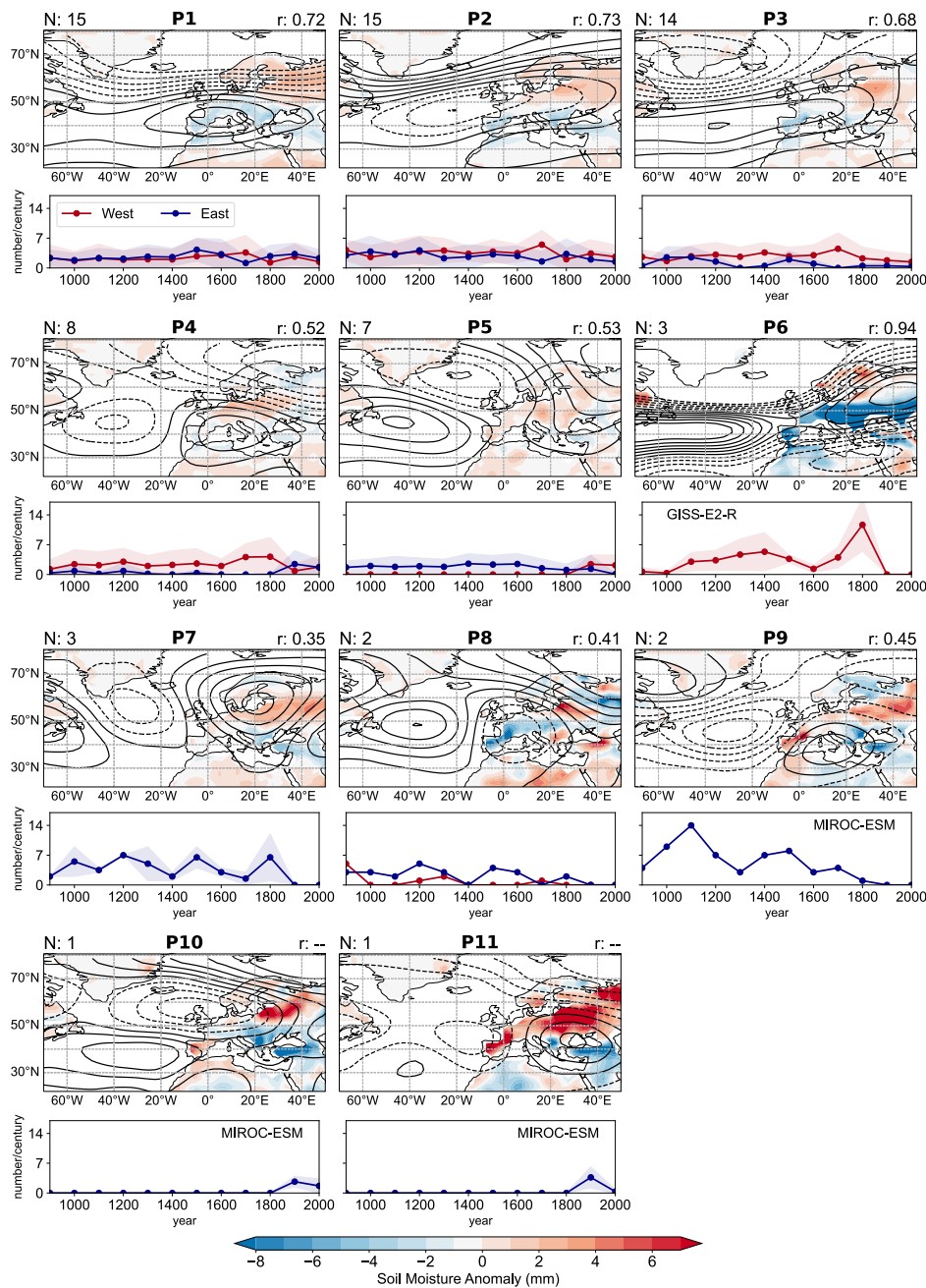

**Figure 6.** Z500 anomalies that represent the extratropical circulation patterns from $P1$ to $P11$ associated with Mediterranean droughts in black contours (every 0.3 geopotential meters), and the mean soil moisture anomalies corresponding to each circulation pattern in colors. The number of clusters $N$ averaged to make these mean circulation patterns are presented on the top-left part of each map. The mean Pearson correlation $r$ between the clusters within each circulation pattern appears on the top-right part. Below each map, the time series of the number of occurrences of the corresponding patterns per century for the western (red) and eastern (blue) Mediterranean are plotted, together with the respective ensemble spread of occurrence (shaded). When only one climate model belongs to a circulation pattern $P$, its name appears on the panel.

circulation pattern is estimated using a power spectra analysis (i.e., Cook et al., 2016), and the result reveals that the values noticeably vary across the models (Fig. A1). In general, the most dominant frequencies span a wide range of the multi-decadal (10–100 years) time scales, which partially seems to explain the time scales of the co-variability of SOIL and droughts during some periods in Fig. 5.

Fig. 7 summarizes the mean frequencies of the circulation patterns during the entire LM and Hist in Fig. 6, represented by the mean number of patterns per century for each model. The time series of the number of occurrences of the patterns per century can be found in the supplement Fig. S3. For simplicity, in Fig. 7, the patterns are classified by their similarity in spatial Z500 anomalies. For instance, those with high pressure over central Europe or the Atlantic Ocean and a positive NAO-like are shaded in red (positive-type pattern). The same is true for the patterns characterized by the low-pressure system and/or negative anomalies and a negative NAO-like over the same region but in blue (negative-type pattern).

Fig. 7 indicates that the positive- and negative-type patterns are shared by all models. As shown by Fig. 4, the frequencies of these patterns vary across the model and time period. In addition, some patterns are more noticeable in some models than others. The EA-WR-like pattern appears in CCSM4 and bcc-csm1-1 but not in CESM1, GISS-E2-R, and MIROC-ESM. Some patterns occur dominantly during one period. For example, P6, P7, and P9 are patterns that are visible during LM. However, P10 and P11 in the eastern and P5 over the western region appear only during Hist. In addition, the ensemble spread of pattern occurrence (black bars in Fig. 7) is also an important quantity to consider as it shows large variability in the occurrence of a pattern during the same period.

## 5  Discussion

It is arguable that the detected drought-associated circulation patterns and their frequencies depend on the numbers of $N$ PCs and $k$ clusters used for the method. We only take some PCs and clusters that explain an acceptable percentage of variance to increase the robustness of the clustering method. This approach may exclude certain Z500 anomalies that are outside the threshold. However, the percentage of variance in the same period is not distinctly different between the models (Table 2). They are between 72% to 78% in LM (around 80% in Hist) and with $k = 3$ in most of the models (maximum 6 or 7 for a few models). Hence, it is possible that the method may be sensitive to the selected parameters, and it may exclude some variability with a low occurrence, but it does not affect the comparability between the models.

Another finding is the difference in the detected patterns between LM and Hist. Some patterns appear only in one period and not in another. This difference in LM and Hist may occur because Z500 of each period is fed separately into the clustering method (Section 3.5), and the Hist simulations cover a much shorter period than LM. Some patterns may not occur frequently during this short period. Taking out an ensemble mean from the ensemble members may also influence the spatial patterns of Hist, as the values of ensemble means can be sensitive to the number of simulations (Maher et al., 2018). However, any statistical detrending method to exclude the recent anthropogenic forcing contains similar statistical drawbacks. For instance, the trend during Hist is not uniform over the entire period, which could cause some difficulties in selecting a suitable detrending period in a linear or polynomial detrending method.

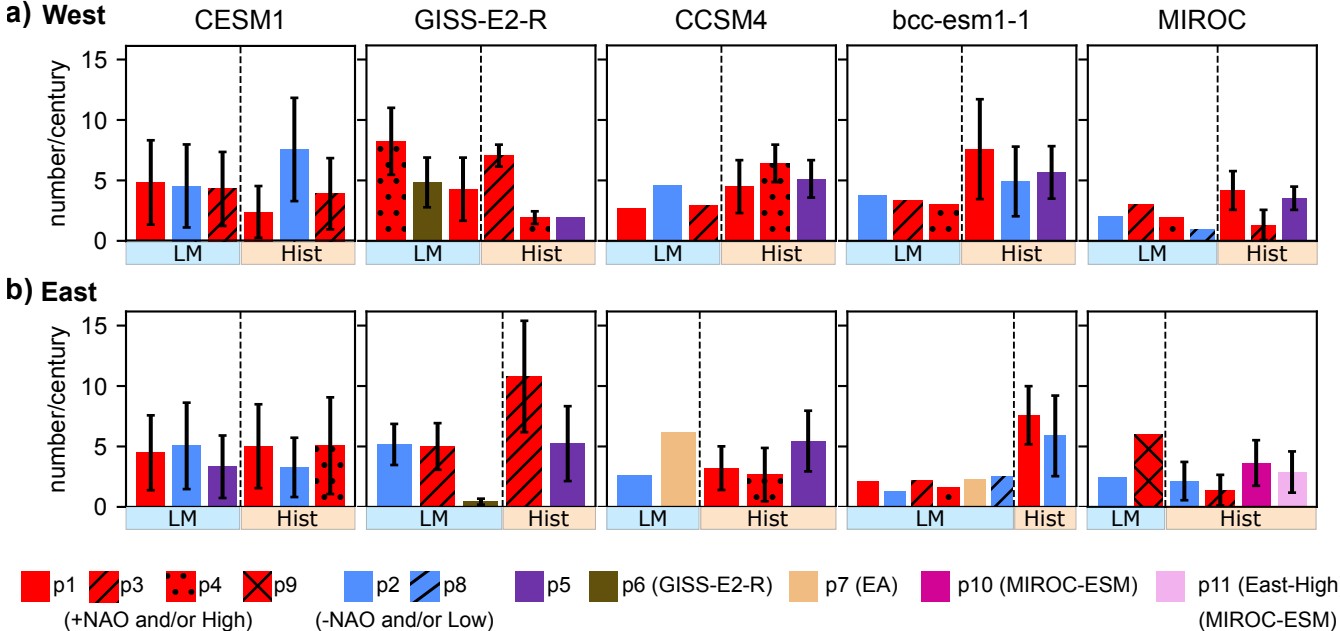

**Figure 7.** Mean occurrence of drought-associated pattern groups during the last millennium (LM) and historical (Hist) periods for climate model over western and eastern Mediterranean. Similar patterns in spatial Z500 anomalies are plotted colored with the same color but with different patterns. The black vertical lines indicate the spread of occurrence.

It needs to be pointed out that MIROC-ESM presents many unique patterns that fail to join other circulation patterns, largely occurring with the eastern Mediterranean patterns. Several reasons can be put forth to explain this particular characteristic of MIROC-ESM. As depicted in Fig. 2, MIROC-ESM presents a different SOIL trend compared to the other models during Hist. This distinct trend could potentially be attributed to different circulation types compared to those of other models.

Another argument to consider is that, in general, SOIL-related circulation fields in the eastern region seem to exhibit lower statistical similarity with the observation-based circulation condition, based on correlation coefficients in Fig. 3b. While this is the case for all models, lower correlation coefficients in the eastern regions can be associated with a reduced number of horizontal grid points compared to the western regions. Fewer grid points may not reflect the entire temporal variability of SOIL, and, therefore, the associated circulation variability.

Particularly, MIROC-ESM has only half of the grids compared to bcc-csm1-1 over both the western and eastern regions and presents relatively coarse vertical soil layers (Table 1). The coarse horizontal resolutions over the land and atmosphere may affect the variables associated with soil moisture, such as precipitation, which is sensitive to the horizontal grid size (Haren et al., 2015). In addition, relatively poor land vertical resolutions are probably insufficient to represent soil hydrology associated with vegetation and soil moisture memory effects that affect regional hydroclimate conditions (Hagemann and Stacke, 2015). This also can be the reason why MIROC-ESM shows spatial correlation patterns that differ distinctly from NOAH-LSM outside the focus region in Fig. 3.

Assessing which model represents soil moisture variability better, therefore also its related circulation, is a complicated task since it is known that the magnitudes of soil moisture depend largely on the internal physics of land surface models (Fang et al., 2016; Berg and Sheffield, 2018). Moreover, soil moisture interactions with the atmosphere (Berg and Sheffield, 2018) and vegetation dynamics related to soil processes (Huang et al., 2004) vary across the CMIP5 models. Choosing the same vertical soil moisture level (70 cm) across all the models to represent ecosystem-related depth, i.e., root zone, over the entire Mediterranean region may be another influential factor since the root zone varies with the region (Kleidon, 2004).

As here the objective is not to choose the best or worst model but to examine differences among the climate models, we use the same threshold for the soil moisture for all models. By setting the same criterion, we can better distinguish the differences among the PMIP3-CESM1 models in droughts and related circulation and also relate them to the models' vertical and horizontal spatial resolutions.

Regarding the pattern groups, in all models, NAO-like and high pressure and ridge over central Europe occur more frequently in the western Mediterranean droughts. The influence of positive NAO is weaker over the eastern region, presenting a lower frequency of occurrences. The eastern region is more dominated by eastern-type patterns, such as an EA-like pattern or eastern-centered high, where the center of positive Z500 anomalies is located over the eastern Mediterranean and weak or negative Z500 anomalies are found over the western region. These findings agree with already known studies on present observation-based dry periods (Xoplaki et al., 2004, 2012). The link between the NAO and EA patterns and Mediterranean droughts is also identified in a tree-ring-based reconstruction study by Cook et al. (2016) and Markonis et al. (2018). Different to the proxy-based studies, our study shows that the EA influence is largely concentrated in the eastern region.

The detected circulation patterns explain to some extent the anti-phase drought occurrence between the western and eastern Mediterranean in Fig. 4. In general, patterns involved in droughts in one region are characterized by strong positive Z500 in the focus region but by weak anomalies in another region. Such Z500 anomalies would bring dryness and warm conditions to the focus region and a relatively less dry condition to another region, agreeing with Dünkeloh and Jacobeit (2003) and Roberts et al. (2012). If these patterns are able to persist longer in time over a target region, that is, occurring more frequently during a certain period, then the other region may experience a long-lasting opposite condition. However, the mechanisms that maintain these interannual patterns to persist longer or occur more frequently, contributing to a multi-decadal scale anti-phase relationship, remain elusive. This result showing the anti-phase drought occurrence between the west and east seems to be in line with the lake sediment-based reconstruction by Roberts et al. (2012). However, in the models, the periods of the anti-phase co-variability of droughts periods, also in-phase co-variability of soil moisture anomalies in Fig. 5 are not uniform across the time periods, and the dominant time scales also vary across the models. This indicates a clear difference between the models and proxy-based records.

Our analysis also shows that the contribution of circulation patterns to droughts may greatly depend on the choice of the model. For example, the importance of EA-WR in the eastern droughts is apparent in CCSM4 and bcc-csm1-1 but not in other models (Fig. 7). The frequencies of a shared pattern between some models also vary greatly among them. This highlights the fact that climate models have their preferred circulation patterns associated with Mediterranean droughts.

## 6 Conclusions

We identify circulation patterns in the Euro-Atlantic domain associated with persistent droughts in the western and eastern Mediterranean regions during 850–2005 CE in several CMIP5-PMIP3 and CESM-LME climate simulations and perform comparisons across these climate models. Droughts are quantified through annual anomalies of vertically integrated 70 cm soil moisture and circulation patterns through the annual anomalies of geopotential height at 500 hPa.

Our findings emphasize that Mediterranean droughts are related to circulation patterns from internal climate dynamics, which is in line with previous studies (e.g., Cook et al., 2016; Xoplaki et al., 2018; Kim and Raible, 2021). Extratropical circulation patterns associated with Mediterranean droughts resemble the major climate patterns in the Euro-Atlantic region: western Mediterranean droughts are dominated by a high-pressure system over central Europe and an NAO-like pattern, while eastern Mediterranean droughts are linked to positive pressure anomalies in the southern and eastern Mediterranean, negative NAO, EA and EA-WR like patterns. For some periods, an anti-phase of drought periods between the west and east is found in this study, which agrees with Roberts et al. (2012) and Dubrovský et al. (2014). The circulation patterns and their frequencies seem to explain the anti-phase behavior between the western and the eastern Mediterranean. If one region experiences strong positive Z500, another region is characterized by weak anomalies during the same period. The mechanisms that drive the persistence of these patterns leading to a multi-decadal scale relationship remain elusive. However, the periods of this co-variability of soil moisture anomalies and droughts periods are not temporally synchronous across all the models and are significant only during some specific periods. This result of fewer significant periods and frequencies differs from the study by Cook et al. (2016) based on a proxy reconstruction during the growing season and indicates a clear difference between the models and proxy-based records.

Some circulation patterns associated with droughts occur more frequently than others, but not a single pattern dominates a certain region and period. Some patterns are only apparent in one model, indicating that the main drivers of droughts are different between the models. Moreover, large discrepancies in drought occurrence exist between the models and within the ensemble members of the models. This observation highlights model-dependent internal climate dynamics. Model differences in drought occurrence and patterns can also be attributed to the resolution of a model: it is noticed that coarse horizontal and vertical resolutions of land grid points might not reflect well the soil moisture variability and its associated circulation. For instance, MIROC-ESM shows distinct soil moisture variability and drought-related circulation patterns to other models. All these differences between the models can be a source of uncertainty that complicates model-proxy comparisons.

This work attempts to identify drought-associated extratropical circulations, focusing on natural climate variability in each individual climate model and comparing these drought-related characteristics across the models. In this way, differences between the models are identified better, including how they represent the baseline climate for droughts. Our results can also help understand model discrepancies and uncertainties in future drought projections, for instance, to examine which drought-associated modes of climate variability are preferred by each model and how these modes will change under different climate change conditions. A more detailed understanding of these differences may contribute to better future projections and, hence, aid long-term preparedness for droughts over the region.

*Code and data availability.* The scripts to reproduce the analysis and the figures presented in this manuscript are available on GitHub https://github.com/wmk21/Mediterranean_droughts_circulation. All the datasets used in this study are freely available online: CESM-LME at https://www.earthsystemgrid.org/, CMIP5 at https://esgf-node.llnl.gov/projects/cmip5/, ERA5 at https://cds.climate.copernicus.eu/, and NOAH-LSM soil moisture at https://disc.gsfc.nasa.gov/datasets/GLDAS_NOAH025_3H_2.1/summary.

*Author contributions.* WMK designed the study in discussion with CCR and set up the methodology with the input of SJGR. WMK conducted the analysis. WMK and SJGR prepared the first draft of the manuscript. All authors contributed to the scientific discussion and writing of the manuscript.

*Competing interests.* The authors declare that they have no conflict of interest.

*Acknowledgements.* We acknowledge the World Climate Research Programme's Working Group on Coupled Modelling, which is responsible for CMIP, and we thank the climate modeling groups (listed in Table 1 of this paper) for producing and making available their model output. For CMIP, the U.S. Department of Energy's Program for Climate Model Diagnosis and Intercomparison provides coordinating support and leads the development of software infrastructure in partnership with the Global Organization for Earth System Science Portals. We also acknowledge the Copernicus program for the ERA5 data (Hersbach et al., 2018) available in the Copernicus Climate Change Service Climate Data Store, and the NASA/NOAA Global Land Data Assimilation System for the Noah Land Surface Model dataset (Rodell et al., 2004). The authors also thank the editor Dr. Hugues Goosse, the reviewer Dr. Cecile Blanchet, and an anonymous reviewer for their comments, which have helped us to improve the original publication. WMK acknowledges funding from the Swiss National Science Foundation (SNF; grant number P500PN_206653). CCR is supported by the SNF (grant numbers 200020_172745 and 200020_200492) and the Swiss National Supercomputing Centre (CSCS).

## Appendix A: Appendix

**Table A1.** Models and periods pertaining to each pattern group after applying the PC-KCA-PCC method.

| | West | East |
|---|---|---|
| **p1** | CESM1-LM, CESM1-Hist, GISS-E2-R-Hist, CCSM4-LM, CCSM4-Hist, bcc-csm1-1-Hist, MIROC-ESM-Hist | CESM1-LM, CESM1-Hist, CCSM4-Hist, bcc-csm1-1-LM, bcc-csm1-1-Hist |
| **p2** | CESM1-LM, CESM1-Hist, CCSM4-LM, bcc-csm1-1-LM, bcc-csm1-1-Hist, MIROC-ESM-LM | CESM1-LM, CESM1-Hist, GISS-E2-R-LM, CCSM4-LM, bcc-csm1-1-LM, bcc-csm1-1-Hist, MIROC-ESM-LM, MIROC-ESM-Hist |
| **p3** | CESM1-LM, CESM1-Hist, GISS-E2-R-Hist, CCSM4-LM, bcc-csm1-1-LM, MIROC-ESM-LM, MIROC-ESM-Hist | GISS-E2-R-LM, GISS-E2-R-Hist, bcc-csm1-1-LM, GISS-E2-R-Hist, MIROC-ESM-Hist, |
| **p4** | GISS-E2-R-LM, GISS-E2-R-Hist, CCSM4-Hist, bcc-csm1-1-LM, MIROC-ESM-LM | CESM1-Hist, CCSM4-Hist, bcc-csm1-1-LM |
| **p5** | GISS-E2-R-Hist, bcc-csm1-1-Hist, CCSM4-Hist, MIROC-ESM-Hist | CESM1-LM, CCSM4-Hist, GISS-E2-R-Hist |
| **p6** | GISS-E2-R-LM | |
| **p7** | | CCSM4-LM, bcc-csm1-1-LM |
| **p8** | MIROC-ESM-LM | bcc-csm1-1-LM |
| **p9** | | MIROC-ESM-LM |
| **p10** | | MIROC-ESM-Hist |
| **p11** | | MIROC-ESM-Hist |

**Table A2.** Mean percentage (%) of drought years for each model and period (LM and Hist) over western and eastern Mediterranean. The standard deviation across the ensemble members of each model is also included.

| Western Mediterranean | | | | | |
|---|---|---|---|---|---|
| | **CESM1** | **GISS-E2-R** | **CCSM4** | **bcc-csm1-1** | **MIROC-ESM** |
| LM | $11.12 \pm 1.54$ | $9.27 \pm 2.45$ | $10.20$ | $10.10$ | $7.80$ |
| Hist | $10.83 \pm 4.34$ | $9.56 \pm 2.51$ | $14 \pm 3.57$ | $13.78 \pm 2.07$ | $5.56 \pm 3.71$ |
| Eastern Mediterranean | | | | | |
| | **CESM1** | **GISS-E2-R** | **CCSM4** | **bcc-csm1-1** | **MIROC-ESM** |
| LM | $9.29 \pm 1.37$ | $7.93 \pm 0.33$ | $8.80$ | $12$ | $8.40$ |
| Hist | $9.50 \pm 4.51$ | $14.67 \pm 2.37$ | $9.11 \pm 6.89$ | $10.67 \pm 4.71$ | $8.44 \pm 5.38$ |

**Table A3.** Same as table A2, but for mean duration of droughts (years).

| | CESM1 | GISS-E2-R | CCSM4 | bcc-csm1-1 | MIROC-ESM |
|---|---|---|---|---|---|
| **Western Mediterranean** | | | | | |
| LM | $4.21 \pm 0.27$ | $4.75 \pm 0.43$ | 4.63 | 4.81 | 5.20 |
| Hist | $4.02 \pm 0.43$ | $4.05 \pm 0.47$ | $3.52 \pm 0.12$ | $4.55 \pm 1.20$ | $3.33 \pm 0.47$ |
| **Eastern Mediterranean** | | | | | |
| | CESM1 | GISS-E2-R | CCSM4 | bcc-csm1-1 | MIROC-ESM |
| LM | $4.05 \pm 0.28$ | $4.14 \pm 0.46$ | 4.19 | 3.87 | 4.67 |
| Hist | $3.81 \pm 0.62$ | $4.43 \pm 0.49$ | $3.67 \pm 0.47$ | $4.74 \pm 1.68$ | $4.33 \pm 1.25$ |

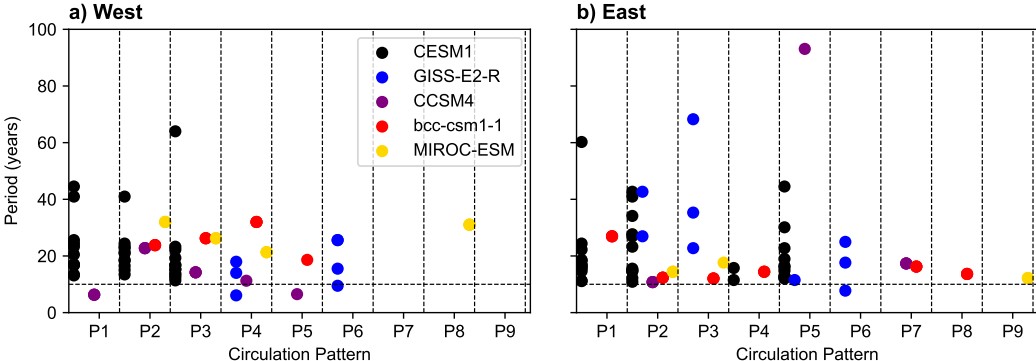

**Figure A1.** The most dominant frequencies (cycles/year) within the range of 0–100 year period from the power spectral density analysis (multitaper method with three tapers) applied to (a) western and (b) eastern mean SOIL during LM. Each dot indicates a value from one ensemble member.

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
