# Peer review of "Extratropical circulation associated with Mediterranean droughts during the Last Millennium in CMIP5 simulations"

_EGUsphere, 2023_

## Referee Comment (RC1)

Review of :
egusphere-2023-119
Extratropical circulation associated with Mediterranean droughts during the Last
Millennium in CMIP5 simulations
W. M. Kim, S. J. González-Rojí and C. Raible

Evaluation:
    The purpose of the manuscript valid and very interesting, and the
    methodological approach is reasonable for the aims of the study. The
    data used also represent a variety of sources that can bring confidence
    to the assessment. I think the study will eventually deserve publication. In
    its present state I would like the authors to consider a number of
    arguments that are listed below. Quite a few are methodological and
    needed for a proper understanding of the results. Also, they are needed
    to properly assess the correctness of the results. It is possible that the
    results will not change much after considering some of the suggested
    methodological issues but I think that at least the manuscript can
    improve from making them more clear.

GC1.  Abstract:
    I think the abstract is in general well written and it emphasizes the
    findings of the manuscript.

GC2.  Introduction:
    I think the introduction does its job in representing the frame of the
    problem and the case for this study.

    There are a couple of issues I would like the authors to consider though:

1.  Page 1, lines 21-23. 'The climate of the…'
    This paragraph divides the seasons into wet and dry and refers to the
    wet one specifically as the winter season. There is considerable spatial
    variability in the rainy seasons in the Mediterranean lands and for many,
    the wet seasons are spring and autumn. I think this winter/summer
    separation may be misleading. Consider for instance Xoplaki et al (2004;
    DOI 10.1007/s00382-004-0422-0) or others, where an extended wet
    season is used. I think this discussion should incorporate better this
    seasonality character.
    Note page 2, line 24 where you mention 'highly variable seasonal
    hydroclimate...', although the end of this paragraph (line 30) will make
    the emphasis again on winter.

2.  Page 3, lines 64-65. 'They also found that … is model-dependent'.
    This seems to relate specifically to the results of the current manuscript
    and perhaps should be discussed specifically either here, providing more
    specific background and describing in which way this can happen, or/and

at the end of the manuscript discussing what this manuscript provides in the context of previous evidence.

GC3. Section 2: Data
The data section can perhaps be improved by discussing a bit more the model sources and the rationale for decisions concerning soil moisture data, as indicated in the following.

1. Section 2.1 Page 3, lines 83-86.
The CMIP and PMIP and other model efforts could include a reference. I am aware they were just provided in the previous paragraph, but it seems pertinent to me that they get those also in the Data section. I leave this to the taste of the authors.

2. Section 2.1 Page 3, lines 85+ 'We consider only simulations…'
There are some issues that can be considered related to the use of soil moisture in different models:

- The models have very different depths and therefore they will possibly produce different soil moisture statistics, even if depths only down to 0.7 m are considered. Using this set up of models is fair and possibly it contributes to inter-model differences, but also it is arguable that models with shallower depths may be less realistic. Note that some models like GISS only have a depth of 3m. This will produce potentially a different vertical distribution of moisture and different temporal variability. Perhaps it is something the authors would like to comment on.

- In Table 1 it is included that a depth of 0.7 m is considered. I think this is relevant and should also be included in the text. It would be desirable to include a rationale for this decision and how it may influence the timescales of variability of soil moisture.
What is the typical range of depths for the bedrock, that limits the presence of water, for the area of interest in general in these models? Including a shallow limit may be more representative for the whole area but also exclude lower frequency variability more typical of deeper levels. I think including a rationale for this in the text of Section 2.1 would be good.

3. Section 2.2
Why are NOAH-LSM data used for soil moisture and ERA5 only for circulation? I mean what does NOAH-LSM offer that you would not get in this manuscript from ERA5 soil moisture?. There is probably good reasons for it. I would just suggest that a motivation for the use of these data, beyond the fact that soil moisture observations are scarce, is provided.

Also:

- Line 87 '… analysis: Giss-E2-R, CCSM4…'
References should be provided for these simulations. Considering previous points, I would suggest that a minimal background is at least provided in how different the soil moisture modelling can be among all

models. This would help giving the reader an idea of what to expect in terms of inter-model differences. For instance, vertical resolution can be very influential for soil moisture, as indicated in line 90…

- Line 87 '… 12 ensemble members of CESM1'
  There are 13 available if I am not wrong. Nothing critical but maybe you want to state why that selection.
  Here it can also be stated this refers to the all forcing simulations, although it is quite clear in the context (see next).

- Line 91 ' All simulations were run with the volcanic, solar…'
  It is clearly stated earlier in the introduction that this is all about internal variability. Perhaps some comment about focussing on forced simulations and not in long control runs (also available in cmip) is pertinent.

- Line 101 ' …has four layers up (down) to two meters'
  Ok, but I understand that only the soil moisture of the first 0.7 m is used, right? Explaining this in relation to the previous information in Section 2.1 would be clarifying.

GC4. 3. Methods
3.1     Calculation of anomalies.

I find the definition of annual anomalies and the explanations for it somewhat confusing. There is an emphasis in this explanation that I do not really understand. Perhaps this is my fault but, please, consider it and see if this influences the text as it stands now.

1. Section 3.1 Page 5, lines 111+.
   - 'These variables are transformed into the annual mean anomalies at each grid point'
     At this point I would say it is not possible to understand what is meant if additional information is not provided first, and I would argue to simply write '…transformed into annual anomalies'. I think that will be enough also for the rest of it.
   - 'We use annual mean anomalies in order to include winter conditions in the analysis, as it is an important season for the annual hydroclimate in the Mediterranean'
     I think it is important to discuss this in the context of what is indicated in GC2.1
     Specifically for this sentence, it is not possible at this stage to understand what is meant. Not a big thing because one can perhaps understand it from the following, but:
     a) I would suggest to include the description for the intra-annual anomalies before and then explain they are averaged; if these arguments stay in the text.
     b) I would actually (respectfully) challenge that this is true. I think it is not, but maybe I understood things wrong. Se below.

- 'Prior to the anomaly calculation … to the 70 cm level'
  These two sentences are relevant and the reader could go with knowing this earlier in the data section (GC3.2).
- 'Then, first, the monthly anomalies… monthly values' […] 'Second, the necessary annual mean … these monthly anomalies'
  I do not understand these two sentences. If you calculate monthly anomalies (differences between monthly data and the long-term monthly mean you are essentially subtracting the annual cycle. If you average out the resulting monthly anomalies what you get is the annual anomaly, equal to calculating anomalies from annual data by subtracting the long term mean.
  Therefore, I do not understand the previous sentence about the importance of winter or the unnecessary emphasis on the monthly anomalies if you end up in annual anomalies.
  The sentences in the middle about reference periods read fine to me. Perhaps it fits better with the next paragraph and the issue of the trends since their definition is somewhat related.
- 'Thereby, the ensemble means of the anomalies… Maher et al (2018)'
  With the preceding sentences I am confused. I understand the purpose is subtracting the ensemble average to each member to get rid of the long-term trends; or better said of the forced response in general as it will also affect volcanic events for instance. This would be consistent with the following arguments in the paragraph. Consider explaining/rephrasing these paragraphs.
- 'This method guarantees that only internally driven variability remains in the time series of the variables'.
  … mostly internally driven variability, but only in the post 1850 period. This should be perhaps indicated specifically.
  However, what I miss a bit is the rationale of why it is done like this. Why is the forced signal intended to be filtered out in the post1850 (natural and anthropogenic) and not in the pre-1850 period? The reader has to guess that perhaps this is intended to get rid of the long-term trends, and the rest of the forced signal goes with it

- Note also that the ensemble is calculated over different models in CMIP5-PMIP3 if I understood well (?). Now this is an issue that can be relevant, as different models will show different levels of response to external forcing. When this is done over an ensemble with the same model you can reduce the external response in each model run as it is assumed that the external response is common to all runs. One aspect here is that depending on the number of runs you are also subtracting also internal variability that has not been averaged out in the ensemble average, and this is a limitation that can be discussed. A more relevant aspect can be that if you are using different models in the ensemble as it is the case, the assumption that a common response to external forcing is filtered out does not hold and by subtracting from each model the ensemble average you do not actually filter out the external response of that specific model.

Therefore, this would work for the CESM-LME ensemble but not if you mix in the ensemble different models. This can be a sensitive issue that should be at least clearly discussed here or indicated here and better discussed later in the text.

- The two periods, LM and Hist, … models and regions'
  The pre-1850 and post-1850 are merged. However, they do not have the same nature, as in the pre-1850 the natural externally forced variability exists and in the post-1850 it is not intended to exist. Although if calculations have been done for some models out of a multi-model ensemble, some leftovers of externally forced variance should remain. Nevertheless, this should be in principle not important if externally forced signals are not pursued and are not expected to play a role either; actually, for that purpose, control runs could be as adequate or more adequate.
  I am trying to highlight with the previous sentences that some level of confusion can be transmitted to the readers.

I think the last paragraph is a good example that this section could benefit from some rethinking. I would say that not just in the details of the anomalies, but it is also important to provide a rationale for the reader to understand the strategy of the study… its advantages, and its limitations.

2. Section 3.3 Page 6.
   I wonder why not considering the soil moisture from ERA5 or ERA5-land additionally or instead to NOAH-LSM. This relates to GC3.3. Since ERA5 produces soil moisture out of its assimilation system, this would be physically consistent with other ERA5 variables used here. Perhaps the authors want to make some comments in the text about the advantage of using the NOAH data here instead.

3. Section 3.4 Page 6, lines 161+
- 'When these temporal and spatial …'
  The regional arguments about drought incorporate the condition of having 60% of the horizontal gridpoints with negative SOIL. This faces some difficulties with the fact of using different horizontal resolutions. However, this could be overcome by considering the spatial size of anomalies by using grid box area and the total spatial extension that the anomalies represent, instead of the number of gridpoints.

- 'Next, the weighted spatial average of SOIL is … Mediterranean'
  How is this weighting done? Perhaps I missed it earlier in the text?
  If this is done, why not using this regional average to determine the occurrence of spatially large enough droughts?. I do not think the result will be very different from the previous approach and it would overcome the use of a percentage of gridpoints with different model resolutions.

4. Section 3.4 Page 7, lines 171+
   'Only climate models with some ensemble members … not detectable'
   This is already a report of results. I have no major concern with
   anticipating it, but what is the reason for it?. The sentences are
   descriptive of the fact that pan Mediterranean droughts happen in some
   simulations with an specific feature and not in others. Can you at this
   stage argue about this feature?, or should this moved further down in the
   text and a discussion provided?

5. Section 3.5 Pages 7-9
   I get the structure of the methodology in general, however I suggest the
   authors revise this section for a more clear explanation, revising notation
   and perhaps the current state of the explanation of details of the methods
   that can be more important for the understanding of this section. I will not
   go to details in an exhaustive way, just provide some examples.
   The text is providing the sequence of a method. How the outcome of
   each of the steps feeds the following steps should be clear in explanation
   and, I would suggest, a homogeneous notation. I also recommend there
   is a rationale/justification for the conceptual use of each step. I will briefly
   try to highlight this with examples on the following, but please, go beyond
   those.

   References: the authors provide reference but please take care they are
   appropriate. For instance, there are many ways of applying PCA and the
   reference to the correct texts that describe the approach presented
   herein should be provided (see below).

   Use of maps and series: This should be clear from the text. For instance,
   in the PCA, I understand $T(t)$ are the principal components. Please,
   indicate that and also the range of the parameter $t$. The same with $s$, it is
   good to indicate the range so that the number of modes or the retained
   number of modes is well defined

   In equation (1), where are the eigenvalues?. I understand they are
   multiplying either the spatial or the temporal component. They should be
   indicated. I understand it is particularly relevant if they multiply the
   temporal component. The reason for it is that the temporal component
   will have a standard deviation 1 or different from it and this will impact the
   Kmeans procedure as it will affect the distance.

   Explained variances are mentioned but it is not said that the 70% reflect
   accumulated variance accounted for by a number of pcs.

   Some sentences are confusing. E.g. line 191 'PCA is applied to the Z500
   fields during droughts for each model'. I do not understand this.

   How do the PCs enter the following analysis? They have unit standard
   deviation (perhaps not), and their inter-pc correlation is 0. How does this
   play a role in the Kmeans clustering, what does it mean physically

because pcs should indicate different modes in time that are afterwards grouped, even if they are uncorrelated.

The notation of how the $T(t)$ go into equation (2) should be consistent with this equation. I suggest that the notation is blended for the various steps of the analysis. If it is not done, readers will have to accommodate how things fit from their knowledge and from the different steps of the method. If notation is blended, this section would actually describe one thing, the approach followed in the manuscript, not several independent methods. Some features of notation are repeated for different things, e.g., s as parameter for points in space and the Silhouette coefficient for each point.

There are 71 clusters. I learned this from the text but realized that this is the sum of all clusters from different models in Table 2. For each model 5 to 6 pcs are retained and from this, between (mostly) 3 and 6 clusters are formed. What does this mean? Are the 3 clusters gathering the information of the 5-6 pcs? In which way? Some rationale/explanation for what is conceptually happening is good for the reader.
Line 220: at this level I do not know what the correlation between clusters means.

All in all, section 3 needs, in my opinion, to be well revised do deliver a more clear and consistent text. This does not necessarily imply changes in the calculations, nor the results of the following sections. But it may impact (positively) the interpretation.

GC5. 4. Results
   4.1    Observation-model comparison

1. Related to previous comments:

   Line 237: '… by subtracting from each of the ensemble members the anomalies at each grid point'
   This relates to previous comments and could be explained better.

   Line238: 'The spatial correlations between SOIL and Z500 of NOAH-LSM and each of the climate simulations are presented in Fig. 3'
   I do not think it is wrong at all but I wonder what is the gain of using NOAH instead of the soil moisture from ERA5 in this figure.

2. Figure 2 and related
   2b and 2c are good in showing the impact of using different references. However I would say that the two are not really needed. With one of them it would be enough to explain it. Perhaps the space could be saved to accommodate a time series of the full 850-2005 period. I leave it for the authors to decide.

The shading in Fig 2b seems to indicate consistency in the range of variability with the range of 'observed' NOAH-LSM variability, which is good to indicate.

3. Figure 3 and related

- Watch some statements like ' … negative correlations over southern Europe, but the correlations outside Europe are not significant'
They are for CCSM4 and bcc in the western low latitudes of the domain.

   Line 255, '… all models present similarities to the NOAH-LSM, fed…'
What we see in Fig. 3 is the result of both NOAH-LSM and ERA5, right?. Actually, it is likely that the large scale structure we see there is more dependent on the global model; one could actually test if it changes with other reanalysis products. I think this is likely out of the scope of the study, but it is not totally off the line of argumentation because this correlation field with ERA5 is what we consider 'truth', but it could change if we would have used a different reanalysis product.

- Perhaps a more relevant issue: if you consider the variability among patterns in Figure 3, what is the variability among simulations of one single model ensemble?. I would assume that it is smaller, but it may be worth reporting.
In the actual figure 3, for those models with ensembles, is the pattern that is shown the result of one single experiment? If so, I think it should be clearly stated in the caption and in the text. I would advise against including correlations using ensemble averages, but I don't think this is what is being shown.

   4.2      Mediterranean drought …

1. Figure 4 and related
In relationship to the role of external forcing, or the lack of it ('This fact emphasizes that external forcing signals do not play a role in droughts over the Mediterranean …' line 286), there are several issues that may be worth commenting.

- How are droughts calculated over sub-ensembles of simulations (GISS, CESM1) in Fig. 4, and how is the ensemble spread provided for them? This should be explained (sorry if I missed it) in the text and figure caption. Also, I expect it will justify the different temporal resolution of the curves in Fig 4 for GISS and CESM1 in comparison to the others. However, it can be misleading as the results of those two models may be read as if soil moisture for CESM1 would be consistently higher (smaller) in the western (eastern) Mediterranean during the late 17th and 18th century, or in GISS also during the 18th and early 19th centuries… or in GISS opposite to that during the 17th century. This would not be possible and would contradict the first statement of no role in external forcing as it is very unlikely that different model runs of a sub-ensemble with different initial conditions will coincide systematically in simulating relatively dry or

wet periods unless forcing would play a major role. The only reason for that would be external forcing.

Also, the finding of opposite phasing between west and east is interesting, and I would argue that it should be more perceptible for GISS and CESM1 if individual runs are considered, in relation to the statement '…observed, more clearly in those models and periods with one ensemble member' (line 296). Therefore, I suggest the authors revise how the ensemble behavior is presented for those two models.

For instance, the last statement of Section 4.2 is sensitive: 'For those models and periods with more ensemble members… sometimes this association is blurred…', it should indeed be the expected behavior, even more than what is shown. If you resort to individual simulations, this should be more clearly evidenced. The ensemble spread should expectedly blur everything since a dry or wet century in one run should not be expected to be consistently dry or wet in most of the other sub-ensemble runs.

2. Figure 5 and related

I am not against showing temperature anomalies in association to the Z500 patterns, but why not showing precipitation and actual drought patterns. How do the geopotential anomalies account for drought occurrence?

Figure 6: I need a better description of the methods section to better interpret results and figure out whether they can be dependent on methodological choices.

Minor comments:

MC1.   Introduction, page 1 line 19
'  Ocean, teleconnections and large-scale modes of variabity.'
 Ocean, and teleconnections with other large-scale modes of variability (?)

MC2.   Page 2, line 40
'…change in climate boundary conditions…'
'… change in external forcing…' seems to me better in this context. The current sentence is maybe a bias from climate modelling

MC3.   Page 2, line 42
'Several natural proxy-based …'
Natural meaning?

MC4.   Page 2, line 44
'…summer dry and wetness…'
Dryness?

MC5.   Page 2, line 42
'…of droughts variability…' → of drought variability

MC6.    Page 5, line 111
        '…at vertical soil layers…'
        You mean perhaps, vertically integrated soil moisture content?, or
        something of the sort…

MC7.    Page 7, line 172
        '…show few numbers of…'
        Cases of?

---

## Referee Comment (RC2)

Review of Kim et al., 2023 submitted to Climate of the Past
By Cécile Blanchet

Disclaimer: not being a modeller, I will not be able to judge the technicalities of the study. I will therefore provide general comments on Mediterranean climates, the comparison to proxy data and (hopefully) help to improve the readability and accessibility of the manuscript for non-specialist audiences.

Scope and relevance: The manuscript shows modelling results from the CMIP5-PMIP3 ensemble to elucidate the main climatic regimes associated to droughts in the Mediterranean during the last millennium. This is an important topic that fully deserves our attention. I find the paper well-written and very clear, with scope and relevance suited for CP. However, the goals of the paper are at times not very clear with regard to knowledge gaps (see below) and some clarifications need to be provided about the methodology and the evaluation of the models selected to conduct the analysis. I therefore request a minor to moderate revision before the paper can be accepted and hope that my comments will be helpful.

Main comments

Before I list specific points in the manuscript, I wanted to raise the issue to the authors (it is not a must, more a proposition): I personally found Fig. 2 and 3 very interesting but under-utilised (and to some extend Fig. 6 too). Clearly, some models are not very skilful at capturing the synoptic climate during historical droughts and this raises in itself an important issue: how can we confidently understand drivers using longer simulations? What setup does "work better" to capture what part of the signal (temporal vs. spatial)? I know that expending on this aspect would modify the paper substantially but it is of crucial importance to build on. It would also give more confidence to the method applied later on to determine the dominant climatic modes (I must say, as is, I am wondering how much trust we can have – which might also be due to the fact that I am not savvy with the methodology used here).

Another (more minor) issue: Perhaps I am not familiar with the term, but it would be useful to clarify what you mean by "internal variability" (and that might be done by just explaining what is considered an external forcing in your study). If not used to the terminology, one might wonder for instance, whether teleconnections are part of the internal variability or not?

-   Abstract: lines 4-5: "The focus is […] during 850-2005, this excluding the anthropogenic trends from 1850 CE onwards". This is very confusing… It can only be understood if the methodology has been explained, otherwise one starts to wonder which time range was actually used.
-   Introduction: Lines 66-71, the definition of the knowledge gap and the leading questions is a little weak. "necessary to understand which modes of climate variability or atmospheric circulation patterns are involved in each climate model": this sounds very ambitious and is actually not what this study is tackling. Instead, it might be useful to explain why studying the "patterns in the mid-latitudes" and their effect on Mediterranean climates is important.
-   2. Data: $2.1: "All simulations were run with the volcanic, solar, and greenhouse has (GHG) focing
-   §3.1: this section of the methods is at times quite confusing. I have read the comments of Rev. 1 and agree with their comments so I will just emphasize two points that I found particularly confusing:
    o   Line 112: "We use annual mean anomalies in order to include winter conditions in the analyses […]": I don't understand this sentence.

- o Lines 121-126: This part is at the same time important and confusing: why not normalising the whole record instead of just the Hist part? Would a z-scoring be useful?
- §3.5: section very hard to follow, please provide additional information and background on the choices of the methods and their limitation.
- Results
  - o Fig. 2: if I understand right, this is your control run for time series. a) the modelled soil data are quite different from the observation time series (range and trend are similar but changes are often not synchronous). How does that affect your analysis? What does the intermodal comparison look like? Could you have a look at the cross-correlation between NOAH LSM and CMIP5 mean? Just so that we understand the limitations of the study.
  - o Line 238: "SOIL and Z500 from NOAH-LSM" isn't Z500 from ERA?
  - o Fig. 3: control run for spatial analyses. I think that it would be good to indicate that the upper left panel shows the observation (SOIL vs. Z500). How do you deal with models not capturing the synoptic climate very well (are they still used for further analyses, and if so, why?)? Can you quantify the skilfulness of the models? To me this exercise seems to be providing an evaluation of the models to track spatiotemporal climate conditions associated to droughts. It could be a goal in itself (as proposed earlier).
  - o Lines 286-288: I struggle with these sentences. Either too little or too much is said here. What is the role of models setup and skilfulness in this observation? I also do not understand what is meant by "counterfactual"?
  - o Fig. 4: Would it be possible to quantify the antiphase? (bi-plot, cross-correlation) How do these time series compare to actual data (e.g., OWDA, HYDRO2K), even if the variables are different, there should still be some similarities? A minor issue but might avoid confusion: I would recommend using years CE/AD on the scale (I am more often dealing with yrs BP so I tend to read that directly… It's a matter of community standards)
  - o Line 319: it might be me not being familiar with the jargon (and perhaps I missed it earlier, sorry): what are EA-WR and Eastern Atlantic pattern (perhaps need some description earlier in the manuscript?). I would generally recommend refraining from using abbreviations if not needed, it is much nicer to the reader in plain words.
  - o Fig. 5: how reliable are these results with regard to Fig. 3?
  - o Fig. 6: Very hard to read and to capture the essence of these results. But once again, it seems to me here that the authors are showing an evaluation of the different models, so there is a tension between showing actual climatic results (fig 5) and showing how models perform and the inter-model spread, which is hard to reconcile.
  - o Line 338: "the mean occurrence of pattern groups for each model": what is exactly meant here?
- Discussion:
  - o On the NAO/EA-Mediterranean climate relationship: this is very interesting but I am a little puzzled to see that both NAO configurations can be associated to droughts. I am wondering if that could be discussed more? Another point of curiosity: can you detect in the time series (not averaged in 100-yr windows) any fingerprint of NAO (e.g., 7 yrs periodicity)? The identification of a "multi-decadal scale anti-phase" is very interesting and could also be further explored (is there an oscillatory component? Has it been observed before?). I had to think of an article by Mann et al. (Nat. Comms 2020) "Absence of internal multidecadal and interdecadal oscillations in climate model simulations", not sure if it is relevant here.
  - o Is there any way possible to test the observed seasonality pattern of climatic associations in the OWDA? Do you also observe a N-S antiphase (also

mentioned in Markonis et al. 2018)? The E-W antiphase: is it stable on all timescales (Indeed the results you obtain are contradictory to Cook et al. 2016, this might be further discussed)?

- o I am also surprised to read that climate background (e.g., global temperatures) had no effect on droughts (or did I understand wrong?). See for instance Dermody et al. (Clim Past 2012).
- o Minor comment: perhaps cite Douville et al (2021, Water cycle change IPCC report) instead of Masson-Delmotte et al (2021)?
- o Finally: I liked the approach of Hanel et al (Sci. Rep. 2018) to distinguish meteorological (rainfall), agricultural (soil moisture) and hydrological (runoff) droughts. Perhaps an idea here to frame the research (most previous papers are looking at meteorological droughts)?

---

## Author Comment (AC3)

**Response to Reviewer 1**

We would like to thank the reviewer once again for his/her exhaustive and constructive comments. We sincerely appreciate the time and effort the reviewer dedicated to reviewing our manuscript.
In this response, we will provide more detailed explanations addressing the reviewer's comments, as well as presenting our plans for further analysis.
Our responses are in blue font.

*GC2.*
*1) Page 1, lines 21-23. 'The climate of the...' This paragraph divides the seasons into wet and dry and refers to the wet one specifically as the winter season. There is considerable spatial variability in the rainy seasons in the Mediterranean lands and for many, the wet seasons are spring and autumn. I think this winter/summer separation may be misleading. Consider for instance Xoplaki et al (2004; DOI 10.1007/s00382-004-0422-0) or others, where an extended wet season is used. I think this discussion should incorporate better this seasonality character. Note page 2, line 24 where you mention 'highly variable seasonal hydroclimate...', although the end of this paragraph (line 30) will make the emphasis again on winter.*

Thanks for the point. We will correct the paragraph and include more details about the extended seasonality character of the region.

*2) Page 3, lines 64-65. 'They also found that ... is model-dependent'. This seems to relate specifically to the results of the current manuscript and perhaps should be discussed specifically either here, providing more specific background and describing in which way this can happen, or/and at the end of the manuscript discussing what this manuscript provides in*
*the context of previous evidence.*

We will include more details about this particular article (Xoplaki et al., 2018) in the introduction.

*GC3.*
*3). Section 2.1 Page 3, lines 83-86.*
*The CMIP and PMIP and other model efforts could include a reference. I am aware they were just provided in the previous paragraph, but it seems pertinent to me that they get those also in the Data section. I leave this to the taste of the authors.*

We will incorporate the reviewer's suggestion in the next phase.

*4). Section 2.1 Page 3, lines 85+ 'We consider only simulations...'*
*- There are some issues that can be considered related to the use of soil moisture in different models: The models have very different depths and therefore they will possibly produce different soil moisture statistics, even if depths only down to 0.7 m are considered. Using this set up of models is fair and possibly it contributes to inter-model differences, but also it is arguable that models with shallower depths may*

*be less realistic. Note that some models like GISS only have a depth of 3m. This will produce potentially a different vertical distribution of moisture and different temporal variability. Perhaps it is something the authors would like to comment on.*

Thanks very much for the point. We agree with the reviewer's comment regarding the need for an additional description of soil moisture for each model and a discussion on inter-model differences in our study. We will include this detail in the revised manuscript.

One point that we want to comment on is that soil depth below two meters is generally considered less important for atmospheric processes. Hence, the fact that GISS has only a three-meter depth would not significantly affect the atmospheric processes we focus on. However, vertical soil layers (up to two-meter depth) could be a more important factor contributing to the observed model differences. We briefly discussed this issue in the discussion section relating to the discrepancy between MIROC-ESM and other models (lines 366–373). We will extend this discussion.

*5) In Table 1 it is included that a depth of 0.7 m is considered. I think this is relevant and should also be included in the text. It would be desirable to include a rationale for this decision and how it may influence the timescales of variability of soil moisture. What is the typical range of depths for the bedrock, that limits the presence of water, for the area of interest in general in these models? Including a shallow limit may be more representative for the whole area but also exclude lower frequency variability more typical of deeper levels. I think including a rationale for this in the text of Section 2.1 would be good.*

As we responded to comment 4, we will include more details on soil moisture and soil components of the models in the revised manuscript.

*6) 3. Section 2.2. Why are NOAH-LSM data used for soil moisture and ERA5 only for circulation? I mean what does NOAH-LSM offer that you would not get in this manuscript from ERA5 soil moisture?. There is probably good reasons for it. I would just suggest that a motivation for the use of these data, beyond the fact that soil moisture observations are scarce, is provided.*

We used the soil moisture from NOAH-LSM instead of ERA5 because NOAH-LSM is forced with the observation-based dataset and the reanalysis data that the biases were corrected with respect to the observations (https://hydro1.gesdisc.eosdis.nasa.gov/data/GLDAS/GLDAS_NOAH025_3H.2.1/doc/README_GLDAS2.pdf). ERA5 does not directly assimilate any rain gauge data except for the United States (Lavers et al., 2018). Therefore, we assume that NOAH-LSM could be more appropriate to show more realistic present-day soil moisture variability, which is mainly influenced by precipitation variability. We will add this detail to the manuscript.

*7) Line 87 '... analysis: Giss-E2-R, CCSM4...'*
*References should be provided for these simulations. Considering previous points, I would suggest that a minimal background is at least provided in how different the soil moisture modelling can be among all models. This would help giving the reader an*

*idea of what to expect in terms of inter-model differences. For instance, vertical resolution can be very influential for soil moisture, as indicated in line 90...*

We will include this detail in the revised version (see our responses to comments 4 and 5.).

*8) Line 87 '... 12 ensemble members of CESM1'*
*There are 13 available if I am not wrong. Nothing critical but maybe you want to state why that selection. Here it can also be stated this refers to the all forcing simulations, although it is quite clear in the context (see next).*

When we retrieved the CESM-LME dataset from https://www.earthsystemgrid.org, the first ensemble member (member 001) of the variable geopotential height (Z3) was missing for 850 – 1849. Hence, this member was not taken for the analysis. We will include this explanation in the new version of the manuscript.

*9) Line 91 ' All simulations were run with the volcanic, solar...' It is clearly stated earlier in the introduction that this is all about internal variability. Perhaps some comment about focussing on forced simulations and not in long control runs (also available in cmip) is pertinent.*

In our research, we focus on investigating the natural variability of drought and extratropical circulation associated with the events during the last millennium. Hence, the effect of the post-1850 increase in GHG is excluded, but not the volcanic forcing as is a natural internal forcing. We do not assume from the beginning that internal variability is the main driver of drought (although our result, in the end, shows this point). We are aware that our text is misleading on that point, therefore, we will correct the manuscript for clarification.
As we mentioned already, we use the last millennium simulations instead of long control runs as we examine drought variability during the last millennium.

*10) Line 101 ' ...has four layers up (down) to two meters'*
*Ok, but I understand that only the soil moisture of the first 0.7 m is used, right? Explaining this in relation to the previous information in Section 2.1.*

Yes, from NOAH-LSM, also the soil moisture at 70 cm level was used. The sentence was to describe the soil variable of NOAH-LSM. We will include the same information for other climate models in the revised manuscript.

*GC4. 3. Methods*
*3.1 Calculation of anomalies.*
*I find the definition of annual anomalies and the explanations for it somewhat confusing. There is an emphasis in this explanation that I do not really understand. Perhaps this is my fault but, please, consider it and see if this influences the text as it stands now.*

*11) 1. Section 3.1 Page 5, lines 111+.*

*- 'These variables are transformed into the annual mean anomalies at each grid point'.
At this point I would say it is not possible to understand what is meant if additional
information is not provided first, and I would argue to simply write '...transformed into
annual anomalies'. I think that will be enough also for the rest of it.*
*- 'We use annual mean anomalies in order to include winter conditions in the analysis,
as it is an important season for the annual hydroclimate in the Mediterranean'*
*I think it is important to discuss this in the context of what is indicated in GC2.1.
Specifically for this sentence, it is not possible at this stage to understand what is
meant. Not a big thing because one can perhaps understand it from the following, but:
a) I would suggest to include the description for the intra-annual anomalies before
and then explain they are averaged; if these arguments stay in the text.*

Here we wanted to emphasize that we use the annual mean time series to include
hydroclimate conditions of all seasons, including the wet seasons (referred to as only winter
in our study, but we will correct this according to the reviewer's first comment.), unlike other
studies focusing on the region that usually consider only the summer mean time series. We
will clarify our point better in the revised manuscript.

*12) b) I would actually (respectfully) challenge that this is true. I think it is not, but
maybe I understood things wrong. See below.*
*- 'Prior to the anomaly calculation ... to the 70 cm level'.*
*These two sentences are relevant and the reader could go with knowing this earlier in
the data section (GC3.2).*

We will include this change.

*13) 'Then, first, the monthly anomalies... monthly values' [...] 'Second, the necessary
annual mean ... these monthly anomalies'.*
*I do not understand these two sentences. If you calculate monthly anomalies
(differences between monthly data and the long-term monthly mean you are
essentially subtracting the annual cycle. If you average out the resulting monthly
anomalies what you get is the annual anomaly, equal to calculating anomalies from
annual data by subtracting the long-term mean. Therefore, I do not understand the
previous sentence about the importance of winter or the unnecessary emphasis on
the monthly anomalies if you end up in annual anomalies. The sentences in the
middle about reference periods read fine to me. Perhaps it fits better with the next
paragraph and the issue of the trends since their definition is somewhat related.*

We agree with the reviewer's comment, therefore, to avoid redundancy, we will remove the
sentences about the annual cycle in the revised version and move other sentences in the
middle to the next paragraph.

*14) 'Thereby, the ensemble means of the anomalies... Maher et al. (2018)'.*
*With the preceding sentences I am confused. I understand the purpose is subtracting
the ensemble average to each member to get rid of the long-term trends; or better
said of the forced response in general as it will also affect volcanic events for
instance. This would be consistent with the following arguments in the paragraph.
Consider explaining/rephrasing these paragraphs.*

It would not affect the volcanic events during the pre-industrial period (850–1849) as the ensemble averages were only subtracted from the historical simulations. We will clarify the paragraph in the revised manuscript.

**15) 'This method guarantees that only internally driven variability remains in the time series of the variables'. ... mostly internally driven variability, but only in the post-1850 period. This should be perhaps indicated specifically. However, what I miss a bit is the rationale of why it is done like this. Why is the forced signal intended to be filtered out in the post-1850 (natural and anthropogenic) and not in the pre-1850 period? The reader has to guess that perhaps this is intended to get rid of the long-term trends, and the rest of the forced signal goes with it**

The filtering is applied only to the post-1850 period mainly to remove the anthropogenic GHG forcing which becomes apparent after 1850. We did not apply the filtering before 1850, as there is no apparent long-term up- or downward trend in soil moisture before 1850. In addition, except for CESM and GISS, other models have only one run during the last millennium which inhibits applying the method to all simulations. We will try to clarify this paragraph better for the next phase.

**16) Note also that the ensemble is calculated over different models in CMIP5-PMIP3 if I understood well (?). Now this is an issue that can be relevant, as different models will show different levels of response to external forcing. When this is done over an ensemble with the same model you can reduce the external response in each model run as it is assumed that the external response is common to all runs. One aspect here is that depending on the number of runs you are also subtracting also internal variability that has not been averaged out in the ensemble average, and this is a limitation that can be discussed.**

The ensemble means are calculated for each of the five CMIP5-PMIP3-CESM and not over all CMIP5-PMIP3-CESM models. Therefore, each model has one ensemble mean for the period of 1850-2005 (These means are shown in Fig 2.c). Then, the ensemble mean is subtracted from each of the historical simulations of the corresponding model. We may not have been clear on this point, therefore, will try to clarify this paragraph better.

**17) A more relevant aspect can be that if you are using different models in the ensemble as it is the case, the assumption that a common response to external forcing is filtered out does not hold and by subtracting from each model the ensemble average you do not actually filter out the external response of that specific model. Therefore, this would work for the CESM-LME ensemble but not if you mix in the ensemble different models. This can be a sensitive issue that should be at least clearly discussed here or indicated here and better discussed later in the text.**

See our response to comment 16. The ensemble mean is calculated for each of the models and later, subtracted from each of the historical simulations of the corresponding model. Therefore, we filtered out the external response of that specific model, which takes into account the model's climate sensitivity.

*18) The two periods, LM and Hist, ... models and regions.*
*The pre-1850 and post-1850 are merged. However, they do not have the same nature, as in the pre-1850 the natural externally forced variability exists and in the post-1850 it is not intended to exist. Although if calculations have been done for some models out of a multi-model ensemble, some leftovers of externally forced variance should remain. Nevertheless, this should be in principle not important if externally forced signals are not pursued and are not expected to play a role either; actually, for that purpose, control runs could be as adequate or more adequate. I am trying to highlight with the previous sentences that some level of confusion can be transmitted to the readers.*

Thanks very much for the point. We are aware that the two simulations (pre-1850 and post-1850) do not have the same nature, as they were not run seamlessly and the external forced variability is only subtracted from the historical simulations. The two periods are merged only to calculate continuously the temporal variability of droughts in Fig. 4. But note that for the pattern detection, each period was fed separately to the algorithm. We will clarify this point better in the revised manuscript. About control runs, see our response for comment 9.

*19) 2. Section 3.3 Page 6.*
*I wonder why not considering the soil moisture from ERA5 or ERA5- land additionally or instead to NOAH-LSM. This relates to GC3.3. Since ERA5 produces soil moisture out of its assimilation system, this would be physically consistent with other ERA5 variables used here. Perhaps the authors want to make some comments in the text about the advantage of using the NOAH data here instead.*

See our response to comment 6.

*20) 3. Section 3.4 Page 6, lines 161+*
*- 'When these temporal and spatial ...'*
*The regional arguments about drought incorporate the condition of having 60% of the horizontal gridpoints with negative SOIL. This faces some difficulties with the fact of using different horizontal resolutions. However, this could be overcome by considering the spatial size of anomalies by using grid box area and the total spatial extension that the anomalies represent, instead of the number of gridpoints.*

Thanks for the point. Indeed, we used the number of grid cells as a spatial threshold because the models have different horizontal resolutions, and it would be difficult to set the same threshold based on the spatial extension. We checked that the 60% of the total grid cells correspond to about 56.20% (if covering only the southern grid areas) – 61.54% (if covering only the northern grid areas) of spatial extent in the western Mediterranean, and 56.31% – 61.65% in the eastern Mediterranean (see the table below). These values are close to the 60% level that we use with the number of grid cells.
The maximum difference in the spatial coverage between the models is around 4%, which for all models except CCSM4, is the spatial extension of one grid cell. Therefore, using a threshold proposed by the reviewer that is based on the spatial extent may exclude a few drought events in our analysis, but it would not change our results much. We will add more details on our choice of threshold in the revised manuscript.

In addition, we noticed that we made a mistake in the number of grid cells for CESM and GISS-E2-R in Table 1 in the manuscript. We will also correct it.

Table 1. The number of 60% of land grid cells and the corresponding spatial coverage (spatial extension divided by the total area of the region) for the southmost (minimum) and northmost (maximum) regions.

| | % of spatial coverage | | |
| --- | --- | --- | --- |
| | Minimum coverage (southern area) | Maximum coverage (northern area) | Difference between the minimum and maximum (%) |
| **WEST** | | | |
| CESM | 58.49 | 61.54 | 3.05 |
| GISS-E2-R | 58.01 | 61.05 | 3.04 |
| CCSM4 | 57.81 | 61.59 | 3.78 |
| bcc-csm1-1 | 56.20 | 59.58 | 3.38 |
| MIROC-ESM | 57.56 | 60.71 | 3.15 |
| **EAST** | | | |
| CESM | 56.31 | 58.97 | 2.66 |
| GISS-E2-R | 56.89 | 59.61 | 2.72 |
| CCSM4 | 58.15 | 60.98 | 2.83 |
| bcc-csm1-1 | 58.20 | 61.65 | 3.45 |
| MIROC-ESM | 57.13 | 59.66 | 2.53 |

*21) 'Next, the weighted spatial average of SOIL is ... Mediterranean'*
*How is this weighting done? Perhaps I missed it earlier in the text? If this is done, why not using this regional average to determine the occurrence of spatially large enough droughts? (??). I do not think the result will be very different from the previous approach and it would overcome the use of a percentage of gridpoints with different model resolutions.*

The weighting average was performed by weighting the soil moisture values considering the area of the corresponding grid cell. We will include this detail in the revised manuscript.
We added a spatial constraint on droughts by taking into account the percentage of grid points over the region to make sure that we take droughts that have a considerable spatial extension and are regional events and not local events (lines 159-160).
We could also have used directly the spatially weighted time series and taken a certain threshold for droughts, but in this case, a few strong negative anomalies in a few grid points can cause strong negative anomalies, not ensuring any spatial extension of droughts.

*22) 4. Section 3.4 Page 7, lines 171+*
*'Only climate models with some ensemble members ... not detectable'*
*This is already a report of results. I have no major concern with anticipating it, but what is the reason for it?. The sentences are descriptive of the fact that pan*

*Mediterranean droughts happen in some simulations with an specific feature and not in others. Can you at this stage argue about this feature?, or should this moved further down in the text and a discussion provided?*

We will move the sentence to the result section.

*23) 5. Section 3.5 Pages 7-9*
*I get the structure of the methodology in general, however, I suggest the authors revise this section for a more clear explanation, revising notation and perhaps the current state of the explanation of details of the methods that can be more important for the understanding of this section. I will not go to details in an exhaustive way, just provide some examples. The text is providing the sequence of a method. How the outcome of each of the steps feeds the following steps should be clear in explanation and, I would suggest, a homogeneous notation. I also recommend there is a rationale/justification for the conceptual use of each step. I will briefly try to highlight this with examples on the following, but please, go beyond those.*

Thanks for the comment. We will adapt the notation so that everything is consistent throughout the entire section.

*24) References: the authors provide reference but please take care they are appropriate. For instance, there are many ways of applying PCA and the reference to the correct texts that describe the approach presented herein should be provided (see below).*

Thanks for the comment. We will go through the section to clarify the text.

*25) Use of maps and series: This should be clear from the text. For instance, in the PCA, I understand T(t) are the principal components. Please, indicate that and also the range of the parameter t. The same with s, it is good to indicate the range so that the number of modes or the retained number of modes is well defined.*

We will correct these issues in the revised manuscript.

*26) In equation (1), where are the eigenvalues?. I understand they are multiplying either the spatial or the temporal component. They should be indicated. I understand it is particularly relevant if they multiply the temporal component. The reason for it is that the temporal component will have a standard deviation 1 or different from it and this will impact the Kmeans procedure as it will affect the distance.*
*Explained variances are mentioned but it is not said that the 70% reflect accumulated variance accounted for by a number of pcs. Some sentences are confusing. E.g. line 191 'PCA is applied to the Z500 fields during droughts for each model'. I do not understand this.*

We will include these details in the paragraph and correct the sentences for clarification.

*27) How do the PCs enter the following analysis? They have unit standard deviation (perhaps not), and their inter-pc correlation is 0. How does this play a role in the*

**Kmeans clustering, what does it mean physically because pcs should indicate different modes in time that are afterwards grouped, even if they are uncorrelated.**

The principal component analysis (PCA) is mainly applied to reduce the spatial dimension of the dataset and increase the performance of the clustering method. With the PCA, the new field of PC($t$)s are obtained from the whole spatio-temporal (*time x latitude x longitude*) Z500 with $t$ drought years. The first $N$ PCs, $N$ in our study ranging between 5 and 6, are taken depending on the silhouette coefficients and explained PC variances. The dimension of space of PC is now reduced to $t \times N$, instead of $t \times latitude \times longitude$ of the original dataset. The K-mean clustering is applied to group similar Z500 in this $t \times 5$ PC space, then assigning the label to Z500 patterns that belong to the same cluster.
It is true that the PCs are uncorrelated, but the clustering method considers the geometrical distance between the points (euclidean or Mahalanobis), and not the covariance between them. It is common to apply PCA before grouping the data for a better performance. We will revise the section for clarification.

**28) The notation of how the T(t) go into equation (2) should be consistent with this equation. I suggest that the notation is blended for the various steps of the analysis. If it is not done, readers will have to accommodate how things fit from their knowledge and from the different steps of the method. If notation is blended, this section would actually describe one thing, the approach followed in the manuscript, not several independent methods. Some features of notation are repeated for different things, e.g., s as parameter for points in space and the Silhouette coefficient for each point.**

We will correct these issues in the revised manuscript.

**29) There are 71 clusters. I learned this from the text but realized that this is the sum of all clusters from different models in Table 2. For each model 5 to 6 pcs are retained and from this, between (mostly) 3 and 6 clusters are formed. What does this mean? Are the 3 clusters gathering the information of the 5-6 pcs? In which way? Some rationale/explanation for what is conceptually happening is good for the reader.**

Also, see our response to 27. The K-mean clustering groups the circulations (Z500) of drought years. After grouping the clusters as explained in 27, the mean values of each cluster are calculated using the Z500 anomalies that correspond to the drought years of the cluster (instead of using the normalized projected values of the PCs). Hence the clusters are able to be correlated based on their Z500 anomalies. We notice we did not include this detail in the manuscript. As we responded in comments 24 – 28, we will revise the section, and if we find it necessary, also we will add this detail to the diagram (Fig 1) for a better understanding.

**30) Line 220: at this level I do not know what the correlation between clusters means. All in all, section 3 needs, in my opinion, to be well revised do deliver a more clear and consistent text. This does not necessarily imply changes in the calculations, nor the results of the following sections. But it may impact (positively) the interpretation.**

See our response to 24.

*GC5.*
*4. Results*
*1. Related to previous comments:*
*4.1 Observation-model comparison*

*31) Line 237: '... by subtracting from each of the ensemble members the anomalies at each grid point'. This relates to previous comments and could be explained better.*

See our response to comment 16.

*32) Line 238: 'The spatial correlations between SOIL and Z500 of NOAH-LSM and each of the climate simulations are presented in Fig. 3'.*
*I do not think it is wrong at all but I wonder what is the gain of using NOAH instead of the soil moisture from ERA5 in this figure.*

See our responses to comment 6.

*33) 2. Figure 2 and related*
*2b and 2c are good in showing the impact of using different references. However, I would say that the two are not really needed. With one of them it would be enough to explain it. Perhaps the space could be saved to accommodate a time series of the full 850-2005 period. I leave it for the authors to decide.*
*The shading in Fig 2b seems to indicate consistency in the range of variability with the range of 'observed' NOAH-LSM variability, which is good to indicate.*

We will try to accommodate the full 850-2005 in the Fig 2 area. Also as reviewer 2 commented that Fig 2 is under-utilized, we will add more details about these time series and the NOAH-LSM – CMIP5 comparison in the revised version.

*34) 3. Figure 3 and related*
*Watch some statements like ' ... negative correlations over southern Europe, but the correlations outside Europe are not significant'. They are for CCSM4 and bcc in the western low latitudes of the domain.*

*We will correct it in the revised manuscript.*

*35)  Line 255, '... all models present similarities to the NOAH-LSM, fed...'*
*What we see in Fig. 3 is the result of both NOAH-LSM and ERA5, right?.*

Yes, that is correct. We will correct the sentence.

*36) Actually, it is likely that the large-scale structure we see there is more dependent on the global model; one could actually test if it changes with other reanalysis products. I think this is likely out of the scope of the study, but it is not totally off the line of argumentation because this correlation field with ERA5 is what we consider 'truth', but it could change if we would have used a different reanalysis product.*

We do not expect to see much difference between different reanalysis products as the present-day reanalyses are assimilated with similar observational-based data (although with different models). Here we provide the maps of correlation between the Z500 from the NCEP/NCAR reanalysis 1 (https://psl.noaa.gov/data/gridded/data.ncep.reanalysis.html; Kalnay et al., 1996) and the NOAH-LSM soil moisture anomalies (Fig 1.b and d), along with the correlations between the ERA5 Z500 and the NOAH-LSM soil moisture (Fig. 1.a and c) which is in Fig. 3 of the manuscript.

The two datasets basically show similar structures. A slight difference is because of the horizontal spatial resolution (NCEP-NCAR has a coarser resolution of 2.5 x 2.5 degrees compared to ERA5 of 0.25 x 0.25 degrees).

[Figure]

*Fig 1. Correlation coefficients between the ERA5 Z500 and the NOAH-LSM (a) west, (c) east Mediterranean soil moisture anomalies, and between the NCEP-NCAR reanalysis and the NOAH-LSM (d) west, (d) east Mediterranean soil moisture anomalies.*

**37) Perhaps a more relevant issue: if you consider the variability among patterns in Figure 3, what is the variability among simulations of one single model ensemble?. I would assume that it is smaller, but it may be worth reporting.**

There could be some variability between the ensemble members, although we do not expect the centers of correlation would differ much between them. See the response to comment 38 (below).

**38) In the actual figure 3, for those models with ensembles, is the pattern that is shown the result of one single experiment? If so, I think it should be clearly stated in the caption and in the text. I would advise against including correlations using ensemble averages, but I don't think this is what is being shown.**

Yes, the correlation fields in Fig 3 are from the first member of each model. As the correlation analysis using the ensemble means may smooth out the fluctuations, we provide here the mean correlation maps of three ensemble members of CCSM4 (Fig 2 below). Compared to the correlation fields of only one member (what is shown in Fig 3 in the manuscript), Fig 2 (below) shows more smoothed correlation coefficients. However, the locations and signs of patterns do not significantly change.

[Figure]

*Fig 2. Correlation coefficients between Z500 and soil moisture anomalies in CCSM4 for (a) and (b) one ensemble member, and (c) and (d) the means of three ensemble members over west and east.*

*39) 4.2 Mediterranean drought …*
*1. Figure 4 and related*
*In relationship to the role of external forcing, or the lack of it ('This fact emphasizes that external forcing signals do not play a role in droughts over the Mediterranean ...' line 286), there are several issues that may be worth commenting.*
*- How are droughts calculated over sub-ensembles of simulations (GISS, CESM1) in Fig. 4, and how is the ensemble spread provided for them? This should be explained (sorry if I missed it) in the text and figure caption.*

Drought years are calculated for each ensemble member using the definition in Section 3.4. Fig. 4 shows the running count of droughts every 100 years which is calculated for each member, and the ensemble spread is a unit standard deviation of the time series of counts of all members. We did not include this detail, therefore, we will add it in the revised version.

*40) Also, I expect it will justify the different temporal resolution of the curves in Fig 4 for GISS and CESM1 in comparison to the others. However, it can be misleading as the results of those two models may be read as if soil moisture for CESM1 would be consistently higher (smaller) in the western (eastern) Mediterranean during the late 17th and 18th century, or in GISS also during the 18th and early 19th centuries... or in*

***GISS opposite to that during the 17th century. This would not be possible and would contradict the first statement of no role in external forcing as it is very unlikely that different model runs of a sub-ensemble with different initial conditions will coincide systematically in simulating relatively dry or wet periods unless forcing would play a major role****. The only reason for that would be external forcing. Also, the finding of opposite phasing between west and east is interesting, and I would argue that it should be more perceptible for GISS and CESM1 if individual runs are considered, in relation to the statement.*

We guess this detail can be better shown when we provide an analysis of the time series for each ensemble member. Below, we include the time series of some ensemble members in CESM and GISS plotted separately in Fig 3.

Notice that in the figure, we do not see anymore a synchronous temporal pattern between the east and west or between the two members of the same model in the same region. Hence, the synchronous temporal pattern observed in Fig 4 in the manuscript mainly comes from averaging all ensemble members. We do not think we can add all the plots of individual ensemble members of all models, but we will include more discussion on this inter-member difference in the revised manuscript.

[Figure]

*Fig 3. Occurrence of drought years in a moving window of a century in the western (red) and eastern Mediterranean (blue), for some members of CESM and GISS-E2-R.*

*41) '...observed, more clearly in those models and periods with one ensemble member' (line 296). Therefore, I suggest the authors revise how the ensemble behavior is presented for those two models. For instance, the last statement of Section 4.2 is sensitive: 'For those models and periods with more ensemble members... sometimes this association is blurred...', it should indeed be the expected behavior, even more than what is shown. If you resort to individual simulations, this should be more clearly evidenced. The ensemble spread should expectedly blur everything since a dry or wet century in one run should not be expected to be consistently dry or wet in most of the other sub- ensemble runs.*

See also our response to comment 40.

*42) 2. Figure 5 and related*
*I am not against showing temperature anomalies in association to the Z500 patterns, but why not showing precipitation and actual drought patterns. How do the geopotential anomalies account for drought occurrence?*

We included the temperature anomalies as the variable can increase the intensity of the events under the presence of high-pressure systems (Zhou et al., 2019). We will include more details on the effects of temperature and geopotential height anomalies on droughts in the revised version.

*43) Figure 6: I need a better description of the methods section to better interpret results and figure out whether they can be dependent on methodological choices.*

See our response to comments 23 to 30.

We will correct the minor comments in the revised manuscript.

**Reference**

*Kalnay et al. (1996), The NCEP/NCAR 40-year reanalysis project, Bull. Amer. Meteor. Soc., 77, 437-470.*

*Lavers, D. A., Simmons, A., Vamborg, F., & Rodwell, M. J. (2022). An evaluation of ERA5 precipitation for climate monitoring. Quarterly Journal of the Royal Meteorological Society, 148(748), 3152-3165.*

*Zhou, S., Williams, A. P., Berg, A. M., Cook, B. I., Zhang, Y., Hagemann, S., ... & Gentine, P. (2019). Land–atmosphere feedbacks exacerbate concurrent soil drought and atmospheric aridity. Proceedings of the National Academy of Sciences, 116(38), 18848-18853.*

---

## Author Comment (AC4)

**Response to Reviewer 2**

We would like to thank the reviewer once again for his/her exhaustive and constructive comments. We sincerely appreciate the time and effort the reviewer dedicated to reviewing our manuscript.
In this response, we will provide more detailed explanations addressing the reviewer's comments, as well as presenting our plans for further analysis.
Our responses are in blue font.

*Main comments*

*Before I list specific points in the manuscript, I wanted to raise the issue to the authors (it is not a must, more a proposition): I personally found Fig. 2 and 3 very interesting but under-utilised (and to some extend Fig. 6 too). Clearly, some models are not very skilful at capturing the synoptic climate during historical droughts and this raises in itself an important issue:*

***1) how can we confidently understand drivers using longer simulations? What setup does "work better" to capture what part of the signal (temporal vs. spatial)? I know that expending on this aspect would modify the paper substantially but it is of crucial importance to build on. It would also give more confidence to the method applied later on to determine the dominant climatic modes (I must say, as is, I am wondering how much trust we can have – which might also be due to the fact that I am not savvy with the methodology used here).***

Previously, we gave a short first response, but here we add some more of our thoughts on this question and how we plan to tackle it.
To understand which set-up works better, it is necessary to perform sensitivity tests changing model configurations which would not be possible using all these CMIP5 models and is out of the scope of this study. Our objective is to identify extra-tropical patterns related to past droughts in CMIP5 model simulations and to understand the difference between these model simulations.
Importantly, we cannot expect the same responses from all models due to their internal physical differences, initial conditions, and model-dependent internal variability (which is what our result also shows at the end).

As we responded previously, in general, if the models are able to represent well the present-day climate modes in an acceptable way (as performed in several studies, e.g. Fasullo et al., 2020; Deser et al., 2018), we assume that the representation of the modes would be still valid in longer simulations.
To test whether the simulations work well, we compared the variability of soil moisture (Fig. 2.a) and soil moisture-geopotential height correlations (Fig. 3) between the CMIP5 simulations and the observation-based data in the manuscript. To assess which one is still better than others at representing the variability of soil moisture, we will add the ensemble spreads of each individual CMIP5 model and compare them against NOAH-LSM in Fig. 2.a in the revised manuscript. In addition, we will add more discussion regarding these methods and complications of model-observation comparisons in past and present simulations.

*2) Another (more minor) issue: Perhaps I am not familiar with the term, but it would be useful to clarify what you mean by "internal variability" (and that might be done by just explaining what is considered an external forcing in your study). If not used to the terminology, one might wonder for instance, whether teleconnections are part of the internal variability or not?*

With internal variability, we refer to different modes of climate variability involved in droughts. By external forcing, we mean volcanic and solar forcings. However, volcanic forcing is an internal climate forcing and not an external forcing. We will correct and clarify better what internal variability and forcings mean in our study in the revised manuscript.

*3) Abstract: lines 4-5: "The focus is [...] during 850-2005, this excluding the anthropogenic trends from 1850 CE onwards". This is very confusing... It can only be understood if the methodology has been explained, otherwise one starts to wonder which time range was actually used.*

We will correct the sentence.

*4) Introduction: Lines 66-71, the definition of the knowledge gap and the leading questions is a little weak. "necessary to understand which modes of climate variability or atmospheric circulation patterns are involved in each climate model": this sounds very ambitious and is actually not what this study is tackling. Instead, it might be useful to explain why studying the "patterns in the mid-latitudes" and their effect on Mediterranean climates is important.*

Thanks very much for the point. We will include more details on this in the revised manuscript.

*5) S3.1: this section of the methods is at times quite confusing. I have read the comments of Rev. 1 and agree with their comments so I will just emphasize two points that I found particularly confusing:*

As both reviewers commented, we will go through and revise the section for the next phase.

*6) Line 112: "We use annual mean anomalies in order to include winter conditions in the analyses [...]": I don't understand this sentence.*

We will correct the sentence as reviewer 1 also commented about it.

*7) Lines 121-126: This part is at the same time important and confusing: why not normalising the whole record instead of just the Hist part? Would a z-scoring be useful?*

The aim of using the method in line 121 is to remove the trends introduced by the anthropogenic increase in GHG after 1850 to focus on studying the natural variability of drought during the last millennium. We will write this paragraph better for the next phase.

**8) S3.5: section very hard to follow, please provide additional information and background on the choices of the methods and their limitation.**

We will go through and revise the section.

**- Results**
**9) Fig.2: if I understand right, this is your control run for time series.**

All time series are from forced last millennium and historical simulations. We will include this detail in the caption.

**10) a) the modelled soil data are quite different from the observation time series (range and trend are similar but changes are often not synchronous). How does that affect your analysis? What does the intermodal comparison look like? Could you have a look at the cross-correlation between NOAH LSM and CMIP5 mean? Just so that we understand the limitations of the study.**

We do not expect the models' soil data to be exactly synchronous with the observation-based time series. That is basically not possible because of the difference in the initial conditions between the models and the observation, and internal variability of the climate system. As long as the variability of the observed variable lies within a model spread, we assume that the model is able to represent the temporal variability of the observations (e.g., Liu et al., 2012). For the same reason, a cross-correlation analysis between NOAH and CMIP5 would not provide additional useful information. However, we can examine the spread of each individual model utilized here (as we responded to comment 1) to verify each model's soil variability and possible limitation compared to the observed data. We will include more discussion on this in the revised manuscript.

**11) Line 238: "SOIL and Z500 from NOAH-LSM" isn't Z500 from ERA?**

That is true. We will correct it.

**12) Fig.3: control run for spatial analyses. I think that it would be good to indicate that the upper left panel shows the observation (SOIL vs. Z500). How do you deal with models not capturing the synoptic climate very well (are they still used for further analyses, and if so, why?)? Can you quantify the skillfulness of the models? To me this exercise seems to be providing an evaluation of the models to track spatiotemporal climate conditions associated to droughts. It could be a goal in itself (as proposed earlier).**

We did not have control simulations in our analysis (see our response to 9). Regarding the question of how to deal with models that do not capture well the observed synoptic climate, as one of the objectives of our study is to understand the model differences in representing drought-related circulation over the Mediterranean region, we do not apply a selection criterion to choose the best or worst model (Also see our response to comment 1). Instead, we address and visualize the differences between the models in representing past drought-associated extra-tropical patterns. We already discussed briefly the implications of the model discrepancies in the discussion section (for instance, about MIROC in lines

370-373), but we will extend the discussion more and clarify our objective in the revised version.

Regarding the skillfulness of the models, it would be hard to assess the skillfulness of all variables associated with droughts, and there is no universal metric for it. We somehow asses the skillfulness of the models in representing soil moisture-associated circulation patterns by comparing the correlation fields between the soil moisture and geopotential heights of the observation-based data and model simulations in Fig 1. We can add, for instance, root mean square errors calculated between the spatial correlation coefficients in Fig 1 to summarize the figure and come up with some numbers.

**13) Lines 286-288: I struggle with these sentences. Either too little or too much is said here. What is the role of model's setup and skilfulness in this observation?**

We will correct the sentences for clarification. About the models' setup and ability, see our response to comment 1, and about their skillfulness, see our response to comment 12.

**14) I also do not understand what is meant by "counterfactual"?**

By "counterfactual", we refer to the climate conditions without the anthropogenic trends from 1850 onward, so the detrended Historical simulations. We will modify this word and also rephrase the sentences for clarification.

**15) Fig.4: Would it be possible to quantify the antiphase? (bi-plot, cross- correlation)**

Thanks for the suggestion. We will include more analysis of the time series (for instance, as the reviewer suggests, a cross-correlation analysis between the west and east regions) in the revised manuscript.

**16) How do these time series compare to actual data (e.g., OWDA, HYDRO2K), even if the variables are different, there should still be some similarities?**

We provide here a more extended response than the previous one:
Regarding actual data (for instance, OWDA), Kim and Raible (2021) have performed the comparison between summer droughts in a CESM simulation and OWDA, and found differences in the temporal occurrence of droughts between the model and the proxy-based reconstruction. If we include OWDA, we expect to see a similar result here (not temporarily synchronous). Also, OWDA provides only the summer hydroclimate variability of the region and not all annual means. We will include more discussions on the finding of Kim and Raible (2021, Clim. Past) in the revised manuscript.

**17) A minor issue but might avoid confusion: I would recommend using years CE/AD on the scale (I am more often dealing with yrs BP, so I tend to read that directly... It's a matter of community standards)**

We will correct this in the revised version.

**18) Line 319: it might be me not being familiar with the jargon(and perhaps I missed it earlier, sorry): what are EA-WR and Eastern Atlantic pattern (perhaps need some description earlier in the manuscript?). I would generally recommend refraining from using abbreviations if not needed, it is much nicer to the reader in plain words.**

They are the circulation patterns in the mid-latitude that mainly affect the European and Eurasian domains. We will include a brief description of these patterns or at least include references for them and not use abbreviations to avoid confusion.

**19) Fig.5: how reliable are these results with regard to Fig.3?**

In the manuscript in lines 371-373 (Discussion section), we briefly discuss the representation of circulation patterns in MIROC which is different from others relating to what is observed in Fig 3. We will extend this discussion to the result of Fig. 5 to all the models in the revised version.
Also, see our response to comment 12.

**20) Fig.6: Very hard to read and capture the essence of these results. But once again, it seems to me here that the authors are showing an evaluation of the different models, so there is a tension between showing actual climatic results (fig 5) and showing how models perform and the inter-model spread, which is hard to reconcile.**

Same as the response to comment 19.

**21) Line 338: "the mean occurrence of pattern groups for each model": what is exactly meant here?**

We refer to a mean frequency (in terms of the count of patterns every century). We will rephrase the word.

**- Discussion:**
**22) On the NAO/EA-Mediterranean climate relationship: this is very interesting but I am a little puzzled to see that both NAO configurations can be associated to droughts. I am wondering if that could be discussed more?**

We consider droughts that last at least two years (lines 155). During these periods, there could be fluctuation in climate patterns, although it looks like positive NAO-like conditions occur much more frequently (patterns 1 and 3 in Fig 5). We will add more discussion on this in the revised manuscript.

**23) Another point of curiosity: can you detect in the time series (not averaged in 100-yr windows) any fingerprint of NAO (e.g., 7 yrs periodicity)? The identification of a "multi-decadal scale anti-phase" is very interesting and could also be further explored (is there an oscillatory component? Has it been observed before?). I had to think of an article by Mann et al. (Nat. Comms 2020) "Absence of internal multidecadal and interdecadal oscillations in climate model simulations", not sure if it is relevant here.**

To detect the periodicity, we need to perform extra analysis like a spectral analysis (e.g., Stevenson et al., 2018). We will perform the analysis of the time series of soil moisture anomalies and the time series of occurrence of droughts in Fig. 4 to also quantify the periodicity of the occurrence of droughts. We will be able to add more discussions to the reviewer's questions and address the "absence of internal multidecadal oscillation in climate models" with the result from the new analysis.

**24) Is there any way possible to test the observed seasonality pattern of climatic associations in the OWDA?**

We did not include the time series of OWDA, as our focus is on how some climate models represent droughts and associated extra-tropical circulation on an annual time scale. OWDA is the reconstructed time series of summer (growing season) hydroclimate. We do not think we can see similar temporal and spatial patterns to OWDA (see our response to comment 16). Regarding the spatial circulation patterns, summer circulations over the region are in general much weaker than those during the wet period, which is shown in Kim and Raible (2021) Fig 9. We can provide more details on this in the revised manuscript.

**25) Do you also observe a N-S antiphase (also mentioned in Markonis et al. 2018)?**

About the N-S antiphase in Markonis et al. (2018), in my understanding, the N-S antiphase occurs between northern Europe and southern Europe. As our study covers only the Mediterranean region (southern Europe), we would not be able to observe a similar result to Markonis et al. (2018).

**26) The E-W antiphase: is it stable on all timescales (Indeed the results you obtain are contradictory to Cook et al. 2016, this might be further discussed)?**

To answer this question, we need to perform further analysis of the time series, for instance, a wavelength coherence analysis between the west and east time series to compare their frequencies. We will add more details and discussion about the antiphase when the analysis is performed for the next phase.

**27) I am also surprised to read that climate background (e.g.,global temperatures) had no effect on droughts (or did I understand wrong?). See for instance Dermody et al. (Clim Past 2012).**

We did not particularly examine how the global temperature (or SST as used by Dermody et al., 2012) is related to the Mediterranean droughts. However, the changes in the background climate condition caused by volcanic eruptions can influence the Mediterranean hydroclimate on an annual time scale by bringing more wetter conditions (Gao et al., 2019; Kim and Raible, 2021), as we mentioned in lines 53-55. This change seems to be more dynamically driven than thermodynamically (Rao et al., 2017).

**28) Minor comment: perhaps cite Douville et al. (2021,Water cycle change IPCC report) instead of Masson-Delmotte et al. (2021)?**

We will change it in the revised version.

**29) Finally: I liked the approach of Hanel et al (Sci. Rep.2018) to distinguish meteorological (rainfall), agricultural (soil moisture) and hydrological (runoff) droughts. Perhaps an idea here to frame the research (most previous papers are looking at meteorological droughts)?**

Thanks for the suggestions. We will be clear that our focus is on soil moisture droughts in the revised manuscript.

**References**

Gao, Y., & Gao, C. (2017). European hydroclimate response to volcanic eruptions over the past nine centuries. International Journal of Climatology, 37(11), 4146-4157.

Kim, W. M., & Raible, C. C. (2021). Dynamics of the Mediterranean droughts from 850 to 2099 CE in the Community Earth System Model. Climate of the Past, 17(2), 887-911.

Liu, C., Allan, R. P., & Huffman, G. J. (2012). Co‑variation of temperature and precipitation in CMIP5 models and satellite observations. Geophysical Research Letters, 39(13).

Rao, M. P., Cook, B. I., Cook, E. R., D'Arrigo, R. D., Krusic, P. J., Anchukaitis, K. J., ... & Griffin, K. L. (2017). European and Mediterranean hydroclimate responses to tropical volcanic forcing over the last millennium. Geophysical research letters, 44(10), 5104-5112.

Stevenson, S., Overpeck, J. T., Fasullo, J., Coats, S., Parsons, L., Otto-Bliesner, B., ... & Cole, J. (2018). Climate variability, volcanic forcing, and last millennium hydroclimate extremes. Journal of Climate, 31(11), 4309-4327.

---

## Author Response (AR1)

Manuscript egusphere-2023-119

Dear Editor Dr. Goose,

We would like to thank you again for this opportunity to submit our manuscript and for dealing with it. We also would like to thank the reviewer for his/her constructive comments. We sincerely appreciate the time and effort the reviewers and the editor dedicated to our manuscript. Note that because some new results were added, the manuscript is updated, and some parts were modified (these parts are marked in the marked-up file).

In this response, we provide more detailed explanations addressing the reviewers' comments.  Our responses are in blue font, and the line numbers are within [the brackets].

The response to Reviewer 1 is on pages 2–16, and for Reviewer 2 is on pages 17 –27

Sincerely,

Woon Mi Kim, on behalf of all authors

**Response to Reviewer 1**

*Abstract:*
*0) I think the abstract is in general well written and it emphasizes the findings of the manuscript.*

Thanks very much for the comment. We'd like to inform you that the abstract is modified in the revised version. We included some more details as some results have been updated.

*GC2. Introduction:*
*I think the introduction does its job in representing the frame of the problem and the case for this study.*

Thanks very much for the comment.

*There are a couple of issues I would like the authors to consider though:*
*GC2.*
*1) Page 1, lines 21-23. 'The climate of the...' This paragraph divides the seasons into wet and dry and refers to the wet one specifically as the winter season. There is considerable spatial variability in the rainy seasons in the Mediterranean lands and for many, the wet seasons are spring and autumn. I think this winter/summer separation may be misleading. Consider for instance Xoplaki et al (2004; DOI 10.1007/s00382-004-0422-0) or others, where an extended wet season is used. I think this discussion should incorporate better this seasonality character. Note page 2, line 24 where you mention 'highly variable seasonal hydroclimate...', although the end of this paragraph (line 30) will make the emphasis again on winter.*

Thanks again for the suggestion. As the reviewer suggested, we corrected the corresponding sentences to:
[30-34] "While considerable spatial and temporal variability exists across the Mediterranean region, the climate is characterized by a wet cold season and a dry hot season (Xoplaki et al., 2004; Lionello et al., 2006). The wet cold season, typically extending from fall to spring, is a crucial period for moisture supply, where a large proportion of precipitation is provided by mid-latitude circulations and westerlies. Therefore the season dictates the intensity and impacts of droughts on the region."

*2) Page 3, lines 64-65. 'They also found that ... is model-dependent'. This seems to relate specifically to the results of the current manuscript and perhaps should be discussed specifically either here, providing more specific background and describing in which way this can happen, or/and at the end of the manuscript discussing what this manuscript provides in*
*the context of previous evidence.*

We included some more details on Xoplaki et al. (2018):
[74-80] "Using simulations from the CCSM4 and MPI-ESM climate models, Xoplaki et al. (2018) showed that multidecadal variations in the eastern Mediterranean hydroclimate are explained by internal climate dynamics. By comparing three historical periods with large

hydroclimate events over the region, they found notable differences in the climate patterns during the same periods between the two models. The observed discrepancies in climate patterns and timing of hydroclimate events between the models, and also between the models and the proxy records, indicate that exact temporal and spatial agreement of events between climate models and proxy records cannot be expected."

*GC3.*
*3). Section 2.1 Page 3, lines 83-86.*
*The CMIP and PMIP and other model efforts could include a reference. I am aware they were just provided in the previous paragraph, but it seems pertinent to me that they get those also in the Data section. I leave this to the taste of the authors.*

We included more references in the Data section, including the references for each of the models used in this study in Table 1.

*4). Section 2.1 Page 3, lines 85+ 'We consider only simulations...'*
*- There are some issues that can be considered related to the use of soil moisture in different models: The models have very different depths and therefore they will possibly produce different soil moisture statistics, even if depths only down to 0.7 m are considered. Using this set up of models is fair and possibly it contributes to inter-model differences, but also it is arguable that models with shallower depths may be less realistic. Note that some models like GISS only have a depth of 3m. This will produce potentially a different vertical distribution of moisture and different temporal variability. Perhaps it is something the authors would like to comment on.*

Thanks very much for the suggestion. We discuss more the difference in soil moisture from the models in the discussion section of the revised manuscript:

**[468-480]** "Particularly, MIROC-ESM has only half of the grids compared to bcc-csm1-1 over both the western and eastern regions and presents relatively coarse vertical soil layers (Table 1). The coarse horizontal resolutions over the land and atmosphere may affect the variables associated with soil moisture, such as precipitation, which is sensitive to the horizontal grid size (Haren et al., 2015). In addition, poor land vertical resolutions are probably insufficient to represent soil hydrology associated with vegetation and soil moisture memory effects that affect regional hydroclimate conditions (Hagemann and Stacke, 2015). This also can be the reason why MIROC-ESM shows spatial correlation patterns that differs distinctly from NOAH-LSM outside the focus region in Fig. 3.
Assessing which model represents soil moisture variability better is a complicated task since it is known that the magnitudes of soil moisture depend largely on the internal physics of land surface models (Fang et al., 2016; Berg and Sheffield, 2018). Moreover, soil moisture interactions with the atmosphere (Berg and Sheffield, 2018) and vegetation dynamics related to soil processes (Huang et al., 2016) vary across the CMIP5 models. Choosing the same vertical soil moisture level (70 cm) across all the models to represent ecosystem-related depth, i.e., root zone, over the entire Mediterranean region may be another influential factor since the root zone varies with the region (Kleidon, 2004)"

And as we responded before, a soil depth below two meters is generally considered less important for atmospheric processes. Hence, the fact that GISS has only a three-meter depth would not significantly affect the atmospheric processes we focus on.

**5) In Table 1 it is included that a depth of 0.7 m is considered. I think this is relevant and should also be included in the text. It would be desirable to include a rationale for this decision and how it may influence the timescales of variability of soil moisture. What is the typical range of depths for the bedrock, that limits the presence of water, for the area of interest in general in these models? Including a shallow limit may be more representative for the whole area but also exclude lower frequency variability more typical of deeper levels. I think including a rationale for this in the text of Section 2.1 would be good.**

We included that we used the 70 cm level and the reason for it in:
**[170-172]** "The 70 cm level is a deep soil level that can reflect the impacts of soil moisture change on vegetation and ecosystems, hence, better representing soil moisture droughts, including their persistent characteristics (Dirmeyer, 2011; Ghannam et al., 2016; Esit et al., 2021). "

Rather than the bedrock, we discuss briefly root-zone level as it is the level related to vegetation and is important for soil moisture droughts.

**[478-480]** "Choosing the same vertical soil moisture level (70 cm) across all the models to represent ecosystem-related depth, i.e., root zone, over the entire Mediterranean region may be another influential factor since the root zone varies with the region (Kleidon, 2004)"

Also, refer to our response in 4)

**6) 3. Section 2.2. Why are NOAH-LSM data used for soil moisture and ERA5 only for circulation? I mean what does NOAH-LSM offer that you would not get in this manuscript from ERA5 soil moisture?. There is probably good reasons for it. I would just suggest that a motivation for the use of these data, beyond the fact that soil moisture observations are scarce, is provided.**

As we responded during the first revision phase, we used the soil moisture from NOAH-LSM instead of ERA5 because NOAH-LSM is forced with the observation-based dataset. Therefore, we assume that NOAH-LSM could be more appropriate to show more realistic present-day soil moisture variability. We included this detail in:

**[124-126]** "We use the NOAH-LSM land variable instead of the one from ERA5, as NOAH-LSM is forced with the biases-corrected observation-based datasets. Therefore, NOAH-LSM could be a better choice that reflects more realistic present-day soil moisture variability."

**7) Line 87 '... analysis: Giss-E2-R, CCSM4...'**
 **References should be provided for these simulations. Considering previous points, I would suggest that a minimal background is at least provided in how different the soil moisture modelling can be among all models. This would help giving the reader an**

*idea of what to expect in terms of inter-model differences. For instance, vertical resolution can be very influential for soil moisture, as indicated in line 90...*

Thanks for your suggestion. We included more information in Table 1, such as the names of the land component models and the references for each model.

*8)  Line 87 '... 12 ensemble members of CESM1'*
  *There are 13 available if I am not wrong. Nothing critical but maybe you want to state why that selection. Here it can also be stated this refers to the all forcing simulations, although it is quite clear in the context (see next).*

When we retrieved the CESM-LME dataset from https://www.earthsystemgrid.org, the first ensemble member (member 001) of the variable geopotential height (Z3) was missing for 850 – 1849 CE. Hence, this member was not taken for the analysis.
We included the number of CESM-LME members  (2 to 13) we took from the data portal in

[104-105] "we use 12 ensemble members of CESM1 from members 2 to 13 available in https://www.earthsystemgrid.org."

*9) Line 91 ' All simulations were run with the volcanic, solar...' It is clearly stated earlier in the introduction that this is all about internal variability. Perhaps some comment about focussing on forced simulations and not in long control runs (also available in cmip) is pertinent.*

We primarily focus on investigating the temporal variability of droughts and the associated mid-latitude circulation patterns during the last millennium (850-2005 CE), and therefore, we use transient simulations instead of control runs. From the previous studies (e.g., Cook et al., 2017; Rao et al., 2017; Xoplaki et al., 2018), we know that droughts over the region are mainly driven by the internal climate dynamics, but we do not assume this from the beginning in our study. We were not clear about it in the former version, therefore, in the revised version, we clarified our motivation in the abstract and the introduction better, adding that the objective is to understand drought-related circulation patterns during 850-2005 CE in the coupled climate models:

[2-5] "The Mediterranean region is expected to experience significant changes in hydroclimate, reflected in increases in the duration and severity of soil moisture droughts. While numerous studies have explored Mediterranean droughts in coupled climate models under present and future scenarios, understanding droughts in past climate simulations remains relatively underexplored. Such simulations can offer insights into long-term drought variability that observational records cannot capture. Therefore, our study investigates circulation patterns in the Euro-Atlantic domain associated with multi-year soil moisture droughts over the Mediterranean region during the last millennium (850–2005 CE) in CMIP5-PMIP3 and CESM-LME simulations"

[83-85] "the objective of this study is to examine how different coupled climate models depict the temporal variability of droughts and associated circulation patterns during the last

millennium. Additionally, we assess the discrepancies among the models in representing drought-related circulations."

10)  Line 101 ' ...has four layers up (down) to two meters'
 *Ok, but I understand that only the soil moisture of the first 0.7 m is used, right? Explaining this in relation to the previous information in Section 2.1.*

Yes, from NOAH-LSM, also soil moisture at a 70 cm level was used. Here, we wanted to provide some more detail about the soil variable from NOAH-LSM. To avoid confusion, we slightly modified the sentence to:
**[117-118]** "The soil moisture variable from NOAH-LSM has four layers that extend to a depth of two meters"

And in the method section, we mention that we take soil moisture at 70 cm from all data sets.
**[130-131]** "The soil moisture content of soil layers from NOAH-LSM and the climate models are vertically integrated to 70 cm."

*GC4. 3. Methods*
 *3.1 Calculation of anomalies.*
*I find the definition of annual anomalies and the explanations for it somewhat confusing. There is an emphasis in this explanation that I do not really understand. Perhaps this is my fault but, please, consider it and see if this influences the text as it stands now.*

*11) 1. Section 3.1 Page 5, lines 111+.*
*- 'These variables are transformed into the annual mean anomalies at each grid point'. At this point I would say it is not possible to understand what is meant if additional information is not provided first, and I would argue to simply write '...transformed into annual anomalies'. I think that will be enough also for the rest of it.*
*- 'We use annual mean anomalies in order to include winter conditions in the analysis, as it is an important season for the annual hydroclimate in the Mediterranean'*
*I think it is important to discuss this in the context of what is indicated in GC2.1. Specifically for this sentence, it is not possible at this stage to understand what is meant. Not a big thing because one can perhaps understand it from the following, but:*
*a) I would suggest to include the description for the intra-annual anomalies before and then explain they are averaged; if these arguments stay in the text.*

Thanks very much for your suggestion. We included the detail you mentioned and modified the paragraph to:
**[133-137]** "The variables are all transformed into annual mean anomalies. Wet seasons are critical periods for the moisture supply of the region, and therefore, when strong circulation patterns take place (Xoplaki et al., 2004; Lionello et al., (2006). Therefore, we use annual mean anomalies to capture the mean variability of hydroclimate throughout the entire year instead of focusing on some particular seasons, i.e., summer growing seasons. This is to include the influences of wet seasons, which is the crucial season for moisture in the region, in the analysis."

**12) b) I would actually (respectfully) challenge that this is true. I think it is not, but maybe I understood things wrong. See below.**
**- 'Prior to the anomaly calculation ... to the 70 cm level'.**
**These two sentences are relevant and the reader could go with knowing this earlier in the data section (GC3.2).**

We moved the sentence earlier and modified it to:
**[130-131]** "The soil moisture content of soil layers from NOAH-LSM and the climate models are vertically integrated to 70 cm."

**13) 'Then, first, the monthly anomalies... monthly values' [...] 'Second, the necessary annual mean ... these monthly anomalies'.**
**I do not understand these two sentences. If you calculate monthly anomalies (differences between monthly data and the long-term monthly mean you are essentially subtracting the annual cycle. If you average out the resulting monthly anomalies what you get is the annual anomaly, equal to calculating anomalies from annual data by subtracting the long-term mean. Therefore, I do not understand the previous sentence about the importance of winter or the unnecessary emphasis on the monthly anomalies if you end up in annual anomalies. The sentences in the middle about reference periods read fine to me. Perhaps it fits better with the next paragraph and the issue of the trends since their definition is somewhat related.**

We agree with the reviewer's comment. We removed all the redundant sentences about calculating the annual cycles and modified the pointed part as:
**[137-138]** "The annual mean anomalies are calculated by subtracting the multi-year means from the annually averaged time series."

**14) 'Thereby, the ensemble means of the anomalies... Maher et al. (2018)'.**
**With the preceding sentences I am confused. I understand the purpose is subtracting the ensemble average to each member to get rid of the long-term trends; or better said of the forced response in general as it will also affect volcanic events for instance. This would be consistent with the following arguments in the paragraph. Consider explaining/rephrasing these paragraphs.**

As we responded in the previous phase, the method would not affect the volcanic events occurring in the pre-industrial period (850–1849 CE) as the ensemble averages were only subtracted from the historical simulations. We have revised the paragraph related to this particular methodology for further clarification:

**[141-147]** "In the next step, we remove the strong unprecedented trends in the Hist simulations (1850--2005 CE). To achieve this, we calculate the ensemble means of the annual anomalies for each climate model during the period 1850--2005~CE. Then, for each model, we subtract their corresponding ensemble mean from each of the ensemble member anomalies. The approach follows a similar method to Maher et al. (2018), which aims to exclude a trend caused by increasing anthropogenic GHG concentration from a time series. Note that the method is not applied to the LM period, therefore, the anomalies during LM still contain forced signals such as those from volcanic eruptions. The number of ensemble members can have an influence on the final output of Hist."

*15) 'This method guarantees that only internally driven variability remains in the time series of the variables'. ... mostly internally driven variability, but only in the post-1850 period. This should be perhaps indicated specifically. However, what I miss a bit is the rationale of why it is done like this. Why is the forced signal intended to be filtered out in the post-1850 (natural and anthropogenic) and not in the pre-1850 period? The reader has to guess that perhaps this is intended to get rid of the long-term trends, and the rest of the forced signal goes with it*

As we responded in the previous phase, the filtering is applied only to the post-1850 period mainly to remove the anthropogenic GHG forcing, which becomes apparent after 1850. Refer to our response in 14).

*16) Note also that the ensemble is calculated over different models in CMIP5-PMIP3 if I understood well (?). Now this is an issue that can be relevant, as different models will show different levels of response to external forcing. When this is done over an ensemble with the same model you can reduce the external response in each model run as it is assumed that the external response is common to all runs. One aspect here is that depending on the number of runs you are also subtracting also internal variability that has not been averaged out in the ensemble average, and this is a limitation that can be discussed.*

Refer to our response in 14).
The ensemble means are calculated for each of the five CMIP5-PMIP3-CESM and not over all CMIP5-PMIP3-CESM models. Therefore, each model has one ensemble mean for the period of 1850-2005. We have revised the paragraph about the method.

*17) A more relevant aspect can be that if you are using different models in the ensemble as it is the case, the assumption that a common response to external forcing is filtered out does not hold and by subtracting from each model the ensemble average you do not actually filter out the external response of that specific model. Therefore, this would work for the CESM-LME ensemble but not if you mix in the ensemble different models. This can be a sensitive issue that should be at least clearly discussed here or indicated here and better discussed later in the text.*

Refer to our responses in 14) and 16).

*18) The two periods, LM and Hist, ... models and regions.*
 *The pre-1850 and post-1850 are merged. However, they do not have the same nature, as in the pre-1850 the natural externally forced variability exists and in the post-1850 it is not intended to exist. Although if calculations have been done for some models out of a multi-model ensemble, some leftovers of externally forced variance should remain. Nevertheless, this should be in principle not important if externally forced signals are not pursued and are not expected to play a role either; actually, for that purpose, control runs could be as adequate or more adequate. I am trying to highlight with the previous sentences that some level of confusion can be transmitted to the readers.*

Thanks very much for the point. As we responded in the previous phase, we are aware that the two simulations (pre-1850 and post-1850) do not have the same nature, as they were not run seamlessly, and external forced variability is only subtracted from the historical simulations. The two periods are merged only to calculate continuously the temporal variability of droughts in Fig. 4. But note that for the pattern detection, each period (also each model and each region) was fed separately into the algorithm. We mentioned it in:

**[250-251]** "The PC-KCA procedure is performed for each model – CESM1, GISS-E2-R, CCSM4, bcc-csm1-1, and MIROC-ESM –, experiment – LM or Hist –, and region – western or eastern Mediterranean – separately."

About control runs, see our response in 9).

*19) 2. Section 3.3 Page 6.*
*I wonder why not considering the soil moisture from ERA5 or ERA5- land additionally or instead to NOAH-LSM. This relates to GC3.3. Since ERA5 produces soil moisture out of its assimilation system, this would be physically consistent with other ERA5 variables used here. Perhaps the authors want to make some comments in the text about the advantage of using the NOAH data here instead.*

Refer to our response in 6).

*20) 3. Section 3.4 Page 6, lines 161+*
*- 'When these temporal and spatial ...'*
*The regional arguments about drought incorporate the condition of having 60% of the horizontal gridpoints with negative SOIL. This faces some difficulties with the fact of using different horizontal resolutions. However, this could be overcome by considering the spatial size of anomalies by using grid box area and the total spatial extension that the anomalies represent, instead of the number of gridpoints.*

Thanks for the point. As we responded in the previous phase, we used the number of grid cells as a spatial threshold because the models have different horizontal resolutions, and it would be difficult to set the same threshold based on the spatial extension. We checked that 60% of the total grid cells correspond to about 56.20% (when a drought encompasses only the southern areas) – 61.54% (when a drought encompasses only the northern areas) of spatial extent in the western Mediterranean. For the eastern Mediterranean, the percentages are from 56.31% to 61.65%. These values are close to the 60% threshold that we used with the number of grid cells. We included the approximated values of the areal extent (56%-61%) in the revised manuscript.

**[181-183]** "At least 60% of all horizontal grid points (which is approximately from 56% to 61% of the areal extent) within the region (the west or east Mediterranean) need to be under negative SOIL conditions during all consecutive drought years."

*21) 'Next, the weighted spatial average of SOIL is ... Mediterranean'*
*How is this weighting done? Perhaps I missed it earlier in the text? If this is done, why not using this regional average to determine the occurrence of spatially large enough droughts? (??). I do not think the result will be very different from the*

*previous approach and it would overcome the use of a percentage of gridpoints with different model resolutions.*

The weighting average was performed by weighting the soil moisture values considering the area of the corresponding grid cell. We included this detail in the revised manuscript.
**[198-200]** "The time series of SOIL is generated by applying spatial weights to the soil moisture anomalies, taking into account the spatial extent of each grid cell within the confined region."

We added a spatial constraint on droughts by taking into account the percentage of grid points under negative anomalies within the region to make sure that we take droughts that have a considerable spatial extension and are regional events and not local events as explained in **[181-183].** Refer to our response in 21).

We could also have used the spatially weighted time series directly without applying a spatial threshold, but in such cases, a few strong negative anomalies in a few grid points can cause strong negative anomalies, not ensuring a minimum spatial extension of droughts.

*22) 4. Section 3.4 Page 7, lines 171+*
*'Only climate models with some ensemble members ... not detectable'*
*This is already a report of results. I have no major concern with anticipating it, but what is the reason for it?. The sentences are descriptive of the fact that pan Mediterranean droughts happen in some simulations with an specific feature and not in others. Can you at this stage argue about this feature?, or should this moved further down in the text and a discussion provided?*

We removed the sentence as we mainly focus on analyzing east and west regions and not on the pan-Mediterranean events (as we included in the former version, they are rarely detected in the last millennium when the increase in GHG is not present)

*23) 5. Section 3.5 Pages 7-9*
*I get the structure of the methodology in general, however, I suggest the authors revise this section for a more clear explanation, revising notation and perhaps the current state of the explanation of details of the methods that can be more important for the understanding of this section. I will not go to details in an exhaustive way, just provide some examples. The text is providing the sequence of a method. How the outcome of each of the steps feeds the following steps should be clear in explanation and, I would suggest, a homogeneous notation. I also recommend there is a rationale/justification for the conceptual use of each step. I will briefly try to highlight this with examples on the following, but please, go beyond those.*

Thanks for the comment. We revised and modified the entire section 3.5, and corrected the notations.

*24) References: the authors provide reference but please take care they are appropriate. For instance, there are many ways of applying PCA and the reference to the correct texts that describe the approach presented herein should be provided (see below).*

We corrected the reference. Also refer to our response in 23).

**25) Use of maps and series: This should be clear from the text. For instance, in the PCA, I understand T(t) are the principal components. Please, indicate that and also the range of the parameter t. The same with s, it is good to indicate the range so that the number of modes or the retained number of modes is well defined.**

We modified the text as
**[216-218]** "PCA decomposes a spatio-temporal field $X(t,l)$ using spatial functions $u_i(l)$, where $l$ is the spatial dimensions (*latitude x longitude* in our study), and their associated temporal functions $T_i(t)$, where $t$ is the time steps in years ($t$ is the total drought years)"

Also, refer to our response in 23).

**26) In equation (1), where are the eigenvalues?. I understand they are multiplying either the spatial or the temporal component. They should be indicated. I understand it is particularly relevant if they multiply the temporal component. The reason for it is that the temporal component will have a standard deviation 1 or different from it and this will impact the Kmeans procedure as it will affect the distance.**
**Explained variances are mentioned but it is not said that the 70% reflect accumulated variance accounted for by a number of pcs. Some sentences are confusing. E.g. line 191 'PCA is applied to the Z500 fields during droughts for each model'. I do not understand this.**

We did not include the information on eigenvalues as we found it redundant. PCA is applied prior to a k-clustering analysis (KCA), with the objective of enhancing the effectiveness of the clustering technique. But we included a reference (Hannachi et al., 2007), which explains in detail the usage of PCA in atmospheric and climate sciences. We went through and modified the entire section 3.5. Refer to our response in 23).

**27) How do the PCs enter the following analysis? They have unit standard deviation (perhaps not), and their inter-pc correlation is 0. How does this play a role in the Kmeans clustering, what does it mean physically because pcs should indicate different modes in time that are afterwards grouped, even if they are uncorrelated.**

Refer to our response in 23).
We used the PC fields with the dimension of time $t$ x truncated spatial $N$ for the k-mean clustering analysis. We explained the transition from PCA to KCA in:
**[226-228]** " The resulting new PC field of Z500 has a spatio-temporal dimension of $N \times t$, with $t$ being the total droughts.
In Step 3, KCA is applied to the $N \times t$ PC field to detect similar patterns among $t$ drought years and group them together."

**28) The notation of how the T(t) go into equation (2) should be consistent with this equation. I suggest that the notation is blended for the various steps of the analysis. If it is not done, readers will have to accommodate how things fit from their knowledge and from the different steps of the method. If notation is blended, this section would**

*actually describe one thing, the approach followed in the manuscript, not several independent methods. Some features of notation are repeated for different things, e.g., s as parameter for points in space and the Silhouette coefficient for each point.*

Refer to our response in 23).

*29) There are 71 clusters. I learned this from the text but realized that this is the sum of all clusters from different models in Table 2. For each model 5 to 6 pcs are retained and from this, between (mostly) 3 and 6 clusters are formed. What does this mean? Are the 3 clusters gathering the information of the 5-6 pcs? In which way? Some rationale/explanation for what is conceptually happening is good for the reader.*

Refer to our response in 23).

*30) Line 220: at this level I do not know what the correlation between clusters means. All in all, section 3 needs, in my opinion, to be well revised do deliver a more clear and consistent text. This does not necessarily imply changes in the calculations, nor the results of the following sections. But it may impact (positively) the interpretation.*

Refer to our response in 23).

*GC5.*
*4. Results*
*1. Related to previous comments:*
*4.1 Observation-model comparison*

*31) Line 237: '... by subtracting from each of the ensemble members the anomalies at each grid point'. This relates to previous comments and could be explained better.*

Refer to our response in 14).

*32) Line 238: 'The spatial correlations between SOIL and Z500 of NOAH-LSM and each of the climate simulations are presented in Fig. 3'.*
*I do not think it is wrong at all but I wonder what is the gain of using NOAH instead of the soil moisture from ERA5 in this figure.*

Refer to our response in 6).

*33) 2. Figure 2 and related*
*2b and 2c are good in showing the impact of using different references. However, I would say that the two are not really needed. With one of them it would be enough to explain it. Perhaps the space could be saved to accommodate a time series of the full 850-2005 period. I leave it for the authors to decide.*
*The shading in Fig 2b seems to indicate consistency in the range of variability with the range of 'observed' NOAH-LSM variability, which is good to indicate.*

Thanks very much for the suggestion. We extended Fig. 2 by including the time series for the entire 850-2005 CE, we added more details about these time series, and the NOAH-LSM – CMIP5 comparison in the revised version.

[291-295] "The ensemble means of standardized SOIL during 850--1849 CE (Fig. 2b) shows no apparent monotonic trend during LM. In general, the ensemble means of each model exhibits decreases in SOIL since 1850 CE. The decreases are noticeable even considering the ensemble spreads of each model, except for MIROC-ESM. In MIROC-ESM, a decline of SOIL is observed at the beginning of Hist, but then SOIL increases around 1950 CE. This trend of SOIL in MIROC-ESM is remarkably different from other models and may emphasize model-dependent response to forcing drivers, i.e., GHG forcing."

*34) 3. Figure 3 and related*
*Watch some statements like ' ... negative correlations over southern Europe, but the correlations outside Europe are not significant'. They are for CCSM4 and bcc in the western low latitudes of the domain.*

We corrected the sentences based to:
[314-315] "The rest of the models also show negative correlations over southern Europe. For GISS-E2-R and MIROC-ESM, the correlations are mostly significant in the European domain but not outside of the continent."

*35) Line 255, '... all models present similarities to the NOAH-LSM, fed...'*
*What we see in Fig. 3 is the result of both NOAH-LSM and ERA5, right?.*

We corrected it to NOAH-LSM-ERA5 in the text and in Fig. 3.

*36) Actually, it is likely that the large-scale structure we see there is more dependent on the global model; one could actually test if it changes with other reanalysis products. I think this is likely out of the scope of the study, but it is not totally off the line of argumentation because this correlation field with ERA5 is what we consider 'truth', but it could change if we would have used a different reanalysis product.*

Thanks for the point. As we already responded in the previous phase, we did not find much difference between different reanalysis products as the present-day reanalyses are assimilated with similar observation-based data.

*37) Perhaps a more relevant issue: if you consider the variability among patterns in Figure 3, what is the variability among simulations of one single model ensemble?. I would assume that it is smaller, but it may be worth reporting.*

As we already responded in the previous phase, there could be some variability between the ensemble members, although we do not expect that the centers of correlation would differ much among the ensemble members. Refer to the response in 38) (below).

*38) In the actual figure 3, for those models with ensembles, is the pattern that is shown the result of one single experiment? If so, I think it should be clearly stated in*

*the caption and in the text. I would advise against including correlations using ensemble averages, but I don't think this is what is being shown.*

Yes, the correlation fields in Fig. 3 are from the first member of each model. We included this information in the caption for Fig. 3, in the revised manuscript:
[Figure 3.] " For the climate models – CESM1, GISS-E2-R, CCSM4, bcc-csm1-1, and MIROC-ESM – the first ensemble member is used for the calculation."

As we already responded in the previous phase, the correlation analysis using the ensemble smooths out the fluctuations, however, the locations and signs of patterns do not significantly change.

*39) 4.2 Mediterranean drought …*
*1. Figure 4 and related*
*In relationship to the role of external forcing, or the lack of it ('This fact emphasizes that external forcing signals do not play a role in droughts over the Mediterranean ...' line 286), there are several issues that may be worth commenting.*
*- How are droughts calculated over sub-ensembles of simulations (GISS, CESM1) in Fig. 4, and how is the ensemble spread provided for them? This should be explained (sorry if I missed it) in the text and figure caption.*

Drought years are calculated for each ensemble member using the definition in Section 3.4. Fig. 4 shows the number of drought years in a moving window of a century, and this counting is performed for each ensemble member. The ensemble spread is a unit standard deviation of the time series of the number of droughts across all ensemble members. We included this detail in the caption for Fig 4.

[Figure 4] "Number of drought years in a moving window of a century in the western (red) and eastern Mediterranean (blue). Thick lines for those models and periods that have more than one ensemble member correspond to the ensemble mean, and color-shading indicates the ensemble spread represented by a unit standard deviation."

*40) Also, I expect it will justify the different temporal resolution of the curves in Fig 4 for GISS and CESM1 in comparison to the others. However, it can be misleading as the results of those two models may be read as if soil moisture for CESM1 would be consistently higher (smaller) in the western (eastern) Mediterranean during the late 17th and 18th century, or in GISS also during the 18th and early 19th centuries... or in GISS opposite to that during the 17th century. This would not be possible and would contradict the first statement of no role in external forcing as it is very unlikely that different model runs of a sub-ensemble with different initial conditions will coincide systematically in simulating relatively dry or wet periods unless forcing would play a major role. The only reason for that would be external forcing. Also, the finding of opposite phasing between west and east is interesting, and I would argue that it should be more perceptible for GISS and CESM1 if individual runs are considered, in relation to the statement.*

As we responded in the previous phase, we did not find a synchronous temporal response between the east and west across all ensemble members of the same model. Hence, the

synchronous temporal response observed in Fig. 4 in the manuscript mainly comes from averaging all ensemble members. Here, we add again the same figure that we included in our response during the initial phase, which compares the time series of the frequencies of droughts from some ensemble members in CESM1 and GISS-E2-R between the west and east regions.

[Figure]

*Fig 1. Number of drought years in a moving window of a century in the western (red) and eastern Mediterranean (blue), for some members of CESM and GISS-E2-R.*

We included some more description about this finding in the revised manuscript in:

**[354-357]** " Increased drought events in the west during mid-1500~CE are only shown in CCSM4. The same is also observed across the ensemble members of the same model. The ensemble members of CESM1 and GISS-E2-R do not exhibit unanimous periods of low or high drought occurrence (figure not shown)."

*41) '...observed, more clearly in those models and periods with one ensemble member' (line 296). Therefore, I suggest the authors revise how the ensemble behavior is presented for those two models. For instance, the last statement of Section 4.2 is sensitive: 'For those models and periods with more ensemble members... sometimes this association is blurred...', it should indeed be the expected behavior, even more than what is shown. If you resort to individual simulations, this should be more clearly evidenced. The ensemble spread should expectedly blur everything since a dry or wet century in one run should not be expected to be consistently dry or wet in most of the other sub- ensemble runs.*

Refer to our response in 40).

*42) 2. Figure 5 and related*
*I am not against showing temperature anomalies in association to the Z500 patterns, but why not showing precipitation and actual drought patterns. How do the geopotential anomalies account for drought occurrence?*

Thanks very much for the suggestion. As we do not analyze in detail the effect of temperature or precipitation on droughts, we decided to leave only the Z500 patterns in Figure 5.

*43) Figure 6: I need a better description of the methods section to better interpret results and figure out whether they can be dependent on methodological choices.*

Refer to our responses for comments 23 to 30.

*Minor comments:*

*MC1. Introduction, page 1 line 19. ' Ocean, teleconnections and large-scale modes of variabity.'  Ocean, and teleconnections with other large-scale modes of variability*
We corrected it to "high in the Atlantic Ocean and other teleconnection patterns" in **[29]** .

 *MC2. Page 2, line 40.  '...change in climate boundary conditions...'*
 *'... change in external forcing...' seems to me better in this context. The  current sentence is maybe a bias from climate modelling,*
Modified as suggested in **[51]**.

 *MC3. Page 2, line 42. 'Several natural proxy-based ...'  Natural meaning?*
We modified it to "proxy-based reconstruction" in **[53] .**

 *MC4. Page 2, line 44.  '...summer dry and wetness...'.  Dryness?*
Modified as suggested in **[55].**

*MC5. Page 2, line 42.  '...of droughts variability...' drought variability*
[Modified as suggested in **[53].**

  *MC6.  Page 5,  line 111.  '...at vertical soil layers...'  You mean perhaps, vertically integrated soil moisture content?, or something of the sort…*
We modified it to " soil moisture content of vertical soil layers" in **[130]**.

 *MC7. Page 7, line 172.  '...show few numbers of...'*
 *Cases of?*
We removed the sentence in the revised manuscript.

**Response to Reviewer 2**

*Main comments*

*1) Before I list specific points in the manuscript, I wanted to raise the issue to the authors (it is not a must, more a proposition): I personally found Fig. 2 and 3 very interesting but under- utilised (and to some extend Fig. 6 too).*

Thanks for the suggestions. We extended Fig. 2 by including the time series for the entire 850-2005 CE period and added more discussion about it **[291-295]**. We also extended the discussion on Fig. 6 (Fig. 5 in the old manuscript), which is related to Fig 7 (Fig. 6), focusing on the dominant frequencies of the detected patterns in **[424-429].**

**[291-295]** "The ensemble means of standardized SOIL during 850–1849 CE (Fig. 2b) shows no apparent monotonic trend during LM. In general, the ensemble means of each model exhibits decreases in SOIL since 1850 CE. The decreases are noticeable even considering the ensemble spreads of each model, except for MIROC-ESM. In MIROC-ESM, a decline of SOIL is observed at the beginning of Hist, but then SOIL increases around 1950 CE. This trend of SOIL in MIROC-ESM is remarkably different from other models and may emphasize model-dependent response to forcing drivers, i.e., GHG forcing."

**[424-429]** "The frequencies of occurrences of patterns presented in the lower panels of Fig. 6 indicate that the occurrence is not uniform over time, with certain patterns appearing only during specific periods. Nevertheless, the dominant frequency of each circulation pattern is estimated using a power spectra analysis (i.e., Cook et al., 2016), and the result reveals that the values noticeably vary across the models and ensemble members (Fig. A2). In general, the most dominant frequencies span a wide range of the multi-decadal (10-100 years) time scales, which partially seems to explain the time scales of the co-variability of SOIL and drought periods in Fig. 5."

*2) Clearly, some models are not very skillful at capturing the synoptic climate during historical droughts and this raises in itself an important issue: how can we confidently understand drivers using longer simulations? What setup does "work better" to capture what part of the signal (temporal vs. spatial)? I know that expending on this aspect would modify the paper substantially but it is of crucial importance to build on. It would also give more confidence to the method applied later on to determine the dominant climatic modes (I must say, as is, I am wondering how much trust we can have – which might also be due to the fact that I am not savvy with the methodology used here).*

As we responded in the previous phase, our objective is to identify extra-tropical patterns related to past droughts in CMIP5 model simulations and to understand the difference between these model simulations in depicting these patterns. For this, we compare the patterns and their frequencies (Figs 6 and 7). However, we do not particularly measure which model is the best or worst, as there is no universal objective criterion for it, and we cannot expect the same responses from all models due to their internal physical differences,

initial conditions, and model-dependent internal variability (which is what our result also shows at the end).

Still, we assume that if the models are able to represent well the present-day drought-related circulation in an acceptable way (shown in the correlation patterns in Fig. 3), the representation of the modes would still be valid in longer simulations. Therefore, we performed correlation tests between the SOIL-Z500 correlation fields in NOAH-LSM-ERA5 and other models in Fig. 3. to compare the spatial patterns of soil-related circulation. The coefficients are included in Fig.3 and discussed in **[314-325]**.

**[314-325]** "The numbers on the top of the panels indicate the correlation between the correlation patterns of NOAH-LSM–ERA5 and each of the climate models presented in Fig. 3. The correlation analysis is performed on the entire regions in Fig. 3, which means that these values measure the overall closeness of the entire correlation fields to NOAH-LSM–ERA5 without considering the statistical significance of individual grid locations. We use this quantity to compare the model's representation of SOIL-related Z500 patterns between the two regions, the western and eastern Mediterranean. Overall, the correlation coefficients are higher in the western region than in the eastern Mediterranean. In the western region, the maximum coefficient is 0.82, shown by bcc-csm1-1, while the maximum in the eastern region is 0.62, also by the same model. The minimum value in the western region is 0.42 by GISS-E2-R, and in the eastern region, it is 0.14 by CCSM4. The overall comparison of spatial correlation coefficients implies that the variability of Z500 associated with SOIL in the climate models is better represented over the western region than over the eastern region."

For temporal variability, we assess whether the ranges and ensemble spreads of soil moisture anomalies from the models encompass the variability of the present-day observation-based soil moisture anomalies (NOAH-LSM), shown in Fig. 2a, and discussed in **[282-290].**

**[282-290]** " During 1950–2005 CE (Fig. 2a), the variability of SOIL from NOAH-LSM (the range of maximum and minimum over 1950–2005 CE is 18.29 mm/month) is within the range of variability of the CMIP5-PMIP3 model SOIL values (the range of maximum and minimum across the models of 22.75 mm/month). This observation indicates that the overall magnitudes of variability between the observation-based data and the models are comparable. However, the standard deviation ($\sigma$) of SOIL across the four models over the entire time (3.06 mm/month) is lesser than the $\sigma$ of SOIL of NOAH-LSM (4.37 mm/month). This distinct $\sigma$ indicates that there is some degree of discrepancies in SOIL variability among the models. GISS-E2-R, CESM1, and CCSM4 show $\sigma$ of SOIL of 7.73, 5.44, and 4.43 mm/month, respectively, which are higher than that of NOAH-LSM, while bcc-csm1-1 and MIROC-ESM presents $\sigma$ of 2.03, and 2.46 mm/month, respectively"

***3) Another (more minor) issue: Perhaps I am not familiar with the term, but it would be useful to clarify what you mean by "internal variability" (and that might be done by just explaining what is considered an external forcing in your study). If not used to the terminology, one might wonder for instance, whether teleconnections are part of the internal variability or not?***

With internal variability, we refer to different modes of climate variability that arise from the interaction of climate systems without the influence of external or internal forcings (volcanic eruptions). We rephrased it to "internal climate dynamics" when we want to distinguish internal variability (without eruptions) from external climate signals. For instance,

[352-360] "This large variability in drought occurrence across the models and the ensemble members, depicted by the ensemble spreads in Figs 4 and 2b, implies that internal climate dynamics is the primary driver of droughts in the region during LM and Hist with the anthropogenic forcing removed. In these time series, external or volcanic forcing signals are not visible in the variability of droughts."

*4) Abstract: lines 4-5: "The focus is [...] during 850-2005, this excluding the anthropogenic trends from 1850 CE onwards". This is very confusing... It can only be understood if the methodology has been explained, otherwise one starts to wonder which time range was actually used.*

We modified the sentence to:
[7-9] "Primarily, we examine the differences among the models in representing drought variability and related circulation patterns. For the analysis, we exclude the anthropogenic trends from 1850–2005 CE."

Also note that the abstract is modified in the revised manuscript, as the results are updated.

*5) Introduction: Lines 66-71, the definition of the knowledge gap and the leading questions is a little weak. "necessary to understand which modes of climate variability or atmospheric circulation patterns are involved in each climate model": this sounds very ambitious and is actually not what this study is tackling. Instead, it might be useful to explain why studying the "patterns in the mid-latitudes" and their effect on Mediterranean climates is important.*

Thanks very much for the suggestion. We modified the motivation in the introduction as:
[83-86] " the objective of this study is to examine how different coupled climate models depict the temporal variability of droughts and associated circulation patterns during the last millennium. Additionally, we evaluate the differences among the models in representing drought-related circulations. The focus is on the mid-latitude circulation patterns that have more impacts on the Mediterranean hydroclimate."

*6) S3.1: this section of the methods is at times quite confusing. I have read the comments of Rev. 1 and agree with their comments so I will just emphasize two points that I found particularly confusing:*
*Line 112: "We use annual mean anomalies in order to include winter conditions in the analyses [...]": I don't understand this sentence.*

We corrected the sentence to:
[133-137] "The variables are all transformed into annual mean anomalies. Wet seasons are critical periods for the moisture supply of the region, and therefore, when strong circulation patterns take place (Xoplaki et al., 2004; Lionello et al., (2006). Therefore, we use annual mean anomalies to capture the mean variability of hydroclimate throughout the entire year

instead of focusing on some particular seasons, i.e., summer growing seasons. This is to include the influences of wet seasons, which is the crucial season for moisture in the region, in the analysis."

*7) Lines 121-126: This part is at the same time important and confusing: why not normalising the whole record instead of just the Hist part? Would a z-scoring be useful?*

The aim of using the method in line 121 is to remove the trends introduced by the anthropogenic increase in GHG after 1850 CE, as this trend can affect the overall analysis of the time series (continuous droughts are found after 1850-1900 when the trends is not removed.). Therefore, the GHG effect is not included in the analysis. We added this detail in the revised manuscript:

**[140-141]** "In the next step, we remove the strong unprecedented trends in the Hist simulations (1850–2005 CE). This means that the effects of increased GHG on droughts is not included in the analysis."

*8) S3.5: section very hard to follow, please provide additional information and background on the choices of the methods and their limitation.*

We revised the entire section 3.5, and the section is now modified from the previous version.

*- Results*
*9) Fig.2: if I understand right, this is your control run for time series.*

All time series are from the forced (transient) last millennium and historical simulations. We included this detail in the caption of Fig 2.
**[Figure 2]** "10-year running means (thick lines) and ensemble spreads (color-shaded) of standardized SOIL during 850–2005 CE with respect to 850–1849 CE, from the PMIP3-CESM1 transient LM and Hist simulations."

*10) a) the modelled soil data are quite different from the observation time series (range and trend are similar but changes are often not synchronous). How does that affect your analysis? What does the intermodal comparison look like? Could you have a look at the cross-correlation between NOAH LSM and CMIP5 mean? Just so that we understand the limitations of the study.*

This question is partially responded in 2). Also, as we responded in the initial response phase, we do not expect the models' soil data to be exactly synchronous with the observation-based time series. That is basically not possible because of the difference in the initial conditions between the models and the observation, and internal variability of the climate system. As long as the variability of the observed variable lies within a model spread (an ensemble spread given by a unit standard deviation or maximum-minimum value range across the models), we assume that the model is able to represent the temporal variability of the observations. For the same reason, a cross-correlation analysis between NOAH-LSM and CMIP5 would not provide additional useful information. We included discussions about the implication of differences in soil data in:

**[459-467]** "It needs to be pointed out that MIROC-ESM presents many unique patterns that fail to join other circulation patterns, largely occurring with the eastern Mediterranean patterns. Several reasons can be put forth to explain this particular characteristic of MIROC-ESM. As depicted in Fig. 2, MIROC-ESM presents a different SOIL trend compared to the other models during Hist. This distinct trend could potentially be attributed to different circulation types compared to those of other models.

Another argument to consider is that, in general, SOIL-related circulation fields in the eastern region seem to exhibit lower statistical similarity with the observation-based circulation condition, based on correlation coefficients in 3b. While this is the case for all models, lower correlation coefficients in the eastern regions can be associated with a reduced number of horizontal grid points compared to the western regions. Fewer grid points may not reflect the entire temporal variability of SOIL, and, therefore, the associated circulation variability."

*11) Line 238: "SOIL and Z500 from NOAH-LSM" isn't Z500 from ERA?*

Yes, we modified it to NOAH-LSM–ERA5 in the text.

*12) Fig.3: control run for spatial analyses. I think that it would be good to indicate that the upper left panel shows the observation (SOIL vs. Z500). How do you deal with models not capturing the synoptic climate very well (are they still used for further analyses, and if so, why?)? Can you quantify the skillfulness of the models? To me this exercise seems to be providing an evaluation of the models to track spatiotemporal climate conditions associated to droughts. It could be a goal in itself (as proposed earlier).*

We did not have control simulations in our analysis (see our response in 9).

As we responded in the initial phase, regarding the question of how to deal with models that do not capture well the observed synoptic climate, one of the objectives of our study is to understand the model differences in representing drought-related circulation over the Mediterranean region. Therefore, we do not apply a selection criterion to choose the best or worst model (Also refer to our response to comment 1). Instead, we attempt to address and visualize the differences between the models in representing drought-associated mid-latitude circulation patterns (For instance, in section 4.2). Regarding the skillfulness of the models, for the same reason, it would be hard to assess the skillfulness of all variables associated with droughts, and there is no universal metric for it.

Still, in the revised version, we performed a comparison of SOIL-related circulation (Z500) patterns between the models and NOAH-LSM–ERA5 by calculating the correlation coefficients of the correlation fields in Fig. 3. These coefficients measure the similarity of the correlation fields between the models and NOAH-LSM–ERA5, and we used them mainly to address the differences between the east and west regions. We included these new values in Fig. 3 top left of the panels, the explanation of the methods **[163-167]** , and the discussion of the result **[314-325].** Also, refer to our response in 2).

**[163-167]** "In addition, to quantify the spatial similarity of the correlation patterns, PCC is calculated between the correlation pattern  NOAH-LSM--ERA5, for the western and eastern regions, separately, and the patterns of each climate model. For this spatial comparison of

correlation patterns, the spatial resolutions are interpolated to those of the coarser models, which are bcc-csm1-1 and MIROC-ESM (Table 1). This PCC would provide a numerical value to evaluate the overall resemblance between the present-day NOAH-LSM–ERA5 field and the climate models."

**13) Lines 286-288: I struggle with these sentences. Either too little or too much is said here. What is the role of model's setup and skilfulness in this observation?**

For clarification, we modified the part to:
[357-360] "This large variability in drought occurrence across the models and the ensemble members, depicted by the ensemble spreads in Figs. 4 and 2b, implies that internal climate dynamics is the primary driver of droughts in the region during LM and Hist with the anthropogenic forcing removed. In these time series, external or volcanic forcing signals are not visible in the variability of droughts."

About the models' setup and ability, refer to our response in 1), and their skillfulness, to our response in 12).

**14) I also do not understand what is meant by "counterfactual"?**

We excluded this word to avoid confusion and rephrased the sentence to:
[359] "Hist with the anthropogenic forcing removed."

**15) Fig.4: Would it be possible to quantify the antiphase? (bi-plot, cross- correlation)**

Thanks for the suggestion. We included wavelet coherence analysis to assess the phase relationship between the two regions and their variations over time, in Fig 5.

[202-204] "We perform wavelet coherence (Grinsted et al., 2004) on the time series of the number of droughts and the soil moisture anomalies between the west and east Mediterranean. The purpose is to assess the phase relationship, dominant frequencies, and temporal variations of correlations in soil moisture anomalies and drought occurrences between these two regions."

The discussion of the result can be found in**:**
[372-390] "To evaluate closely the dominant frequency of the association between the two regions, and their temporal co-variability, the wavelet coherence analysis is performed on the time series of SOIL (Fig. 2b) and presented in Fig. 5a. At first glance, the wavelet coherence analysis suggests that timing and frequencies of co-variability are not the same across the models for both the soil moisture anomalies and the number of droughts. Also, the association is not uniform across the time-frequency space. The analysis performed on SOIL between the western and eastern regions (Fig. 5a) indicates co-variability that ranges from interannual to multi-decadal time scales, depending on the model. CCSM4 shows co-variability of higher frequencies (less than a 32-year period.). In general, the association between the two regions is in-phase. The co-variability between the western and eastern regions from all models is less pronounced and less significant in all time-frequency bands compared to the result presented by Cook et al. (2016) based on OWDA, which has shown significant in-phase co-variability of SOIL between east and west in diverse timescales.

The time series of SOIL does not necessarily indicate that dry periods are in phase. To compare only the dry periods, the wavelet coherence analysis is performed on the number of drought years (Fig. 4), which is presented in Fig. 5b. The co-variability between the two regions is significant in some time scales, for instance, on a multi-decadal time scale (around 32 years and higher) with an anti-phase relationship in CESM1, bcc-csm1-1, and MIROC-ESM. For GISS-E2-R, CCSM4, and MIROC-ESM, the anti-phase association is also significant on a high-frequency band (of less than 32 years). This result seems to agree with the observation from Fig. 4. It is observed that the anti-phase co-variability also depends on the time period, for instance, CCSM4 during 1400-1500 CE, the co-variability is in-phase. Again, the overall result points out the associations that are not uniform across time and that vary across the models. The occurrence of droughts and dry periods can be associated with dominant drought-driving circulation patterns of each region. More details on this are provided in the next sections."

**16) How do these time series compare to actual data (e.g., OWDA, HYDRO2K), even if the variables are different, there should still be some similarities?**

As we responded in the previous phase, regarding actual data (for instance, OWDA), Kim and Raible (2021) have performed the comparison between summer droughts in a CESM simulation and OWDA, and found differences in the temporal occurrence of droughts between the model and the proxy-based reconstruction. If we include OWDA, we expect to see a similar result here (but not temporarily synchronous). Also, OWDA provides only the summer hydroclimate variability of the region and not all annual means as we used.

**17) A minor issue but might avoid confusion: I would recommend using years CE/AD on the scale (I am more often dealing with yrs BP, so I tend to read that directly... It's a matter of community standards)**

We included CE after the years.

**18) Line 319: it might be me not being familiar with the jargon(and perhaps I missed it earlier, sorry): what are EA-WR and Eastern Atlantic pattern (perhaps need some description earlier in the manuscript?). I would generally recommend refraining from using abbreviations if not needed, it is much nicer to the reader in plain words.**

We corrected the text and wrote the full name of the patterns:

**[151-155]** "The separation is motivated by the suggested influences of the circulation patterns over the Mediterranean region (Dunkeloh et al., 2003; Lionello et al., 2006), such as the NAO, East Atlantic (EA), and Eastern Atlantic-West Russian (EA-WR). The western Mediterranean is more intensely influenced by NAO than the eastern region, and the eastern region is not only affected by NAO but also strongly linked with East Atlantic-type patterns."

We used abbreviations after this paragraph.

**19) Fig.5: how reliable are these results with regard to Fig.3?**

In the discussion section in lines **[459-467]** (refer to our response to the comment in 10), we discuss the representation of circulation patterns, mostly relating it to MIROC-ESM which is the model that shows distinct patterns from others in Fig 3 and Fig 5. We discuss the reasons for this possible difference (which is also valid for other models) based on different horizontal resolutions and soil moisture processes in the models.

**[463-467]** "Another argument to consider is that, in general, SOIL-related circulation fields in the eastern region seem to exhibit lower statistical similarity with the observation-based circulation condition, based on correlation coefficients in Fig. 3b. While this is the case for all models, lower correlation coefficients in the eastern regions can be associated with a reduced number of horizontal grid points compared to the western regions. Fewer grid points may not reflect the entire temporal variability of SOIL, and, therefore, the associated circulation variability."

*20) Fig.6: Very hard to read and capture the essence of these results. But once again, it seems to me here that the authors are showing an evaluation of the different models, so there is a tension between showing actual climatic results (fig 5) and showing how models perform and the inter-model spread, which is hard to reconcile.*

Thanks for your comments. In Fig. 7 (Fig 6 in the former version), we try to summarize the frequencies of the patterns in Fig 6 (Fig 5 before), and show the difference in frequencies of these patterns among the models, in which some patterns only appear in one model. We discuss briefly it in:

**[505-509]** "Our analysis also shows that the contribution of the patterns to droughts may greatly depend on the choice of the model. For example, the importance of EA-WR in the eastern droughts is apparent in CCSM4 and bcc-csm1-1 but not in other models (Fig. 7). The frequencies of a shared pattern between some models also vary greatly among them. This highlights the fact that climate models have their preferred circulation patterns associated with Mediterranean droughts."

The inter-model spread of these patterns in terms of their temporal variability is shown by the time series in Fig 6 (old Fig 5), explained in the caption:

**[Figure 6]** "Below each map, the time series of the number of occurrences of the corresponding patterns per century for the western (red) and eastern (blue) Mediterranean are plotted, together with the respective ensemble spread of occurrence (shaded)"

*21) Line 338: "the mean occurrence of pattern groups for each model": what is exactly meant here?*

We refer to the mean frequency during LM and Hist of the circulation patterns. We rephrased the sentence to:
**[430-431]** "Fig. 7 summarizes the mean frequencies of the circulation patterns during the entire LM and Hist in Fig. 6, represented by the mean number of patterns per century for each mode"

*- Discussion:*

**22) On the NAO/EA-Mediterranean climate relationship: this is very interesting but I am a little puzzled to see that both NAO configurations can be associated to droughts. I am wondering if that could be discussed more?**

We consider droughts that last at least two years, therefore, the events are of multi-year duration. During these periods, there could be fluctuation in climate patterns, although it looks like positive NAO-like conditions occur much more frequently (patterns 1 and 3 in Fig 6). We extended this explanation in the revised manuscript.

**[403-406]** "Positive NAO is known to be the dominant climate mode that drives a drier condition over the Mediterranean region (Lionello et al., 2006; Kim and Raible, 2021). This explains a large percentage of occurrence of P1 and P3 (42%) during droughts. Although it seems contradictory that P2 depicting a negative NAO condition also occupies a significant percentage of the occurrence (20%), the occurrence of P2 indicates the fluctuation of NAO patterns throughout multi-year droughts."

**23) Another point of curiosity: can you detect in the time series (not averaged in 100-yr windows) any fingerprint of NAO (e.g., 7 yrs periodicity)? The identification of a "multi-decadal scale anti-phase" is very interesting and could also be further explored (is there an oscillatory component? Has it been observed before?). I had to think of an article by Mann et al. (Nat. Comms 2020) "Absence of internal multidecadal and interdecadal oscillations in climate model simulations", not sure if it is relevant here.**

The periodicity (mainly of co-variability between the two regions) from wavelet coherence analysis is provided in Fig 5. and lines **[372-390]**. The result shows no uniform and stable temporal frequency of co-variability. This may support the ""absence of internal multidecadal oscillation in climate models" of Mann et al., (2020).
Also, refer to our response in 15).

Additionally, we performed power spectra density analysis to find frequencies of the patterns in Fig 6, but we did not find consistent frequencies among the models (Fig. A2). Nevertheless, the patterns have frequencies of occurrence on multi-decadal time scales that are consistent with Fig. 4. We included the discussion in:

**[424-429]** "The frequencies of occurrences of patterns presented in the lower panels of Fig. 6 indicate that the occurrence is not uniform over time, with certain patterns appearing only during specific periods. Nevertheless, the dominant frequency of each circulation pattern is estimated using a power spectra analysis (i.e., Cook et al., 2016), and the result reveals that the values noticeably vary across the models and ensemble members (Fig. A2). In general, the most dominant frequencies span a wide range of the multi-decadal (10-100 years) time scales, which partially seems to explain the time scales of the co-variability of SOIL and drought periods in Fig. 5."

**24) Is there any way possible to test the observed seasonality pattern of climatic associations in the OWDA?**

Refer to our response in 16).

As we responded before, we did not include the time series of OWDA, as our focus is on the representation of drought variability and associated mid-latitude patterns in climate models. OWDA is the reconstructed time series of only summer (growing season) hydroclimate. We do not think we can see similar temporal and spatial patterns to OWDA.

**25) Do you also observe a N-S antiphase (also mentioned in Markonis et al. 2018)?**

As we responded in the previous phase, the N-S antiphase occurs between northern Europe and southern Europe (Markonis et al., 2018). As our study covers only the Mediterranean region (southern Europe), we would not be able to observe a similar result to Markonis et al. (2018).

**26) The E-W antiphase: is it stable on all timescales (Indeed the results you obtain are contradictory to Cook et al. 2016, this might be further discussed)?**

Refer to our response in 15).

**27) I am also surprised to read that climate background (e.g.,global temperatures) had no effect on droughts (or did I understand wrong?). See for instance Dermody et al. (Clim Past 2012).**

As we responded during the initial phase, we did not particularly examine how the global temperature (or SST as used by Dermody et al., 2012) is related to the Mediterranean droughts. However, the changes in the background climate condition caused by volcanic eruptions can influence the Mediterranean hydroclimate on an annual time scale by bringing more wetter conditions (Gao et al., 2019; Kim and Raible, 2021), as we mentioned in **[64-66].** This change seems to be more dynamically driven than thermodynamically (Rao et al., 2017).

**28) Minor comment: perhaps cite Douville et al. (2021,Water cycle change IPCC report) instead of Masson-Delmotte et al. (2021)?**

We changed it in the revised version.
**[338-339]** "a change that is mostly attributed to anthropogenic effects on global and regional climate (Douville et al., 2021; Seneviratne et al., 2021)."

**29) Finally: I liked the approach of Hanel et al (Sci. Rep.2018) to distinguish meteorological (rainfall), agricultural (soil moisture) and hydrological (runoff) droughts. Perhaps an idea here to frame the research (most previous papers are looking at meteorological droughts)?**

Thanks for the suggestions. Our focus is on soil moisture droughts, therefore taking the soil moisture anomalies from the subsurface layers to quantify droughts. We clarified it in the revised manuscript.

**[100-101]** "Our focus is on droughts that affect deep soil moisture conditions, known as soil moisture droughts (Dai, 2011)."

**[170-171]** " The 70 cm level is a deep soil level that can reflect the impacts of soil moisture change on vegetation and ecosystems, hence, better-representing soil moisture droughts including their persistent characteristics."

**References**

Gao, Y., & Gao, C. (2017). European hydroclimate response to volcanic eruptions over the past nine centuries. International Journal of Climatology, 37(11), 4146-4157.

Kim, W. M., & Raible, C. C. (2021). Dynamics of the Mediterranean droughts from 850 to 2099 CE in the Community Earth System Model. Climate of the Past, 17(2), 887-911.

Rao, M. P., Cook, B. I., Cook, E. R., D'Arrigo, R. D., Krusic, P. J., Anchukaitis, K. J., ... & Griffin, K. L. (2017). European and Mediterranean hydroclimate responses to tropical volcanic forcing over the last millennium. Geophysical research letters, 44(10), 5104-5112.

---

## Referee Report (RR1)

2nd Review of :
egusphere-2023-119
Extratropical circulation associated with Mediterranean droughts during the Last Millennium in CMIP5 simulations
W. M. Kim, S. J. González-Rojí and C. Raible

Evaluation:
    The authors have included responses and changes to the comments. I am satisfied with most of them. I am not convinced about a couple of them that I mark with (*). I do suggest improving the description of the methodology. Additional to that there are several points which I think should be considered. I regard them as minor and easy to do, so I leave it to the Editor whether I should see the responses and the corrected manuscript again.

GC1.  Introduction:
    I think the introduction has improved with the changes implemented. A minor comment:

1.  Page 2, line 51. '… no substantial change in external forcing…'
This statement can be missleading. I understand in the context of what the authors want to convey but I suggest rephrasing. Forcing changes are substantially smaller than say the glacial-interglacial transition, but still detectable in reconstructions and substantial enough to be climatically relevant. Consider rephrasing.
I think the relevant issue here is that the LM is the period in which the (natural) forcings are most comparable to nowadays and the one, if we think preindustrial, that offers the reference before the development of the large anthropogenic forcing.

2.  As a reference for drought variability at global scales during the last millennium and potential responses to internal variability and external forcing consider https://cp.copernicus.org/articles/16/1285/2020/

GC2.  Section 2: Data

1.  Page 4, line 99: (LM; 850-1850)
See consistency with pate 5 line 140: 850-1849

Page 4, line 109: '…are transient  and were run…'

GC3.  Section 3: Methods

1.  Line 142: '… the effects of GHG on drought  are not…'

Line 167: '… present-day NOAH-LSM-ERA5 correlation field and that obtained from the climate models …' (?)

Line 175+ Drought definition. It is not clear if the drought is composed only of negative SOIL values (as it is literally stated) or if it contains the dates of the two positive values at the end ('… and continue until two consecutive years of positive anomalies…'
'.. of only negative values without being interrupted by a particularly wet year in between…' However, I understand from the definition that if there is a positive SOIL anomaly, the drought continues until two positive anomalies are found' – I am only trying to point out that I find the explanation confusing in this respect

Line 183 '… a substantial portion of the region experience_s_ drought conditions …'

Line 186. 'Nevertheless, this approach avoids changes in the initial SOIL values, as …' I do not understand what is meant here.

Line 198+ 'Also, the time series of SOIL is generated by applying spatial weights to the soil moisture anomalies, taking into account the spatial extent of each grid cell within the confined region'.
This is not clear to me and I think it should be explained, what is done, how is it taken into account. It is pointed out that it is somehow taken into consideration, but I suggest to indicate how.

2. (*) Line 214+, PCA calculation
   - Line 222: '…PCs, represented as …Ti(t) * ui(l)
     In my understanding this is not correct. Please, indicate which variable are the PCs and which the eigenvectors, also indicate where the eigenvalues are included in eq (1).
     I think this part of the methods is not satisfactory. The explanation should state how the covariance (or correlation?) that is diagonalized is defined (in space or time) and then which ones are PCs, eigenvectors, EOFs, eigenvalues, etc, in a clear way. I think this influences the interpretation of the results and helps reproducibility.
   - Why 70%? Likely arbitrary but I would expect the text to say that the results are not very sensitive to this decision. Why should N be < or = to 7?...
   - I understand that the dates corresponding to the identified droughts are selected and the Z500 is considered only for those dates and then the PCA is applied. Perhaps I have skipped this explanation somewhere, but I have not found it. It should be clearly explained.
   - Do the xi in Eq (2) refer to the X(t,l) in eq (1)? This does not make sense to me. Can you refer the notation of Eq (2) to that of Eq (1) so that the reader can understand how the output of the PCA feeds the KCA? In my understanding these should be PC values because you end with N modes X t drought years, with ( I assume) a PC being a time series with t time steps including drought years.
   - Therefore, each value in a cluster could be an array of N values corresponding to how a given date (Z500 anomaly map) is represented by those N values in the space of EOFs. However this

does not fit Eq (2). I may be wrong though and other approaches may be possible. What I am trying to highlight is that 3.5 needs a clear explanation of the methodological approach and its parts, with the notation of the different parts being consistent with each other.

- The final number of clusters is 71. However, I understand that since the analysis is performed on each model, many of those will be similar. Perhaps worth commenting this here?

- There is quite a number of typos, please revise them. Please, also in the rest of the text. I will avoid pointing at the grammar issues, but please take care of this.

GC4. (*) Section 4.1

I agree with the comments about Fig 2 and Fig 3 in general if we consider the details, maps, correlations, etc. However, perhaps I would have a different take on the actual interpretation of them.
Consider Fig. 2 first. Most models have a large low frequency variability, with large multi-decadal or multi-centennial departures from the long term mean. Some of them often longer than the reference period considered in Fig 2. They are to a large extent not consistent among the different models, which therefore indicates that they are more obviously related to internal variability than to the external forcing common to all experiments. If this is the behaviour of a real SOIL variable then the 1959-1979 interval considered as a reference is a very short interval of time and may correspond to a very specific state in the NOAH model, assuming that it also represents reasonably reality. However, a longer integration with NOAH or if we had more observations, would supposedly show a considerable level of low frequency variability; we do not know how much because we do not know how well the models in Fig2b represent reality. With this in mind, the fact that one simulation represents less variability during the selected period or another one represents more, does not mean that this or that model is doing better or worse, because it is not intended that these simulations represent the real 1950-79 variability, unless it would be clearly responding to external forcing or those simulations were driven by observations, which are not.
The previous reasoning extends to the other arguments related to Fig3. All arguments oriented to a better or worse representation (e.g. line 324, 325), I wouldn't agree with them, because Fig 2, indicates that except for the trends in the last decades SOIL responds to internal variability and thus the maps in Fig 3 are expected to show some level of similarity but not to represent faithfully what the NOAH model does within that comparably short period.
I think the authors should consider this argument and see how it impacts the orientation of the text and the interpretation of the figure.

Specific details:
- It would be good to indicate in the text or in the figure caption the time interval used for correlations.

- The maps shown in Fig 3 are indicating some relation to zonal circulation, NAO, which is mentioned in the text. If desired, this could be objectively calculated by indicating correlations with the NAO index in each model. But I would understand also that the authors would not want to go in that direction.

GC5. Section 4.2

1. Figure 4 caption and in the text. The mean percentages of total drought years and the mean duration of droughts are calculated from the ensemble means…
I have reservations about the meaning of these numbers because of being calculated from the ensemble means. If the quantity that was being analysed would depend mostly on external forcing, I would agree, because the metrics based on the ensemble average would be meaningful as a filtered version obtaining after cancelling noise from internal variability. However, the behaviour of soil and drought occurrence here is shown and argued (e.g. Line 358) to be related to internal variability. Therefore, the statistics that would be comparable to the real world or representative for it according to each model are those of individual simulations, not the ensemble average.
Arguing from a different angle: if we would have enough of a high number of runs, the ensemble average should tend to be flat and with no droughts.
The previous arguments also would justify why this happens (L 368): '… more clearly in those models and periods with one ensemble member'

2. Line 371: ' This seems to agree with Cook et al (2016)'
Do you mean the simultaneous occurrence or the period or both? Please indicate more specifically and if you think it is important elaborate…

The comparison with Cook et al (2016) is interesting. For instance also in Lines 378-381.
Do you think this comparison holds even if the reconstructions of Cook et al basically address the growing season (jja)? How can the statements of agreement or disagreement suffer from this? Some comments about this would be pertinent.

GC6. Line 405: 'Although it seems contradictory that P2 depicting a negative NAO condition also occupies a significant percentage of the occurrence…'

I think this could perhaps be due to the size and definition of the windows used. I do not mean that it is wrong but could be an effect of that and if so, it may be worth commenting on it. The P2 pattern favours inflow from the SW into the Iberian Peninsula. The western side of it, over Portugal and southwestern Spain should not be dry with this pattern. However, dryness could affect the lands of northern Africa and central

Mediterranean Islands and over Italy. I think it is likely that the occurrence of drought with this pattern in the western box reflects the balance of wetness in the west/northwest region of the box and dryness in the rest. Perhaps it is worth assessing that and commenting.

This also takes me to suggest that it would be interesting to see the composites of soil for each group of patterns. One could actually show the composites over the whole Mediterranean, not only the boxes. This does not necessarily require an increase in the number of figures. The Z500 anomalies can be shown with lines using hatching for significance and the soil pattern with shading in the same map. It would help to understand how the different patterns influence drought in the region of interest.

---

## Author Response (AR2)

Dear editor Dr. Goosse and the reviewer,

We would like to thank the editor and the reviewer again for dealing with our manuscript and for his/her constructive comments. We sincerely appreciate the time and effort the reviewer dedicated to reviewing our manuscript. Here, we provide our responses to the reviewer's comments.
Our responses are in blue font, and the line numbers are within [the brackets].

Response to the reviewer:

Evaluation:
The authors have included responses and changes to the comments. I am satisfied with most of them. I am not convinced about a couple of them that I mark with (*). I do suggest improving the description of the methodology. Additional to that there are several points which I think should be considered. I regard them as minor and easy to do, so I leave it to the Editor whether I should see the responses and the corrected manuscript again.

**GC1. Introduction:**

I think the introduction has improved with the changes implemented. A minor comment:

1) 1. Page 2, line 51. '... no substantial change in external forcing...'
This statement can be missleading. I understand in the context of what the authors want to convey but I suggest rephrasing. Forcing changes are substantially smaller than say the glacial-interglacial transition, but still detectable in reconstructions and substantial enough to be climatically relevant. Consider rephrasing. I think the relevant issue here is that the LM is the period in which the (natural) forcings are most comparable to nowadays and the one, if we think preindustrial, that offers the reference before the development of the large anthropogenic forcing.

Thanks very much for the point. We modified the sentence to:
[51-52]   "the last millennium is particularly interesting as it is a relatively close period with similar natural forcing conditions to those during the pre-industrial era"

2) 2. As a reference for drought variability at global scales during the last millennium and potential responses to internal variability and external forcing consider https://cp.copernicus.org/articles/16/1285/2020/.

Thanks for the suggestion. We added in the introduction
[71-75] "the impact of volcanic eruptions on hydroclimate-related variables and long-lasting droughts has been assessed on a global scale (Stevenson et al., 2017; Roldán-Gómez et al., 2020) and the Mediterranean region (Kim and Raible, 2021) using the climate simulations from the Community Earth System Model (CESM; Lehner et al., 2015; Otto-Bliesner et al., 2016) and the fifth phase of the Climate Model Intercomparison Project (CMIP5; Taylor et al., 2012) - Paleoclimate Model Intercomparison Project Phase 3 (PMIP3; Schmidt et al., 2012)."

[78-79] "On a global scale, Roldán-Gómez et al., 2020 indicated that there is not a clear relationship between soil moisture and external forcing that can be detected during the last millennium in the CMIP5-PMIP3 simulations.

GC2. Section2:Data

3) 1. Page 4, line 99: (LM; 850-1850). See consistency with pate 5 line 140: 850-1849
We changed it to 850–1849, also in other parts of the text. [103]

4) Page 4, line 109: '...are transient  and were run...'
We changed it as suggested. [113]

GC3. Section3:Methods

5) 1. Line 142: '... the effects of GHG on drought  are not...'
We changed the sentence to:
[144-145] "This means the effects of increased GHG on droughts are not included in the analysis."

6) Line 167: '... present-day NOAH-LSM-ERA5 correlation field and that obtained from the climate models ...' (?)

We modified the sentences for clarification:
[167-171] "In addition, to quantify the spatial similarity of the correlation fields obtained after the PCC between the observation-based data (NOAH-LSM--ERA5) and the climate models, the pattern correlation is calculated between the two datasets for the western and eastern Mediterranean regions. For this calculation, the horizontal resolutions of all data are interpolated to match those of the coarser climate models, which are bcc-csm1-1 and MIROC-ESM (Table 1). The pattern correlations assess the overall resemblance between the correlation field of NOAH-LSM--ERA5 and the climate models for the present day. "

7) Line 175+ Drought definition. It is not clear if the drought is composed only of negative SOIL values (as it is literally stated) or if it contains the dates of the two positive values at the end ('... and continue until two consecutive years of positive anomalies...'
 '.. of only negative values without being interrupted by a particularly wet year in between...'
However, I understand from the definition that if there is a positive SOIL anomaly, the drought continues until two positive anomalies are found' – I am only trying to point out that I find the explanation confusing in this respect.

Thanks for the point. We modified the paragraph for clarification to
[178-184] "A drought commences after two consecutive years of negative SOIL and continues until two consecutive years of positive anomalies occur (Coats et al., 2013). These two wet years are excluded from the drought period, ensuring that droughts consist only of negative SOIL (Kim and Raible, 2021). This definition guarantees the intensity of droughts by considering only negative SOIL without interruption by a particularly wet year in between. It also assures a minimum drought duration of two years and at least two wet years of separation between drought events."

8) Line 183 '... a substantial portion of the region experiences drought conditions …'
We corrected it. [188]

9) Line 186. 'Nevertheless, this approach avoids changes in the initial SOIL values, as ...' I do not understand what is meant here.
We wanted to mention that we use the original SOIL values without interpolation. We modified the sentences to:
[189-191] "At this step, we do not apply any horizontal interpolation in SOIL. Thus, regional coverage (geographical extension and number of grid cells) differs slightly between the models (as shown in Table 1). The reason is that the hydroclimate variables associated with precipitation can be sensitive to the horizontal grid resolution. "

10) Line 198+ 'Also, the time series of SOIL is generated by applying spatial weights to the soil moisture anomalies, taking into account the spatial extent of each grid cell within the confined region'. This is not clear to me and I think it should be explained, what is done, how is it taken into account. It is pointed out that it is somehow taken into consideration, but I suggest to indicate how.

We simply wanted to say that we generated the spatially weighted time series of soil moisture anomalies for each of the regions. We made the sentence simpler to reduce redundancy to:
[204-205] "For the latter, we generated the spatially weighted time series of SOIL for each of the regions."

Also, we moved it to [204-205] after introducing the wavelength analysis, as the time series of SOIL is principally used for this analysis.

The formula that we used to calculate the spatially weighted time series of SOIL for each of the regions, west and east, is

$$SOIL_{mean} = \sum_{i=1}^{N} SOIL_i \cos(latitude_i) / \sum_{i=1}^{N} \cos(latitude_i)$$

where N is the number of grid points in the confined regions.

We did not add this formula in the manuscript, as it is rather technical and is available as an embedded function in many programming languages.

(*) Line 214+, PCA calculation
11) Line 222: '...PCs, represented as ...Ti(t) * ui(l)
In my understanding this is not correct. Please, indicate which variables are the PCs and which the eigenvectors, also indicate where the eigenvalues are included in eq (1).
I think this part of the methods is not satisfactory. The explanation should state how the covariance (or correlation?) that is diagonalized is defined (in space or time) and then which ones are PCs, eigenvectors, EOFs, eigenvalues, etc, in a clear way. I think this influences the interpretation of the results and helps reproducibility.

We included the details pointed out by the reviewer, and corrected the paragraph and the eq 1 in the revised manuscript:

[219-225] "Given a spatiotemporal field U(t,l) where l is the spatial dimensions (latitude x longitude), and t is the time steps in years (t is the total drought years), the PCA decomposes the field into M number of modes or principal components (PC) according to the following equation

$$U = \sum_{i=1}^{M} \lambda_i . a_i x_i^{\mathsf{T}}$$

where $a_i$ is the i'th standardized PC of the U data, the $x_i$ is the i'th empirical orthogonal function of the original data (also the eigenvector), and $\lambda_i^2$ is the corresponding eigenvalue that represents the explained variance of the i'th PC ($a_i$). The resulting EOFs are orthogonal, and the PCs are uncorrelated."

12) Why 70%? Likely arbitrary but I would expect the text to say that the results are not very sensitive to this decision.

The 70% value was chosen based on the Silhouette coefficients (S) calculated using a different combination of k (number of clusters) and N (number of PCs) (Fig 1.a below). The values of S for all the models are presented in the Supplement, Figs. S1 and S2). For all models, the Silhouette coefficients for the optimal k (the higher the S, the better the performance of clustering) are better for lower N, for example, at N=3. But this N explains, in general, less than 60% of the variance.

We needed to choose N, which should include enough variability. At the same time, the performance of the clustering method using PC-applied Z500 needs to be better than using the original (non-PC-applied) Z500 fields.

[Figure]

Fig 1. a) Silhouette coefficients for a range of k clusters and N EOF for CCSM4 in Z500 during drought years in the eastern Mediterranean, and b) Percentage of explained variance with N EOF for CCSM4 in the entire Mediterranean. Yellow lines indicate 50% and 90% variance.

This led to choosing 70% of the variance, which is the N value around 5 to 6, with k around 3 to 7.

We checked that clustered patterns do not change significantly when we use higher N or the original non-PC Z500 fields (Fig. 2) with the same number of k. Therefore, as the reviewer mentioned, the method is not sensitive to N (with N>=5, which explains at least 70% of variance) after an optimal k is chosen.

[Figure]

*Fig. 2. K-means clustering applied to the Z500 PC fields with (a) 7 PCs and (b) 5 PCs and with k=3 for CESM1 (Step S3 in the manuscript). Contours indicate Z500 anomalies, and colors are the temperature anomalies. Percentage values indicate percentages of years included in each cluster.*

We included in the manuscript:
[247-248] "Once an optimal k is chosen, the PC-KCA method is not sensitive to changes in N."

13) Why should N be < or = to 7?...
Thanks for the point. That is our mistake. N needs to explain more than 70% of the variance (which leads to N of 5 or 6). We deleted that part in Fig 1.

14) I understand that the dates corresponding to the identified droughts are selected and the Z500 is considered only for those dates and then the PCA is applied. Perhaps I have skipped this explanation somewhere, but I have not found it. It should be clearly explained.

We may not have been clear with it. We included this detail in:
[211] "For this, only Z500 during drought years are considered."

15) Do the xi in Eq (2) refer to the X(t,l) in eq (1)? This does not make sense to me. Can you refer the notation of Eq (2) to that of Eq (1) so that the reader can understand how the output of the PCA feeds the KCA? In my understanding these should be PC values because you

end with N modes X t drought years, with ( I assume) a PC being a time series with t time steps including drought years.

Thanks for the point. We corrected the eq (2) to be consistent with eq (1). Also, refer to our response 11).

(eq 1) $U = \sum\limits_{i=1}^{M} \lambda_i \cdot a_i x_i^{\top}$

(eq 2) $Q(c_1, ..., c_k) = \frac{1}{t}\sum\limits_{i=1}^{t} min_{k=1,...,k} || a_i - c_k ||^2$

16) Therefore, each value in a cluster could be an array of N values corresponding to how a given date (Z500 anomaly map) is represented by those N values in the space of EOFs. However this does not fit Eq (2). I may be wrong though and other approaches may be possible. What I am trying to highlight is that 3.5 needs a clear explanation of the methodological approach and its parts, with the notation of the different parts being consistent with each other.

Refer to our responses 11) and 15). We went through the method section and tried to make the section clearer.

17) The final number of clusters is 71. However, I understand that since the analysis is performed on each model, many of those will be similar. Perhaps worth commenting this here?

Thanks for the point. Each model, period (LM or Hist), and region (western or eastern Mediterranean) results in 3-7 clusters based on the similarity of the circulation patterns. We showed these numbers in Table 2, but never mentioned them in the text, so we included this detail in the revised manuscript:

[255-256] "After Step 3, 3 to 7 clusters are obtained for each model, period, and region (Table 2), totaling 71 clusters."

Then, 71 clusters are the sum of all clusters from each model. In the next step (Step 4), we performed clustering to gather these 71 clusters based on their similarities. We mentioned this in:
[260-261] "In Step 4, KCA is applied once again to these 71 clusters (from now on, referred to as *cluster*) to group similar clusters across all models, periods, and regions."

How these 71 clusters are grouped into the final 11 patterns is presented in Table A1 in the Appendix.

18) There is quite a number of typos, please revise them. Please, also in the rest of the text. I will avoid pointing at the grammar issues, but please take care of this.
We went through the manuscript to check the typos.

GC4. (*)Section4.1

19) I agree with the comments about Fig 2 and Fig 3 in general if we consider the details, maps, correlations, etc. However, perhaps I would have a different take on the actual interpretation of them. Consider Fig. 2 first. Most models have a large low-frequency variability, with large multi-decadal or multi-centennial departures from the long-term mean. Some of them often longer than the reference period considered in Fig 2. They are to a large extent not consistent among the different models, which therefore indicates that they are more obviously related to internal variability than to the external forcing common to all experiments. If this is the behaviour of a real SOIL variable then the 1950-1979 interval considered as a reference is a very short interval of time and may correspond to a very specific state in the NOAH model, assuming that it also represents reasonably reality. However, a longer integration with NOAH or if we had more observations, would supposedly show a considerable level of low frequency variability; we do not know how much because we do not know how well the models in Fig2b represent reality.

With this in mind, the fact that one simulation represents less variability during the selected period or another one represents more, does not mean that this or that model is doing better or worse, because it is not intended that these simulations represent the real 1950-79 variability, unless it would be clearly responding to external forcing or those simulations were driven by observations, which are not. The previous reasoning extends to the other arguments related to Fig3. All arguments oriented to a better or worse representation (e.g. line 324, 325), I wouldn't agree with them, because Fig 2, indicates that except for the trends in the last decades SOIL responds to internal variability and thus the maps in Fig 3 are expected to show some level of similarity but not to represent faithfully what the NOAH model does within that comparably short period.

I think the authors should consider this argument and see how it impacts the orientation of the text and the interpretation of the figure.

Following the suggestions from the reviewer, we have modified the marked sentence to avoid evaluating the worse or better representation of the relationship between Z500 and SOIL. The new sentence is now:

[326-328] "The overall comparison of pattern correlation coefficients implies that the variability of Z500 associated with SOIL in the climate models is closer to that from NOAH-LSM–ERA5 over the western region than over the eastern region."

Also, about the reviewer's comment on low-frequency variability in the models that is longer than the reference period, we modified the sentences in [334-337] to include this detail:

[334-337] "A time series of 56 years may not include all possible variability of SOIL and Z500, including low-frequency variability on multi-decadal or longer time scales present in the model simulations (Fig. 2b). This unaccounted factor could also influence the comparison between NOAH-LSM--ERA5 and climate models, hence, the significance level of the statistical tests."

Specific details:

20) It would be good to indicate in the text or in the figure caption the time interval used for correlations.

We added the time interval in the caption for Fig. 3.

21) The maps shown in Fig 3 are indicating some relation to zonal circulation, NAO, which is mentioned in the text. If desired, this could be objectively calculated by indicating correlations with the NAO index in each model. But I would understand also that the authors would not want to go in that direction.

Thanks for the suggestion. As the goal of the section is to compare the Z500-soil moisture variability and spatial patterns between the observation-based dataset and the climate models and to assess their similarity during the present day rather than an understanding of involved circulations, we decided not to extend the discussion with the NAO index for this part.

GC5. Section4.2

22) 1. Figure 4 caption and in the text. The mean percentages of total drought years and the mean duration of droughts are calculated from the ensemble means...
I have reservations about the meaning of these numbers because of being calculated from the ensemble means. If the quantity that was being analysed would depend mostly on external forcing, I would agree, because the metrics based on the ensemble average would be meaningful as a filtered version obtaining after cancelling noise from internal variability. However, the behaviour of soil and drought occurrence here is shown and argued (e.g. Line 358) to be related to internal variability. Therefore, the statistics that would be comparable to the real world or representative for it according to each model are those of individual simulations, not the ensemble average. Arguing from a different angle: if we would have enough of a high number of runs, the ensemble average should tend to be flat and with no droughts.
The previous arguments also would justify why this happens (L 368): '... more clearly in those models and periods with one ensemble member.'

Thanks for the point. We included the values of mean duration and percentages of drought years for each period (Last millennium and Hist) with the standard deviations across the ensemble members in Tables A2 and A3 in the appendix.
As the reviewer commented, the standard deviations across the ensemble members indicate that there is some range of discrepancies between the ensemble members. We have included some details about it in the manuscript:

[359-363] "The percentage of drought years and the mean duration of droughts in Fig. 4 for each climate model and period, including their respective standard deviations, are presented in the appendix Tables A2 and A3. The tables show that the percentage of drought years and the mean duration of droughts vary across the ensemble members. For the percentage of drought years, particularly CESM1, CCSM4, and MIROC-ESM exhibit larger standard deviations during Hist. In the case of the mean duration, bcc-csm1-1 and MIROC-ESM show larger standard deviations during Hist over the eastern Mediterranean region."

[367-369] "The ensemble members of CESM1 and GISS-E2-R do not exhibit unanimous periods of low or high drought occurrence (figure not shown), which aligns with the

difference in the drought years and the duration of droughts across the ensemble members as presented in Tables A2 and A3."

23) 2. Line 371: ' This seems to agree with Cook et al (2016)'. Do you mean the simultaneous occurrence or the period or both? Please indicate more specifically and if you think it is important elaborate…

We modified the sentence to:
[384-385] "This result seems to agree with Cook et al. (2016) that have shown the simultaneous occurrence of hydroclimate variability between the western and eastern Mediterranean on a multidecadal time scale."

24) The comparison with Cook et al (2016) is interesting. For instance also in Lines 378-381. Do you think this comparison holds even if the reconstructions of Cook et al basically address the growing season (jja)? How can the statements of agreement or disagreement suffer from this? Some comments about this would be pertinent.

We repeated the wavelength coherence analysis for the annual summer soil moisture anomalies between the western and eastern Mediterranean for each model. The result is shown in Fig. 3.

[Figure]

*Fig 3. Wavelength coherence analysis between the western and eastern Mediterranean for the annual summer (JJA) time series (Similar to Fig. 5a in the manuscript).*

Overall, the result is similar to those of the annual soil moisture anomalies, showing no uniform in- or out-of-phase co-variability across time-period bands. In addition, the co-variability varies depending on the model. We briefly added this detail in the revised manuscript:
[395-396] "The analysis was also repeated for the summer (JJA) SOIL (figure not shown). The summer SOIL shows the same result as the annual variability, indicating no apparent uniform phase co-variability in the climate models."

25) GC6. Line 405: 'Although it seems contradictory that P2 depicting a negative NAO condition also occupies a significant percentage of the occurrence...

I think this could perhaps be due to the size and definition of the windows used. I do not mean that it is wrong but could be an effect of that and if so, it may be worth commenting on it. The P2 pattern favours inflow from the SW into the Iberian Peninsula. The western side of it, over Portugal and southwestern Spain should not be dry with this pattern. However, dryness could affect the lands of northern Africa and central Mediterranean Islands and over Italy. I think it is likely that the occurrence of drought with this pattern in the western box reflects the balance of wetness in the west/northwest region of the box and dryness in the rest. Perhaps it is worth assessing that and commenting.

We updated Fig. 6 with the composites of the soil moisture anomalies, and for P2, as the reviewer mentioned, we can see that the negative soil moisture anomalies are more located in the southern regions compared to the P1 patterns. We commented on this point in the revised manuscript.

[Figure]

Fig. 4. A part of the updated Fig. 6. P1 and P2 resemble a positive and negative NAO, respectively. Z500 anomalies in black contours and soil moisture anomalies in colors.

[423-425] "Additionally, during P2, negative soil moisture anomalies associated with droughts are located predominantly in the southern Mediterranean region, indicating a higher occurrence of drought conditions in the south compared to the northern Mediterranean region. In contrast, central Europe experiences wetter conditions with negative Z500 anomalies."

26) This also takes me to suggest that it would be interesting to see the composites of soil for each group of patterns. One could actually show the composites over the whole Mediterranean, not only the boxes. This does not necessarily require an increase in the number of figures. The Z500 anomalies can be shown with lines using hatching for significance and the soil pattern with shading in the same map. It would help to understand how the different patterns influence drought in the region of interest.

We added the composites of soil moisture anomalies in Fig. 6 as suggested. Also, refer to our response 25).

[409-411] "These patterns are presented in Fig. 6 with their frequencies (in the number of occurrences per century) and the mean composites of soil moisture anomalies corresponding to each circulation pattern during 850–2005 CE."

---

## Author Response (AR3)

Dear editor Dr. Goosse,

Thanks again for dealing with and accepting our manuscript for publication.
Here, we provide our responses to the editor's comments. The corrections are included in the updated version of the manuscript.
Our responses are in blue font, and the line numbers are within [the brackets].

1) Line 54. I do not follow the sentence: it is said that the last millennium had forcing conditions similar to the preindustrial era but the last millennium is within the preindustrial era for me. Instead of 'similar', do you mean 'typical' or 'representative of preindustrial conditions' ?

We mean the period with typical preindustrial forcing conditions, wit no strong changes in them. We modified the text to:

[51-54] "Among the past periods, the last millennium is particularly interesting as it is a relatively close period to the present day. The period is characterized by typical natural forcing of the pre-industrial conditions, with minimal changes, such as those in orbital parameters."

2) Line 179 'the horizontal resolutions of all data'. It is the data that is interpolated, not the resolution.

We corrected the sentence to:

[170-171] "For this calculation, all data are interpolated to match the resolutions of the coarser climate models, which are bcc-csm1-1 and MIROC-ESM."

Sincerely,

Woon Mi Kim on behalf of all co-authors.